# FLEX: A Largescale Multimodal, Multiview Dataset for Learning Structured Representations of Fitness Action Quality

## Abstract

Action Quality Assessment (AQA)—the task of quantifying how well an action is performed—has great potential for detecting errors in gym weight training, where accurate feedback is critical to prevent injuries and maximize gains. Existing AQA datasets, however, are limited to single-view competitive sports and RGB video, lacking multimodal signals and professional assessment of fitness actions. We introduce FLEX, the first large-scale, multimodal, multiview dataset for fitness AQA that incorporates surface electromyography (sEMG). FLEX contains over 7,500 multi-view recordings of 20 weight-loaded exercises performed by 38 subjects of diverse skill levels, with synchronized RGB video, 3D pose, sEMG, and physiological signals. Expert annotations are organized into a Fitness Knowledge Graph (FKG) linking actions, key steps, error types, and feedback, supporting a compositional scoring function for interpretable quality assessment. FLEX enables multimodal fusion, cross-modal prediction—including the novel Video→EMG task—and biomechanically oriented representation learning. Building on the FKG, we further introduce FLEX-VideoQA, a structured question–answering benchmark with hierarchical queries that drive cross-modal reasoning in vision–language models. Baseline experiments demonstrate that multimodal inputs, multi-view video, and fine-grained annotations significantly enhance AQA performance. FLEX thus advances AQA toward richer multimodal settings and provides a foundation for AI-powered fitness assessment and coaching. **Full Dataset and Codebase will be Open-Sourced.**

## 1 Introduction

Weight training supports cognition, muscle, and bone health but carries injury risks from poor form. Proper coaching and correct technique are essential to prevent injuries and maximize benefits. Prior research has demonstrated that computer vision and AI-based solutions can assist in this area. Specifically, videos of individuals performing exercises can be analyzed by models to detect errors in workout form. In the video understanding literature, this task is commonly referred to as *Action Quality Assessment* (AQA)—the process of quantifying how well an action, or in this context, an exercise, is performed. To train AQA models, datasets typically consist of videos of people performing actions along with corresponding annotations such as action quality scores or error labels.

However, existing fitness datasets are limited in several key aspects: **1)** they cover a narrow range of exercise actions; **2)** often exclude weight-loaded exercises; **3)** lack diversity in subject skill levels; and **4)** lack of explicit reasoning structure. To address these gaps, we introduce the **FLEX dataset**—a high-quality resource for exercise assessment and coaching. Our main contributions are:

1. **FLEX Dataset** designed to have the following desirable characteristics: Multiview—five-view markerless MoCap video; Multimodal—synchronized surface EMG, RGB video, 3D joints, point cloud, and body metrics; Largescale—7,512 samples (40+ hours, 38 subjects, 20 exercises × 10 repetitions); Weight-loaded—20 common training exercises to study injury prevention; Multi-repetition—10 repetitions per action enabling early-failure prediction; Skill-diverse—38 subjects spanning Novice, Amateur, and Expert levels. FLEX overview provided in Figure 1.

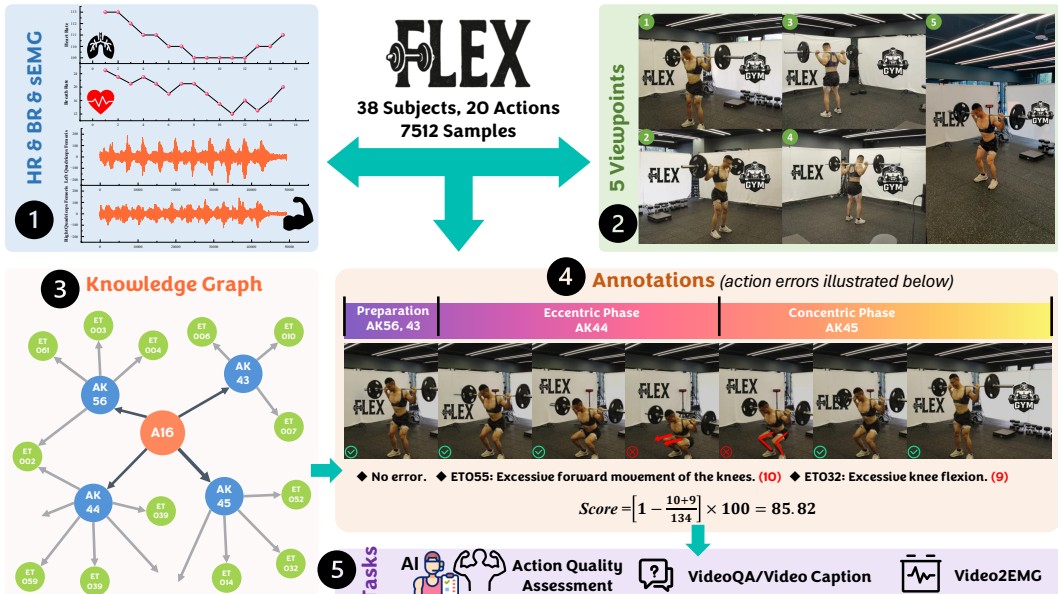

Figure 1: **An overview of the FLEX dataset.** FLEX dataset consists of a core group of 38 subjects, each performing 20 different fitness actions, repeating each action 10 times. Each action repeat was recorded from 5 viewpoints, & sEMG signals and physiological parameters (heart rate, breath rate) were simultaneously collected along with videos. The data annotations contain rich text information such as action keysteps (AK), error types (ET), & action feedback. (Zoom in for the best view.)

2. **Structured Annotations and Fitness Knowledge Graph.** Guided by domain experts, each sequence is annotated with action key steps, error types, and quality scores. These annotations are organized into a Fitness Knowledge Graph linking actions, steps, errors, muscles, and corrective feedback, and are paired with a compositional scoring function that aggregates penalties to yield interpretable action-quality metrics. Whereas existing AQA datasets typically provide quality scores without explicit reasoning structure, FLEX naturally encodes reasoning traces through this graph, enabling models to explain quality judgments by tracing from global scores to key-step penalties, specific errors, and corresponding corrective feedback.

3. **FLEX-VideoQA Benchmark.** Leveraging the FKG, we create FLEX-VideoQA, a fine-grained video question–answering benchmark with hierarchical questions ranging from coarse action recognition to fine-grained error diagnosis and causal feedback generation. This benchmark evaluates a model's ability to perform multi-hop, compositional reasoning over multimodal signals.

4. **Multimodal, Cross-modal, and Biomechanically-oriented Representation Learning.** FLEX supports diverse representation-learning paradigms—multimodal fusion, cross-modal prediction, and biomechanically grounded learning—by pairing synchronized RGB video, 3D pose, surface EMG, and physiological signals. Baseline models for action quality assessment and Video→EMG prediction demonstrate that visual features can be trained to infer hidden muscle activation, while experiments with FLEX-VideoQA showcase cross-modal reasoning in vision–language models.

These contributions establish FLEX as a comprehensive platform for studying fine-grained human action, enabling multimodal & biomechanically grounded representation learning & reasoning.

## 2 RELATED WORK

**Action Quality Assessment Datasets.** Datasets are essential to machine learning and AQA, supporting model design and development. Many high-quality AQA datasets have emerged across diverse domains, including sports Pirsiavash et al. (2014); Parmar & Tran Morris (2017), healthcare Gao et al. (2014); Vakanski et al. (2018); Capecci et al. (2019), fitness Parmar et al. (2022); Li et al. (2024); Ogata et al. (2019), industrial training Sener et al. (2022), and generative AI evaluation Chen et al. (2024). Early datasets, such as MIT-AQA and UNLV-AQA contained only single-action categories.

Table 1: **Comparison of FLEX dataset with representative AQA datasets in terms of sample size, action types, views, modalities, annotations, skills level coverage & sources.** V: Video, Sk: Skeleton, sE: sEMG, P: Physiological Info., S: Score, G: Grade, A: Action, F: Formation, D: Description, E: Error, Fb: Feedback. N: Novice, Am: Amateur, Ex: Expert, Nor: Normal, Abn: Abnormal, Rehab: Rehabilitation.

| Dataset | Sample | Type | View | Modality | Annotation | Skill Levels | Domain | Source |
|---|---|---|---|---|---|---|---|---|
| MIT-Dive Pirsiavash et al. (2014) | 159 | 1 | 1 | V | S | Ex | Sport | Web |
| UNLV-Dive Parmar & Tran Morris (2017) | 370 | 1 | 1 | V | S | Ex | Sport | Web |
| AQA-7 Parmar & Morris (2019a) | 1189 | 7 | 1 | V | S, A | Ex | Sport | Web |
| MTL-AQA Parmar & Morris (2019b) | 1412 | 2 | 1 | V | S, A, D | Ex | Sport | Web |
| Waseda-Squat Ogata et al. (2019) | 2001 | 1 | 1 | V, Sk | E | N | Fitness | Camera |
| TASD-2 Gao et al. (2020) | 606 | 2 | 1 | V | S, A | Ex | Sport | Web |
| Rhy.Gym. Zeng et al. (2020) | 1000 | 4 | 1 | V | S, A | Ex | Sport | Web |
| QMAR Sardari et al. (2020) | 306 | 2 | 6 | V, Sk | S, A | Nor, Abn | Rehab | Camera |
| FR-FS Wang et al. (2021) | 417 | 1 | 1 | V | G, A | Ex | Sport | Web |
| Fitness-AQA Parmar et al. (2022) | 13049 | 3 | 1 | V | G, A | Am | Fitness | Web |
| FineDiving Xu et al. (2022) | 3000 | 52 | 1 | V | S, A | Ex | Sport | Web |
| LOGO Zhang et al. (2023) | 200 | 12 | 1 | V | S, A, F | Ex | Sport | Web |
| FineFS Ji et al. (2023) | 1167 | 1 | 1 | V | S, A | Ex | Sport | Web |
| EgoExo-Fitness Li et al. (2024) | 6131 | 12 | 6 | V | S, A, D | N, Am | Fitness | Camera |
| LucidAction Dong et al. (2024) | 6702 | 8 | 8 | V, Sk | S, A | Am, Ex | Sport | MoCap |
| **Ours FLEX** | **7512** | **20** | **5** | **V, Sk, sE, P** | **S, A, D, E, Fb** | **N, Am, Ex** | **Fitness** | **MoCap** |

This was followed by multi-action and multimodal datasets such as AQA-7 and MTL-AQA (contains action scores, fine-grained action class labels, verbal description of action quality). Datasets like FineDiving split the actions into clips, representing a trend toward more detailed annotation.

Emergence of datasets like Egoexo-Fitness and LucidAction represents a shift from crawling online videos toward in-person collection of AQA data, despite being resource-intensive. This approach guarantees control over both the dataset scale and quality. Advances in AQA datasets have contributed to developing AQA methodologies, with multimodal, fine-grained, and multi-action datasets allowing models to be more comprehensive and fine-grained, improving model generalization capabilities. However, existing datasets predominantly focus on competitive sports actions, with only a few dedicated to the fitness domain (for details see Table 1 and Appendix A). Moreover, current fitness datasets primarily include self-loaded actions Ogata et al. (2019); Li et al. (2024), exhibit narrow action categories Parmar et al. (2022), or are constrained to RGB modalities Ogata et al. (2019); Parmar et al. (2022); Li et al. (2024), thereby limiting the application and innovation. To bridge those gaps and stimulate innovation in AI-based Fitness coaching, we develop FLEX—first AQA dataset that provides multi-view videos with synchronized 3D pose, sEMG, and physiological information. Furthermore, we develop detailed, professional, comprehensive annotation rules for fitness actions that integrate multiple knowledge sources. These rules form a KG containing Action Keysteps, Error Types, and Feedback, making the annotation rich in paired textual information.

**Electromyography (EMG) Datasets.** EMG records electrical activity from skeletal muscles to assess their activation levels, motor coordination, and detect abnormalities. EMG devices are broadly classified into surface EMG (sEMG), which uses non-invasive skin electrodes, and needle EMG (nEMG), which involves inserting fine needles into muscle tissue ALCAN & ZİNNUROĞLU (2023). Owing to its non-invasive nature, existing EMG datasets predominantly utilize sEMG for applications in biomechanical analysis Zhang et al. (2017); Hug et al. (2019); Jarque-Bou et al. (2020); Khan et al. (2020); Dimitrov et al. (2023); Wang et al. (2023), hand gesture estimation Atzori et al. (2014); Liu et al. (2021); Ozdemir et al. (2022); Salter et al. (2024), etc. Existing EMG datasets mainly link signals to motion categories, overlooking their connection to action quality. Since EMG reflects neuromuscular control and muscle function—key in fitness—activation patterns serve as objective quality markers. FLEX is the first AQA dataset to integrate EMG, using a 16-channel sEMG system FASTMOVE (2024a) at 2 KHz to record target muscle activity during fitness movements. sEMG & MoCap were synchronized for precise temporal alignment.

## 3 FLEX DATASET

We present FLEX, a dataset that addresses key limitations in existing fitness AQA datasets, such as limited exercise variety, absence of weight-loaded actions, and low skill diversity. To our knowledge, FLEX is the first weight training dataset to include *multiview multimodal* recordings, detailed exercise

procedures, standardized assessment rules, and corrective feedback. This section outlines data collection, annotation, quality checks, procedures, and evaluation standards.

FLEX dataset advances action quality assessment beyond traditional video benchmarks by combining *five synchronized RGB views with 3D motion-capture, surface EMG, and physiological signals*. It includes 20 weight-training actions performed by 38 subjects of varying skill levels, each repeated 10 times. By pairing video with sEMG—capturing hidden muscle activity—FLEX enables learning not only what movements are performed but also how they are executed internally.

To make this rich sensor data useful, we built a *Fitness Knowledge Graph (FKG)* that encodes the structure of each exercise. Every action is decomposed into ordered key steps—such as the setup, eccentric phase, and concentric phases of a squat. Within each step, annotators list common errors like knee valgus or back rounding, connect them to the primary and secondary muscle groups involved, and attach recommended feedback for correction. These nodes and edges form a hierarchy of actions, steps, errors, muscles, and advice. As a result, every clip in FLEX is not just tagged with a single action quality label or score, but grounded in a web of relationships that describe both the mechanics of the motion and the typical ways it can fail and actionable feedback to rectify it.

We formalize this structure into a *compositional scoring function*. Each key step carries a weight reflecting its importance to overall performance. Within a step, each error type has a penalty coefficient scaling with the observed severity of that error. The final quality score is computed by summing weighted step scores and subtracting accumulated penalties, producing a number that reflects the layered decomposition of the action. Since penalties for individual errors roll up into step scores, and step scores roll up into the global score, the metric mirrors the graph's hierarchy and provides interpretable pathways from low-level mistakes to the final assessment.

## 3.1 PRELIMINARY SETUP

**Action/Exercise Selection.** Bodyweight exercises require no equipment, making them simpler and less injury-prone. In contrast, weight-loaded exercises involve lifting external weights and carry a higher injury risk if done improperly. Thus, proper technique and posture are crucial in weight training to prevent injuries. To support this, we carefully selected 20 common exercises based on the following criteria: **1)** Include exercises targeting both upper and lower body muscle groups. **2)** Cover complex, injury-prone

Table 2: **Comparison between FLEX and existing SOTA fitness datasets.** EWL: equipment (barbell, dumbbell, etc.)-based weight loading; RI: level of risk of injury.

| Dataset | Exercises | EWL | RI |
|---|---|---|---|
| Fitness-AQA Parmar et al. (2022) | 3 | ✓ | high |
| EgoExo-Fitness Li et al. (2024) | 12 | ✗ | low |
| **Ours FLEX** | **20** | ✓ | high |

joints, such as the shoulders. **3)** Feature a variety of higher-risk free-weight exercises. **4)** Focus on widely practiced exercises to ensure broad relevance, aligning with commonly recommended training regimens. A comparison of our dataset with Fitness-AQA and EgoExo-Fitness is provided in Table 2.

**Diverse Subject Pool.** To capture diverse skill levels, we recruited subjects across Novice, Amateur, and Expert skill brackets through surveys at our institutions and local gyms, receiving over 60 registrations from students and coaches. Each applicant completed an online questionnaire, an offline ability test, and 3D/body composition analysis to assess weight-bearing capacity and skill level. Following prior work Li et al. (2024), we selected a total of 38 subjects: 10 experts (ID: P01–P10), 8 amateurs (ID: A01–A10, excluding A06 and A09), and 20 novices (ID: N01–N10). Unlike existing datasets, ours spans all skill levels, as shown in Table 1 and Table 2.

**Recording Multiple Continuous Exercise Repetitions.** To support modeling the temporal evolution of action quality over time, each subject was required to perform 10 repetitions per action. In comparison, existing datasets lack this multirepetition capture of weight training.

**Human subject safety.** Details on ensuring human subject safety provided in the Appendix B.2.

## 3.2 DATA COLLECTION

**Multiview Capture.** Fitness actions often involve complex postures, with key joints occluded when viewed from a single angle. To obtain more complete ground-truth motion, we used a high-precision markerless motion capture system FASTMOVE (2024b) with four synchronized ZCAM E2M4

cinema cameras ZCAM (2020) (Olympus 14–150 mm lens fixed at 14 mm OLYMPUS (2022)), recording 4K video at 120 FPS. For accessibility, we also included a smartphone view from a OnePlus 7 OnePlus (2019), captured at 1080p and 60 FPS. This ubiquitous smartphone camera view supports "in-the-wild" generalizability, significantly improving its real-world relevance, robustness, and practical deployability for fitness-AI applications—without compromising multimodal synchronization.

**Multimodal information.** Existing datasets focus on videos Pirsiavash et al. (2014), text Parmar & Morris (2019b), skeletal points Capecci et al. (2019), or audio Parmar et al. (2021), with little attention to physiological data. EMG, in particular, captures deep muscular activity closely tied to action quality, making it highly relevant for fitness AQA and deeper feedback. Accordingly, the FLEX dataset records: **1)** EMG signals of target muscle groups using a 16-channel FastMove sEMG FASTMOVE (2024a) device operating at a sampling rate of 2000 Hz. sEMG captures muscle activity. **2)** Heart rate, collected with the help of a customized wearable vest, provides insight into a person's exertion rate. **3)** Respiratory rate: Exercise requires proper and procedural breathing. Our exercise procedure base and rules include breathing as an AQA criterion.

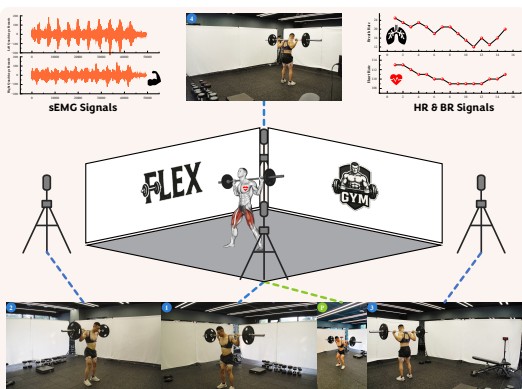

Figure 2: **Data collection environment.** Four cinema cameras and one smartphone camera to support in-the-wild generalization were fixed at the four corners of the collection area. Video, sEMG, heart rate, and breath rate are recorded synchronously during collection.

**Collection process.** Data collection began by informing each subject of the required actions. Staff assisted subjects in wearing the customized vest and attaching sEMG sensors to target muscles to ensure secure attachment of sensors to skin to minimize noise in the collected signals. Unlike FLAG3D Tang et al. (2023) and Egoexo-Fitness Li et al. (2024), subjects entered the collection area individually to avoid observing others or receiving detailed instructions, minimizing external influence. No textual descriptions were provided; instead, a brief demonstration preceded each session. After sensor attachment, subjects faced the curtain between Viewpoints 1 and 2 and, upon the staff's signal, performed the action 10 times. The staff then recorded the corresponding weight data, with the entire process supervised by the authors.

### 3.3 DATA ANNOTATION

The value of a high-quality dataset relies not only on the accuracy and completeness of the raw data but also significantly on the precision and consistency of the annotations. During the data collection phase, we employed multiple devices to acquire multimodal raw data across diverse actions, maintaining close monitoring throughout to guarantee data quality. As elaborated in the following sections, the FLEX dataset implemented stringent measures in formulating annotation rules and selecting annotators to ensure further rigorous annotation.

#### 3.3.1 DEVELOPING STANDARDIZED FITNESS RULES AND KNOWLEDGE-BASE

Unlike competitive sports (e.g., diving, gymnastics) with clearly defined action objectives and scoring criteria, fitness actions aim to serve diverse goals such as muscle growth, fat loss, and rehabilitation. As such, different individuals might adopt varying evaluation standards for the same action. To guarantee high-quality annotation for the FLEX dataset, we extensively gathered existing fitness action standards from multiple sources, including national standards, fitness associations, professional literature, fitness applications, and influential fitness content creators. We established a precise and objective annotation system through comprehensive integration and refinement.

**Sources of Fitness Action Standards.** Specifically, our annotation rules primarily drew upon the "National Occupational Skill Standards—Social Sports Instructor (Occupation Code: 4-13-04-01)" jointly formulated by the Ministry of Human Resources and Social Security of the People's

Republic of China and General Administration of Sport of China Ministry of Human Resources and Social Security of the People's Republic of China & General Administration of Sport of China (2020). Additionally, we utilized the "Fitness and Bodybuilding Tutorial" from Beijing Sport University University (2013), "Occupational Competency Training Textbook for Social Sports Instructors—Fitness Coaches (with Technical Action Videos)" published by the Human Resources Development Center of the General Administration of Sport of China of the General Administration of Sport of China (2023), and "Joe Weider's Bodybuilding System" Weider (1998).

**Developing Standardized Annotation Rules through Multi-Source Integration and Expert Review.** We thoroughly reviewed all sources, meticulously documenting essential information relevant to the collected actions, emphasizing target muscles, action descriptions, and potential error types. Through rigorous comparison and integration, we distilled these insights into annotation rules that adhere to existing standards while accommodating practical training scenarios. These rules underwent rigorous verification by experts from an advanced sports institution, ensuring both theoretical rigor and enhanced consistency and practicality in annotations.

**Biomechanically-Inspired Action Phase and Keystep Modeling** To guide the annotation process and ensure consistency, we developed a biomechanically-inspired framework for structuring each action. In this framework, every collected action is decomposed into three distinct phases based on dominant limb movement patterns: Preparation, Concentric, and Eccentric. Within each phase, the primary limb actions are further subdivided into finer-grained action keysteps (AKs), each paired with integrated textual descriptions. By systematically combining these AKs, we generate rich textual representations of diverse fitness actions, supporting fine-grained action analysis and structured representation learning. Building on this framework, we next introduce extensions for error identification, structured knowledge representation, and annotation efficiency.

**Error Type Identification and Feedback Design.** Additionally, we identified specific error types (ET) associated with localized joint actions within each major limb action and provided corresponding corrective feedback suggestions.

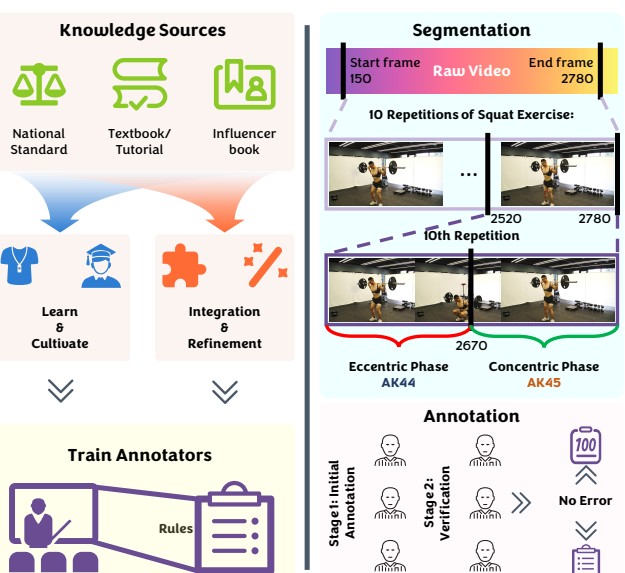

Figure 3: **Annotation Process.** Annotators were trained on the provided guidelines and received centralized instruction to ensure full understanding of the rules. The video data was segmented following predetermined criteria, and a two-stage annotation process was implemented to reduce annotation errors and mitigate subjective bias.

**Fitness Action Knowledge Graph (FKG) Construction.** We constructed a FKG (see in the Appendix B.3.1) comprising of actions, action keysteps (AK), error types (ET), feedback, and their relationships.

**Scoring System for Annotation Efficiency.** Furthermore, different error types were assigned distinct weights within a penalty-based scoring system, enabling annotators to efficiently identify and label errors from action clips. The final scores were obtained through cumulative penalties, ensuring both efficiency and consistency throughout the annotation process (provided in the Appendix B.3.4).

### 3.3.2 ANNOTATION PROCESS

We annotate the following information for each sample: a) **Action segmentation**; b) **Action keysteps**; c) **Action Errors**; d) **Action Quality Assessment Scores** using the following procedure. The whole annotation process is visualized in Figure 3.

1. **Annotator Recruitment.** We carefully recruited 16 fitness professionals and practitioners to annotate our dataset. For details on the recruitment procedure, please refer to the Appendix B.3.2.

2. **Action Segmentation.** During data segmentation, annotators split videos of repeated actions into individual samples following established guidelines and verified each sample.

3. **Task Organization and Annotator Training.** During annotation phase, tasks were organized by target muscle groups into separate work periods. At the start of each period, all 16 annotators received centralized training to ensure consistent understanding of the guidelines.

4. **Annotator Grouping by Experience.** Based on their fitness experience, annotators were organized into two groups: 1) Group 1—trainers with < 3 years of experience and master's students—handled the initial annotation (Stage 1), while 2) Group 2—trainers with > 3 years of experience and doctoral students—was responsible for annotation verification (Stage 2).

5. **Two-Stage Annotation & Verification Workflow.** Stage 1: Each sample was first annotated for error types by three annotators from Group 1, using both video and 3D pose data. Stage 2: These annotations were then reviewed by three annotators from Group 2. If discrepancies arose, the final label was decided by majority vote among the reviewers to reduce subjective bias.

6. **Quality Control Measures.** Authors periodically monitored the quality of the annotated data to ensure that the dataset maintained a high degree of accuracy and reliability.

## 3.4 BENCHMARK DESIGN

Leveraging the rich factorial structure of FLEX—covering varied actions, multiple camera viewpoints, and a diverse set of subjects—we are able to define flexible data partitions and benchmarking settings. This design facilitates the construction of evaluation protocols at different difficulty levels, thereby enabling a systematic study of both in-distribution accuracy and out-of-distribution generalization.

First, we provide a standard split that is consistent with mainstream AQA datasets. FLEX adopts a *Vanilla Split protocol:* within each action class, samples from different subjects and repetitions are mixed and then randomly partitioned into training and test sets. This protocol is currently the most widely used in the AQA community and is mainly employed to assess the upper-bound performance of models on specific actions.

Second, we introduce three more challenging and generalization-oriented split protocols that are closer to real-world applications:

- *Cross-Subject*: the training and test sets consist of disjoint subject groups, used to evaluate cross-person generalization;
- *Cross-View*: the model is trained on one camera view and tested on another unseen view, simulating distribution shift caused by viewpoint changes;
- *Mix-View*: the model is trained on three views and tested on the remaining unseen view, used to assess generalization to unknown viewpoints under partially multi-view supervision.

In summary, these settings naturally induce a hierarchy of evaluation difficulty. We advocate validating methods on the random split as a strong in-distribution baseline, followed by the cross-subject, cross-view, and mix-view splits to examine real-world generalization under person and viewpoint shifts comprehensively.

## 3.5 FLEX-VIDEOQA DATASET

To extend FLEX beyond action quality scoring, we introduce FLEX-VideoQA, a fine-grained video question–answering benchmark built directly on top of the dataset's multimodal recordings and the Fitness Knowledge Graph (FKG). This benchmark is designed to test a model's ability to perform structured, multimodal reasoning about human action—capturing not only visual appearance but also the underlying biomechanical and semantic relationships encoded in the FKG. Furthermore, in non-AQA VideoQA datasets, questions focus on scenes or coarse actions (e.g., what someone is doing, what they will do next, etc.). In contrast, we target fine-grained errors in exercises—posture, movement, and corrective feedback—which are absent from existing VideoQA datasets.

**Dataset Construction.** The FKG provides a rich ontology of actions, key steps, common error types, muscle groups, and corrective feedback. From this graph, we generate a large set of natural-language questions and answers by applying a mixture of rule-based templates and LLMs. Since every question is grounded in explicit graph nodes and relations, the resulting QA pairs are both semantically precise

and verifiable. These QA sets are then **manually verified** and corrected if needed. In total, we collect 30048 QA pairs. The questions span several categories:

- *Descriptive*: ask for observable attributes such as the action being performed or the current keystep (e.g., "Which exercise is shown in the video?", "What keystep is the performer currently executing?").
- *Relational/reasoning*: requires linking two or more graph entities (e.g., "Which muscle group is primarily activated during the descent phase of a squat?" or "Which error in the setup step leads to a lower-back penalty?").
- *Temporal*: probe understanding of sequential dependencies (e.g., "Which key step follows the liftoff phase?").
- *Causal/feedback*: ask for corrective advice given an observed error (e.g., "What feedback should be given if the knees cave inward during the ascent?").

**Comprehensive multimodal reasoning platform.** By transforming the FKG's structured annotations into a large set of graph-grounded question–answer pairs, FLEX-VideoQA evolves FLEX from a dataset for action quality assessment into a comprehensive multimodal reasoning platform. It provides a rigorous testbed for training and evaluating VLMs that must not only recognize human actions but also reason over the hierarchical relationships among actions, key steps, muscle activations, common errors, and corrective feedback. The supervised fine-tuning experiments underscore the value of structured graph supervision for enabling VLMs to acquire representations of action quality and underlying physiology that go well beyond surface-level video understanding.

## 4 EXPERIMENTS

FLEX dataset's Multimodal data, Fitness Knowledge Graph, and the fine-grained annotations not only support detailed action quality assessment but also serve as a rich resource for diverse research scenarios such as chatbot-based fitness assessment and coaching (involves VideoQA) and crossmodal signal estimation. To foster future research on FLEX, we have designed and conducted multiple benchmark evaluations on the dataset to demonstrate its broad application potential.

### 4.1 ACTION QUALITY ASSESSMENT

**Baselines.**

We first investigate the impact of various modalities individually and their combination for the task of AQA. Given the wide adoption and strong performance of: a) CoRe and TPT on video-based AQA, we consider them for video-based fine-grained action modeling; b) STGCN and SkateFormer on pose-based action recognition, we consider them for pose-based fine-grained action modeling. For EMG modality, we develop a comparative model—computing relative muscle contribution and concatenating it with video and pose features. Based on these models, we develop the following baselines. 1) **Unimodal models**: We compared model performance across modalities, thereby isolating each modality's independent contribution. 2) **Multiview**: This model uses multi-view video, based on the hypothesis that combining perspectives provides a more complete view of posture and movement, enhancing AQA performance. 3) **Multiview incorporating explicit Pose information**: We hypothesize that pose features can enhance the video-based AQA model. Thus, this model builds on MV model and incorporates pose features extracted via an ST-GCN Yan et al. (2018) or SkateFormer Do & Kim (2024). 4) **Multiview incorporating explicit Pose information and Muscle-level Physiological information**: sEMG offers direct insight into muscle activity, valuable for AQA. This model extends MV+Pose by integrating sEMG. We do this by computing the relative muscle contribution and concatenating

Table 3: **Performance of AQA models on FLEX dataset.** $R - l2(\times 100)$

| AQA Model | $\rho \uparrow$ | $R - \ell_2 \downarrow$ |
|---|---|---|
| CoRe Yu et al. (2021) | 0.8069 | 1.7582 |
| TPT Bai et al. (2022) | 0.8111 | 1.5464 |
| Qwen Bai et al. (2025) | 0.0575 | 3.1183 |
| Qwen-SFT Bai et al. (2025) | 0.0495 | 3.0592 |
| ST-GCN Yan et al. (2018) | 0.8528 | 1.3273 |
| SkateFormer Do & Kim (2024) | 0.8145 | 1.6645 |
| EMG | 0.3191 | 5.1019 |
| Multiview (MV) | 0.8974 | 0.9095 |
| MV + SkateFormer | 0.8982 | 0.9064 |
| MV + ST-GCN | 0.9003 | 0.8920 |
| MV + SkateFormer + EMG | 0.8996 | 0.9017 |
| MV + ST-GCN + EMG | 0.9019 | 0.8877 |
| *More Challenging Generalization-Oriented FLEX Splits* | | |
| Cross-Subject Split | 0.4288 | 5.9255 |
| Cross-View Split | 0.3641 | 4.8313 |
| Mix-View Split | 0.3897 | 3.6744 |

it with video and pose features. 5) **Generalization-oriented AQA splits**: To assess the model's robustness beyond standard random splits, we evaluate CoRe performance on disjoint protocols. Cross-subject evaluation for subject-independence, and Cross-view/Mix-view settings to verify the model's invariance to camera perspective changes.

**Implementation details.** Provided in the Appendix C.1.1. **Metrics.** Following standard practice, we report Spearman's rank correlation ($\rho$) and $R - l2$ metrics. More details are provided in the Appendix C.1.2.

**Results.** Results of AQA models are summarized in Table 3. In unimodal settings, TPT and CoRe perform comparably; we adopt CoRe for subsequent experiments owing to its lower computational demands and more streamlined design. Surprisingly, even SOTA VLMs—despite extensive pretraining and supervised fine-tuning—perform poorly on Vanilla AQA. Pose alone—implemented via ST-GCN & SkateFormer—outperforms single-view video, as skeletal points encode fine-grained kinematics. sEMG alone yields lower performance, yet provides complementary information by capturing muscle activation and motor effort not visible in video or pose. This complementarity is confirmed in multimodal configurations: adding pose to multiview video improves performance, indicating that structural pose features enrich visual dynamics, while further integrating sEMG delivers the best overall results ($\rho = 0.9019$, $R - \ell_2 = 0.8877$). By explicitly modeling external kinematics (pose) and internal physiological states (sEMG), the multimodal framework furnishes a more comprehensive and mechanistic basis for action quality assessment. While the significant performance gap between standard and disjoint protocols highlights the severity of domain shifts in AQA. For more detailed analysis, please refer to the Appendix C.1.3.

## 4.2 ACTION QUALITY UNDERSTANDING

We envision a future where people can use chatbots as AI-Coaches, who can understand and monitor people's exercise posture and actions and provide corrective feedback on it. Vision-Language Models (VLMs) or Multimodal Large Language Models (MLLMs) are a natural choice to implement such AI-Coach because of their capability to conduct multimodal communication and processing. However, out of the box, these models might not have specialized skills/knowledge to perform fitness action quality assessment. We believe our multimodal data, the fitness action knowledge graph, and the fine-grained annotations in the FLEX dataset can bridge this gap. To that end, we create **FLEX-VideoQA dataset** (Sec. 3.5) containing fitness-based video question-answering samples. This dataset unlocks the integration of VLMs/MLLMs in AQA research.

**Setup.** To comprehensively evaluate the performance of existing VLMs on the FLEX-VideoQA dataset, we selected several recently released SOTA models, including MiniCPM-O-2.6-8B Yao et al. (2024), InternVL3-2B Zhu et al. (2025), and Qwen2.5-VL-3B Bai et al. (2025). We evaluated their performance under three configurations: **1) Off-the-Shelf mode**: Using the original pretrained weights. **2) Rules-based Prompting mode**: Providing standardized action guidelines during the evaluation phase. **3) Supervised fine-tuning mode**: Applying a LoRA strategy to fine-tune the entire model on top of the pretrained weights. Further implementation details are provided in Appendix C.2.1.

**Metrics.** To comprehensively assess VLMs' outputs, we employed a wide range of commonly used metrics in VideoQA research (BLEU, ROUGE-L, METEOR, CHRF++, and BERTScore-F1), as metrics of generated text, alongside $R - \ell_2$ to evaluate score-prediction accuracy. More details are provided in the Appendix C.2.2

**Results.** Analysis of the results in Table 4 shows that supervised fine-tuning delivers an order-of-magnitude improvement in text-generation quality compared to the pretrained model, yet only yields marginal improvements in consistency of action scoring. The results show significant improvement in semantic alignment between fitness terminology and visual entities—*not covered in mainstream video understanding datasets, but covered in our dataset.* Im-

Table 4: **Performance of VLMs on FLEX-VideoQA dataset.** BF1: BERTScore F1, SFT: Supervised Fine-Tuning. $R - \ell_2(\times 100)$

| Metrics | Pretrained | | | Prompt | | | SFT |
|---|---|---|---|---|---|---|---|
| | InternVL | MiniCPM | Qwen | InternVL | MiniCPM | Qwen | Qwen |
| **BLEU** ↑ | 0.0003 | 0.0154 | 0.0388 | 0.0007 | 0.0312 | 0.0415 | **0.1970** |
| **ROUGE-L** ↑ | 0.1689 | 0.2902 | 0.2012 | 0.2002 | 0.3102 | 0.2128 | **0.4010** |
| **METEOR** ↑ | 0.1405 | 0.3024 | 0.3049 | 0.1736 | 0.3345 | 0.3206 | **0.4688** |
| **CHRF++** ↑ | 8.7228 | 20.4180 | 36.1340 | 17.3078 | 28.4865 | 39.5308 | **52.0005** |
| **BF1** ↑ | 0.1017 | 0.2535 | 0.1664 | 0.1550 | 0.2743 | 0.1666 | **0.4390** |
| $R - \ell_2$↓ | 3.9900 | 10.5539 | 3.1183 | 3.2757 | 8.6448 | 3.0942 | **3.0592** |

portantly, across all three evaluation configurations, Qwen consistently achieves the best overall performance, even surpassing larger models such as MiniCPM. In addition, prompt engineering brings moderate but noticeable improvements over the pretrained baseline, further validating the benefit of simple adaptation strategies before full fine-tuning. However, $R - \ell_2$ decreases by less than 2%, indicating that the current fine-tuning strategy optimizes the language-generation head and contributes minimally to the regression head, which relies on fine-grained pose differences. While current VLMs still underperform specialized AQA models in precise action scoring, their ability to generate natural, professional action descriptions offers a viable path for integrating VideoQA task with AQA task. For more detailed analysis, please refer to the Appendix C.2.3.

### 4.3 NOVEL TASK–VIDEO2EMG

Surface-EMG (sEMG) signals provide crucial feedback on muscle functional state and correlate with action quality. However, acquiring sEMG data is costly. To mitigate that, we envision a future where EMG signal could be estimated directly from videos. Such technology would enable providing precise—muscle-level—feedback and intervention in cost-effective manner. To that end, we introduce a new task of estimating sEMG signals from videos. Our FLEX dataset with synchronized recordings of videos and EMG signals can enable this new research direction. **Video2EMG Baseline Model:** Inspired by the emg2pose Salter et al. (2024) framework, we designed end-to-end visual regression models that integrate a visual

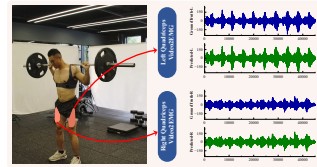

Figure 4: **Qualitative result of Video2EMG.** Notice the faithful prediction of EMG.

encoder (ResNet or ViT) with an sEMG sequence predictor (LSTM or stacked SVR). *Implementation details* provided in Appendix C.3.1. Code will be released.

Table 5: **Performance of Video2EMG models.**

| Model | Vanilla | | | Cross-View | | | Cross-Subject | | |
|---|---|---|---|---|---|---|---|---|---|
| | CCP ↑ | MAE ↓ | RMSE ↓ | CCP ↑ | MAE ↓ | RMSE ↓ | CCP ↑ | MAE ↓ | RMSE ↓ |
| ResNet+LSTM | 0.3706 | 0.1655 | 0.2133 | 0.2723 | 0.2550 | 0.3054 | 0.2872 | 0.2934 | 0.3442 |
| ResNet+SVR | 0.4174 | 0.1466 | 0.1878 | 0.2793 | 0.1508 | 0.2002 | 0.3062 | 0.1705 | 0.2170 |
| ViT+LSTM | 0.4040 | 0.1648 | 0.2069 | 0.3104 | 0.2079 | 0.2619 | 0.3086 | 0.2148 | 0.2655 |
| ViT+SVR | 0.4345 | 0.1465 | 0.1882 | 0.3311 | 0.1483 | 0.1985 | 0.3311 | 0.1631 | 0.2104 |

**Metrics.** To evaluate Video2EMG regression, we report mean absolute error (MAE) and root mean square error (RMSE), reflecting average error and variability. Furthermore, we use the cross-correlation peak (CCP) to quantify the waveform shape.

**Results.** After 150 training epochs under 3 different protocols, Video2EMG models exhibited strong regression performance (see in Table 5). Following inverse normalization, they achieved the best MAE of 0.1465, RMSE of 0.1882, and CCP of 0.4345 on the vanilla split, demonstrating accurate sEMG prediction for most samples with reasonably bounded error variability. Under the cross-subject and cross-view splits, however, performance degrades noticeably, showing that the dataset and the Video2EMG task remain challenging and far from solved, with substantial room for future improvements. These results serve as baselines for future work. For more detailed analysis, please refer to the Appendix C.3.2.

### 5 CONCLUSION

We presented FLEX, a large-scale, multimodal dataset for fitness action quality assessment that pairs five-view RGB video, 3D pose, surface EMG, and physiological signals across diverse weight-loaded exercises. FLEX is the first AQA resource to integrate sEMG and a Fitness Knowledge Graph, providing structured annotations and penalty-based scoring that capture relationships among actions, key steps, errors, and corrective feedback. Baseline experiments—including multimodal AQA, VideoQA, and Video→EMG prediction—demonstrate the potential of FLEX for biomechanically grounded representation learning and structured reasoning. We hope FLEX will catalyze future research in multimodal fitness understanding, cross-modal prediction, and interpretable AQA research.

ETHICS STATEMENT

Our work adheres to the ICLR Code of Ethics, emphasizing responsibility, fairness, transparency, and the minimization of harm in research and its applications.

**Subjects Protection, Safety, and Privacy.** To ensure the safety of participants during the collection of risk-prone exercise data, all subjects performed movements at approximately 80% of their maximum weight-loaded capacity. Safety supervisors were present on-site throughout the sessions, with emergency medical kits available. Additionally, all participants were covered by sports injury insurance. The research protocol was reviewed and approved by the Institutional Review Board (IRB), and written informed consent was obtained from all participants prior to data collection. To further protect participant privacy, all facial regions in the figures were blurred. Access to the dataset requires signing a License Agreement, which restricts usage to non-commercial academic research only, ensuring rigorous protection of personal data.

**Fair and Responsible Compensation.** Given the differences in testing intensity, we implemented a differentiated compensation scheme: expert subjects received compensation equivalent to roughly 15 times the local minimum hourly wage for completing all tests within approximately two hours, while other subjects, who completed their sessions across three days ($\sim$30 minutes per day), were compensated at a rate equivalent to about 7 times the local minimum hourly wage per day. Annotators were also compensated at a rate higher than the local minimum hourly wage, reflecting our commitment to fairness and ethical treatment of research contributors.

**Fairness and Inclusivity.** We issued an open call for participation in our data collection, welcoming individuals from diverse genders, ethnic groups, regions, and socioeconomic backgrounds. Participation from women was not observed in this phase of data collection. Contributing factors include the institute's remote location (distant from urban centers), local demographic imbalances, and sociocultural dynamics. All recruitment followed established ethical principles in AI and data science, including informed consent, voluntary participation, non-coercive recruitment, and transparent communication of dataset limitations. To date, existing literature on AQA has not reported gender-specific performance differences in models. To continue strengthening inclusivity, we have initiated collaborations with partner institutions. Additional contributions from female subjects are already underway and are expected to be integrated into the dataset by the end of this year, further enriching its diversity and representativeness.

**Awareness of Potential Negative Societal Impacts.** The capability of measuring the quality of actions and movements may be used in ways that users/humans may not approve/agree. For example, action quality assessment may be misused by agencies such as health insurance providers to forecast future injuries and diseases reliably. Such health insurance providers may not then unjustly cover future injury-prone clientele. Awareness is the first step towards the solution. We believe our work helps create this awareness. Moreover, detecting bias of this nature in the decision-making process can help provide more equitable service and consideration to society.

REPRODUCIBILITY STATEMENT

Due to the double-blind review policy of ICLR, we do not release the dataset or source code at this stage. Nevertheless, we have taken careful measures to ensure the reproducibility of our work. The main paper and Appendix provide detailed descriptions of the dataset construction process, including data collection protocols and preprocessing steps (see in the Sec. 3 and Appendix B.3). Comprehensive experimental details, including training configurations, hyperparameters, and evaluation protocols, are documented in the Experiments section (Sec. 4) of the main text and in the Appendix C. Together, these materials provide sufficient information for independent researchers to reproduce and verify our results. We plan to release the dataset and code publicly after the review process.

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

# APPENDIX

## A   RELATED WORK

**VideoQA Dataset.**   In the VideoQA field, dataset development has gradually evolved from short-video scene understanding to multimodal spatiotemporal reasoning. Early datasets (e.g., *MSVD-QA*, *MSRVTT-QA* Xu et al. (2017), *TGIF-QA* Jang et al. (2017)) were based primarily on open-domain short clips, with questions centered on high-level semantics such as "What is someone doing?", "What happened at a given moment?", or "What is likely to happen next?". Later datasets, including *ActivityNet-QA* Yu et al. (2019) and *TVQA/TVQA+* Lei et al. (2019), introduced longer video segments, richer contextual information, and spatiotemporal localization annotations, advancing research in cross-modal alignment and temporal reasoning.

Despite this progress, human-action coverage in existing datasets remains coarse: most focus on broad categories (e.g., "running," "jumping," "lifting") without fine-grained annotations of posture, deviations, or quality. Moreover, few—if any—datasets incorporate question–answering mechanisms with feedback or corrective functionality, such as "How should this be improved?" or "What adjustments should be made?". In the AQA domain, datasets Parmar & Tran Morris (2017); Xu et al. (2022); Zhang et al. (2023) typically provide only scores or phase-level labels rather than QA-style supervision. Although Wu et al. Wu et al. (2025) attempted to augment existing AQA datasets with textual annotations, the lack of expert validation limited their reliability.

Compared with conventional VideoQA benchmarks, **FLEX-VideoQA** is tasked with a markedly different orientation: it not only requires models to recognize action categories and scene semantics but also emphasizes diagnosing posture and execution deviations, while further generating targeted improvement suggestions. This shift—from "What happened?" to "How should it be improved?"—underscores **FLEX-VideoQA**'s distinctiveness and innovative value within the broader VideoQA research landscape.

**Weight-Loaded Fitness Challenges.**   Compared with self-loaded training, weight-loaded fitness presents more pronounced challenges in terms of biomechanical loading, motor control demands, and potential injury risk. Epidemiological studies indicate that weight-loaded exercise–related injuries most commonly affect the shoulder, knee, and lumbar regions, with muscle/tendon strains and joint sprains being the predominant types Noteboom et al. (2023); Kerr et al. (2010). A substantial proportion of these injuries are associated with improper control of external load. As external load increases, the joint moments and joint reaction forces at the hip, knee, and ankle rise markedly, and the work done by the joints to absorb impact also increases. This, in turn, imposes higher mechanical and neuromuscular control demands on cartilage, ligaments, and tendons Coffey et al. (2022); Kipp et al. (2011).

Weight-loaded exercises represent some of the most demanding and biomechanically complex human actions: they require high coordinated multi-joint control, engage multiple muscle groups simultaneously, and carry substantially higher injury risk than self-loaded motions. Our 20 selected actions load the most injury-prone joints (shoulders, knees, hips, spine, wrists, ankles) and capture movement patterns absent in existing AQA datasets. These load-induced biomechanical and perceptual complexities are fundamentally absent from existing datasets, which cannot capture the altered dynamics introduced by external weight.

Loaded movements also introduce distinct challenges—higher joint moments, load-dependent torque distributions, and non-visible neuromuscular activation—that fundamentally change body dynamics and error patterns. These factors make weight-loaded exercises a scientifically essential modeling domain beyond what existing datasets represent. We have made this motivation more explicit in the revised main text.

Therefore, whereas the quality of self-loaded actions can often be assessed primarily from joint angle trajectories and temporal rhythm features, the evaluation of weight-loaded actions must additionally account for multiple dimensions, including the magnitude of external load, movement or equipment trajectory, joint torque distribution, and deep muscle activity. Most existing fitness AQA datasets cover only a small number of self-loaded or low-load exercises and generally lack EMG and other physiological signals that are directly relevant to movement quality. As a result, they are insufficient to

systematically characterize the unique biomechanical challenges that arise under high-load conditions, further underscoring the necessity of developing dedicated datasets and assessment methods for weight-loaded fitness scenarios.

**Muscle activity estimation work.** Therefore, to investigate physiological signals in greater depth and explore the relationship between the video modality and physiological parameters, we design a Video2EMG model. At the same time, we note that some prior works, like Peng et al. (2024) pursue predicting active muscle from video based on actions. However, there are essential differences in our task objectives and definition, and the fine-grained nature of our work as compared to active muscle detection. For example, these work tend to learn to correlate active muscle groups with coarse-grained actions, not necessarily ground it into actual muscle usage. In comparison, our work makes use of true muscle activity for the objective measuring action quality in action quality assessment task; or true estimation of muscle activity in the form of sEMG signals from videos, resulting in learning biomechanically-oriented representations.

## B    FLEX DATASET

### B.1    SETUP

#### B.1.1    ACTION/EXERCISE SELECTION

We began by analyzing action types through a comprehensive survey of publicly available fitness AQA datasets, as summarized in a recent survey paper Yin et al. (2025). The survey revealed that the Waseda-Squat Ogata et al. (2019) and Egoexo-Fitness Li et al. (2024) comprise a total of 12 self-loaded fitness actions, whereas the Fitness-AQA Parmar et al. (2022) includes 3 weight-loaded fitness actions. Self-loaded fitness actions do not rely on equipment, making them simpler and less prone to injury—thus more suitable for limited spaces or beginners. Conversely, weight-loaded fitness actions typically involve greater difficulty, with clearly defined target muscle groups and structured progression paths (e.g., systematically increasing dumbbell or barbell weights). Given the relatively narrow range of weight-loaded fitness actions in existing datasets, the FLEX dataset prioritizes these actions. However, due to time and resource constraints, collecting every possible weight-loaded fitness action was not feasible. Therefore, based on three dimensions—action prevalence, target muscle groups, and required equipment — we ultimately selected 20 common weight-loaded actions (including 10 barbell-based and 10 dumbbell-based actions). The chosen fitness actions not only closely mirror practical training regimens but also comprehensively cover both upper and lower body muscle groups, offering a more complete dataset compared to Egoexo-Fitness Li et al. (2024) (see in Figure 5 and Table 6).

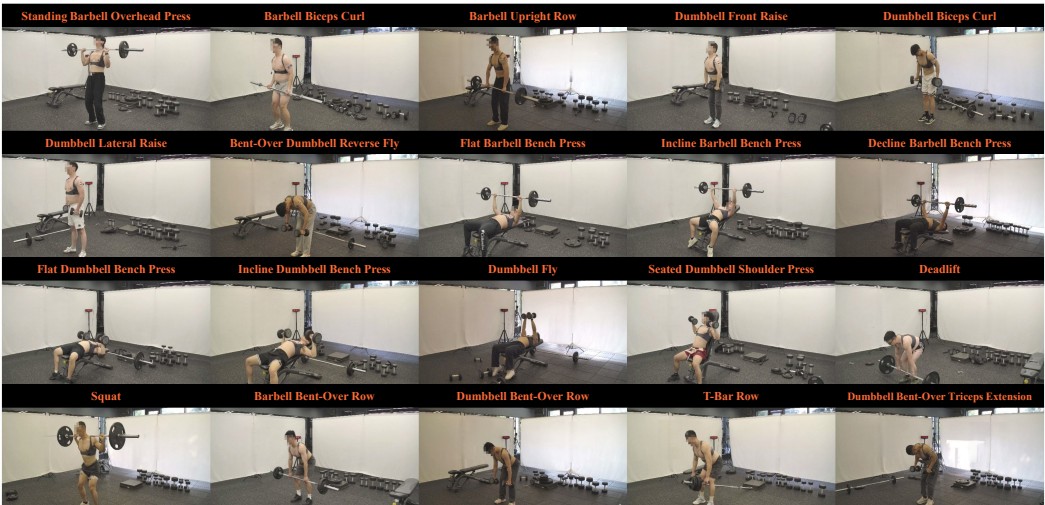

Figure 5: **The overview of the FLEX actions.**

Table 6: Comparison between the FLEX dataset with Egoexo-Fitness regarding detailed action types and target muscles. Both Egoexo-Fitness and FLEX are AQA datasets within the fitness domain. While Egoexo-Fitness primarily focuses on actions utilizing an individual's body weight, FLEX specifically emphasizes fitness actions involving external equipment-based loads. In comparison, FLEX covers actions characterized by greater complexity and targets a more comprehensive range of muscle groups.

| Dataset | Action Type | | Target Muscle | |
|---|---|---|---|---|
| **Egoexo-Fitness** Li et al. (2024) | 1. Kneeling Push-ups
3. Kneeling Torso Twist
5. Shoulder Bridge
7. Leg Reverse Lunge
9. Sumo Squat
11. High Knee | 2. Push-ups
4. Knee Raise + Abs Contract
6. Sit-ups
8. Leg Lunge with Knee Lift
10. Jumping Jacks
12. Clap Jacks | 1. Pectoralis major
3. Triceps brachii
5. Internal obliques
7. Iliopsoas
9. Hamstrings
11. Quadriceps
13. Hip abductors | 2. Anterior deltoid
4. External obliques
6. Rectus abdominis
8. Gluteus maximus
10. Erector spinae
12. Adductors |
| **FLEX** | 1. Standing Barbell Overhead Press
3. Barbell Upright Row
5. Dumbbell Bicep Curl
7. Bent-Over Dumbbell Reverse Fly
9. Incline Barbell Bench Press
11. Flat Dumbbell Bench Press
13. Dumbbell Fly
15. Deadlift
17. Barbell Bent-Over Row
19. Dumbbell Bent-Over Row | 2. Barbell Bicep Curl
4. Dumbbell Front Raise
6. Dumbbell Lateral Raise
8. Flat Barbell Bench Press
10. Decline Barbell Bench Press
12. Incline Dumbbell Bench Press
14. Seated Dumbbell Shoulder Press
16. Squat
18. T-Bar Row
20. Dumbbell Bent-Over Triceps Extension | 1. Pectoralis major
3. Middle deltoid
5. Triceps brachii
7. Brachialis
9. Trapezius
11. Gluteus maximus
13. Hamstrings
15. Latissimus dorsi | 2. Anterior deltoid
4. Posterior deltoid
6. Biceps brachii
8. Supraspinatus
10. Rhomboids
12. Quadriceps
14. Erector spinae |

### B.1.2 SUBJECT RECRUITMENT

Humans, as the core component in action performance, directly influence action quality. To collect more comprehensive data, the FLEX dataset required subjects across various capability levels compared with datasets that only contained expert-level subjects. So, we extensively recruited subjects within our institution and local commercial gyms, and ultimately selected 38 subjects, comprising 10 professional coaches, 8 amateurs, and 20 novices. We entered into a data-collection agreement with each subject, securing their authorization to record video and physiological data and to make these publicly available for academic research; the agreement also explicitly delineates the applicable financial compensation terms.

### B.2 COLLECTION

**Instruction.** As noted above, prior to data collection, the experimenter outfitted each subject with a custom vest and had surface EMG sensors affixed to the target muscle regions. Subjects then entered the capture area individually, and upon the experimenter's start signal, each subject performed the action ten times based on their own experience. Notably, the experimenter informed subjects of the specific action to be recorded before each trial, and the equipment load was set to 80% of the subject's maximum to ensure that the action was executed by their own capability.

**Human subject safety.** To ensure the safety of human subjects during data collection of risk-prone exercises, they were asked to perform exercises at 80% of their maximum weight-loaded capacity. Additionally, all subjects were covered by sports injury insurance, with on-site safety supervisors and emergency medical kits available throughout the process. Our data collection process was reviewed and approved by the Institutional Review Board (IRB). All participants provided written informed consent prior to participation.

**Rest periods in between data collection.** Considering the difficulty of performing 20 consecutive actions, particularly for the 20 novice subjects, only expert-level subjects performed all 20 consecutively. The remaining 28 subjects completed their collections separately over multiple days to ensure high-quality data, grouped by targeted muscle regions.

Given the differences in testing intensity, we implemented a differentiated compensation scheme: expert subjects received compensation equivalent to approximately 15 times the local minimum hourly wage for completing all tests within approximately two hours, while the other subjects completed their sessions across three days, about 30 minutes per day, and were compensated at a rate equivalent to roughly 7 times the local minimum hourly wage per day.

**Motion capture environment and sensor placements.** All devices were installed within a controlled space measuring 5m×5m×2m (as shown in Figure 2). The four cinema cameras for motion capture

were fixed at each corner of the square space, while the smartphone was positioned at the main viewpoint. All four sides of the controlled space were covered with curtains to reduce environmental interference further. The placement of the sEMG sensors was adjusted based on the specific target muscles involved in each action. Each subject wore the customized vest tightly to ensure accuracy in data acquisition.

### B.3 ANNOTATION

#### B.3.1 ANNOTATION RULES

Leveraging established motion-analysis standards while addressing the practical demands of large-scale training data, we formulated a principled annotation protocol governed by multiple criteria. Each captured exercise instance is decomposed into three temporal phases, defined by dominant limb kinematics: *Preparation*, *Concentric*, and *Eccentric*. Within every phase, the primary limb movement is further factorised into an ordered sequence of *action key-steps* (AKs), each paired with a concise semantic description. Concatenating these descriptions produces a textual narrative of the complete exercise.

We also catalogue fine-grained *error types* (ETs) that characterise local joint deviations and specify prescriptive feedback for every major limb action. These entities—actions, AKs, ETs, feedback messages, and their relations—are encoded in a *Fitness-Action Knowledge Graph* (see Figure 6). To ensure rapid yet consistent labelling, we introduce a penalty-based scoring scheme in which distinct ETs carry heterogeneous weights; annotators simply select the observed ETs in a clip, and the overall score is obtained by summing their penalties. Table 10 and Table 11 enumerate the complete annotation rule set and the associated ET weights, respectively.

The criteria for assigning weights to error types are as follows:

- Errors that directly compromise joint safety (e.g., spine, knee, hip, shoulder) receive the highest weight (9–10).
- Errors that undermine core stability, disrupt movement trajectory, or alter force generation (e.g., relying on momentum) receive high weight (7–9).
- Errors in foundational posture (e.g., initial stance, foot spacing) and insufficient movement amplitude receive moderate weight (6–7).
- Auxiliary errors related to fine control receive a lower weight (5).
- Breathing errors (incorrect inhalation or exhalation) receive the minimal weight (2).

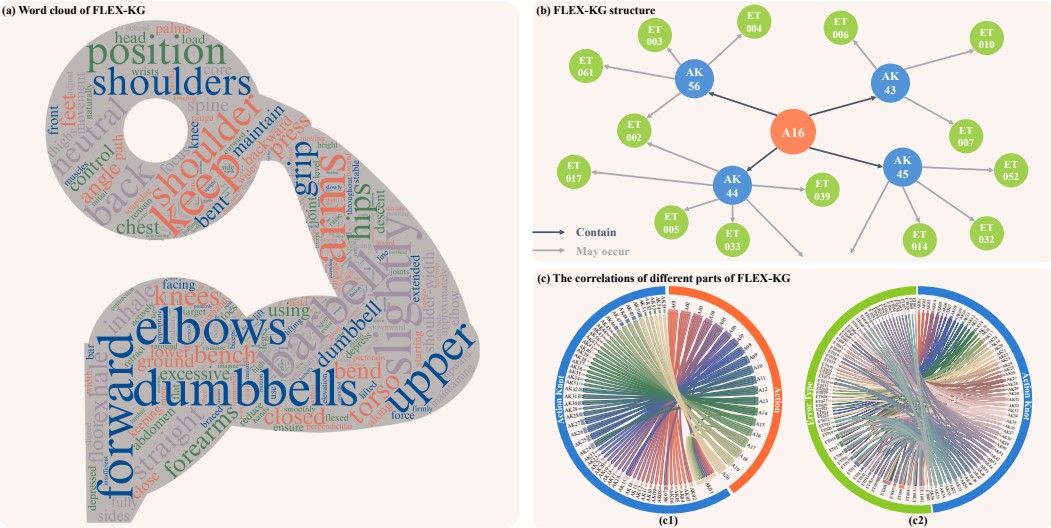

Figure 6: **The overview of the FLEX knowledge graph.** (a) Visualization of frequently used annotation words. (b) FLEX-KG: the structure of the knowledge graph. (c1) Mapping between actions and action keysteps. (c2) Mapping between action keysteps and error types.

From the above (Sec. 3.3), we can see that the dataset annotation standards are largely derived from Chinese sources; however, they are by no means limited to one country. These standards are deeply informed by internationally recognized training principles and sports science knowledge. For instance, the curricula of Beijing Sport University and the guidelines from the General Administration of Sport of China are regularly benchmarked against frameworks from the American College of Sports Medicine (ACSM), the National Strength and Conditioning Association (NSCA), German sport science and rehabilitation models, Russian strength and conditioning traditions, and Australian applied sports science practices. Such integration ensures that the standards encompass comprehensive guidance on movement mechanics, posture, and safety considerations, making them broadly representative and internationally applicable rather than narrowly country-specific.

### B.3.2 ANNOTATOR RECRUITMENT

Due to the vast volume of the FLEX dataset and stringent annotation quality requirements, we recruited 16 professional practitioners from the fitness domain, including fitness trainers and graduate students in sports science, to participate in the annotation work. To ensure consistency and scientific rigor throughout the annotation process, we engaged in in-depth collaboration with all annotators before project commencement, clarifying the annotation requirements, personnel qualifications, and the underlying logic of the guidelines. To validate the feasibility and authority of these rules, we initially conducted a small-scale pilot annotation, during which we reviewed the annotators' professional credentials and, based on the pilot results, made necessary revisions to the guidelines and refined the selection of annotators. The payment of a single annotator is higher than the local minimum hourly wage.

### B.3.3 ANNOTATION CONSISTENCY

As detailed in Sec. 3.3 and the Appendix B.3.2, we recruited 16 annotators and divided them into two groups according to experience level, employing a two-stage annotation process to ensure data quality. Specifically, in Stage 1, each sample was independently annotated for error types by three annotators from the first group, using both video and 3D pose data. This resulted in three initial annotations per sample. In Stage 2, a second group of three different annotators reviewed these annotations. For each sample, they assessed the initial labels and provided their own judgments. If disagreements were identified (i.e., inconsistent annotations or ambiguities), the second group performed a majority vote among themselves to resolve the conflict and produce a single final annotation. This process was designed to mitigate subjectivity and label noise. Ambiguous cases are discussed until an agreement is reached. For rare cases, authors invited domain experts from the Sports University to join meetings and facilitate consensus.

### B.3.4 SCORE CALCULATION

During the annotation-rule formulation phase, we assigned different penalty weights to error types according to their impact on action quality. We then summed the weights of all error types present in a given action to obtain that action's total weight. For each sample, based on the error types identified by the annotators, we calculated the ratio of the sample's cumulative error weight to the action's total weight, and subsequently computed the action score using the following formula (see in Figure 7):

$$Score = [1 - \frac{\sum_{i=1}^{N} W_i}{\sum_{j=1}^{M} W_j}] \times 100 \tag{1}$$

where $N$ represents the number of errors the sample contains, $W_i$ represents the weight of errors that the sample contains, $M$ represents the number of errors that the action type contains, and $W_j$ represents the weight of errors that the action type contains.

### B.3.5 FLEX-VIDEOQA DATASET

We designed a dialogue template following the pipeline "action recognition → action standards → action evaluation → action scoring," with all questions and reference answers automatically generated from our annotation rules and results. In particular, action-evaluation answers were pre-generated by DeepseekV3 by combining video samples, action keysteps, error types, and feedback suggestions

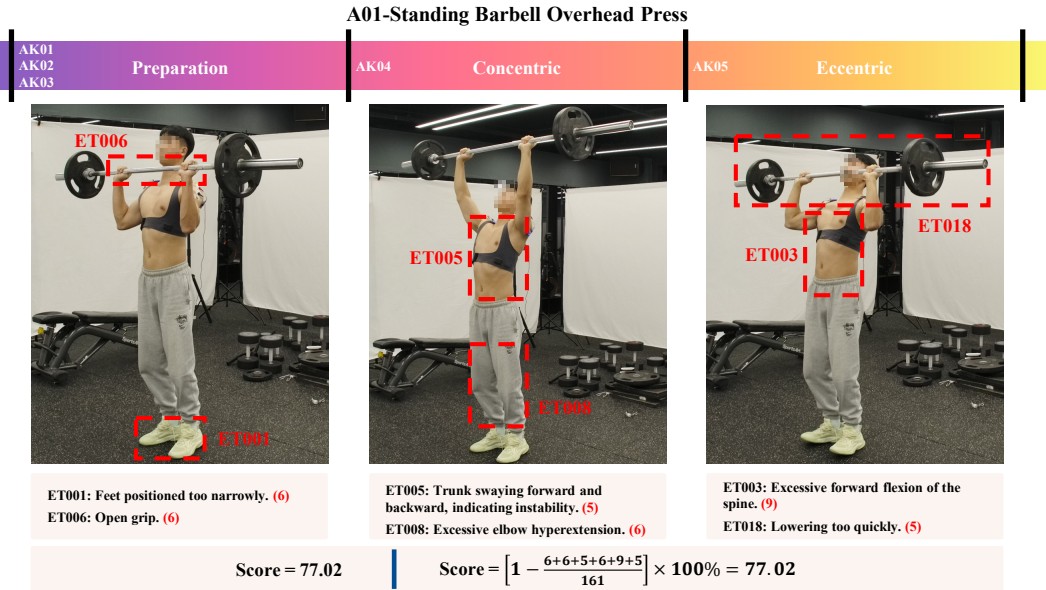

Figure 7: **Visualization of exemplary errors and scoring during one of the exercises—barbell overhead press.** Several key errors were observed that could compromise form and effectiveness. First, in the preparation process, the stance was too narrow and the grip was open rather than closed. When pressing overhead, trunk swaying and excessive elbow hyperextension were noted. The barbell was lowered too quickly, while excessive forward spinal curvature was present. Based on these errors, the final score for this action was computed to be 77.02.

to ensure consistency. Based on this pipeline, we constructed the FLEX-VideoQA dataset, which provides large-scale, fine-grained question–answer pairs tailored to action quality assessment.

Compared with existing non-AQA VideoQA tasks, FLEX-VideoQA is positioned with a clear distinction. Traditional non-AQA VideoQA datasets primarily focus on scene understanding, environmental context, or coarse-grained human actions, such as asking what a person is doing at a certain moment in the video or predicting what they might do next. Some datasets extend slightly further to include questions about more specific aspects, like the direction of a person's gaze. However, these tasks remain largely at a high-level and coarse granularity. In contrast, FLEX-VideoQA targets the fine-grained aspects of actions and exercises, with particular emphasis on identifying errors in posture and movement. More importantly, it incorporates queries that require corrective feedback—for instance, what adjustments or improvements a person should make. Such queries, which shift the focus from simply recognizing "what is happening" to providing guidance on "how it should be improved," are not addressed in existing non-AQA VideoQA datasets. This highlights the uniqueness and novelty of FLEX-VideoQA.

## B.4 DATA STATISTICS

FLEX dataset comprises 7600 experimental trials in which 38 subjects across three ability levels performed 20 distinct weight-loaded fitness actions. After data cleaning and annotation verification, 7,512 action samples were retained. We then conducted statistical analyses of per-action sample counts, sample durations, and action quality scores, shown in Figure 9. Each action's sample number is uniformly distributed between 370 and 380. The overall average duration is 233.76 frames, with per-action durations spanning from 100 to 1,000 frames. The dataset's mean quality score is 68.15, exhibiting an approximately normal distribution, with per-action quality scores ranging from 20 to 100. We also report the top 20 most frequent error types. Additionally, we provide the weight loads each subject uses for each action Figure 10.

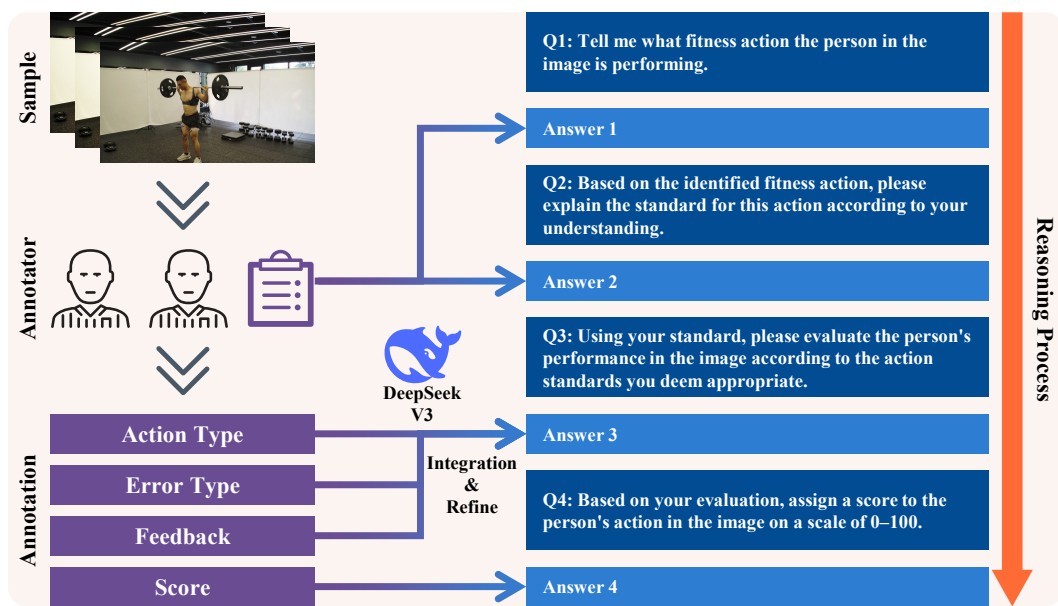

Figure 8: **The construction of FLEX-VideoQA dataset.**

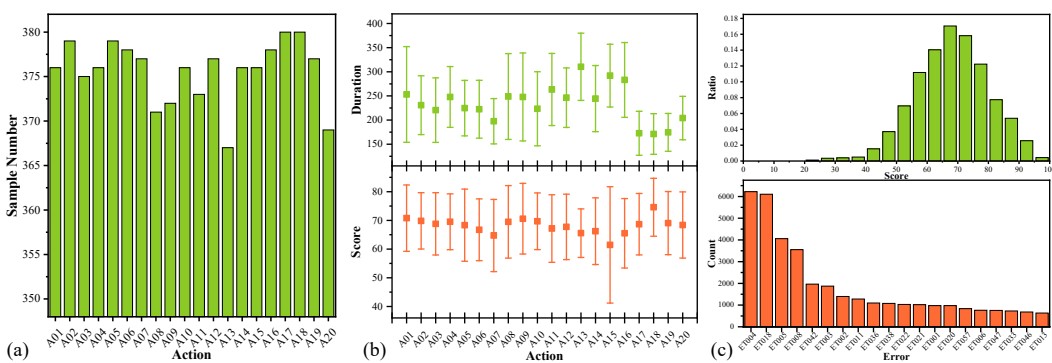

Figure 9: (a) Sample number of 20 actions. (b) Average duration and score of 20 actions. (c) Overall score distribution of the dataset and the top 20 most frequent error types.

### B.5 DATA PROTECTION

The FLEX dataset will be made available for download to all researchers. However, to protect subject privacy, all facial regions in the figures presented in this paper have been blurred, and researchers must sign a License Agreement before receiving the download link, thereby ensuring that the data are used exclusively for non-commercial academic research and that personal privacy is rigorously safeguarded.

## C EXPERIMENTS

### C.1 ACTION QUALITY ASSESSMENT

CoRe Yu et al. (2021) consists of an I3D Carreira & Zisserman (2017) pre-trained on the Kinetics dataset and a Group-Aware Regression Tree (GART). The processing pipeline is as follows: for each video pair (exemplar and target), 1×1024-dimensional features are extracted from both videos using I3D. The target's ground-truth quality score (1×1) is then concatenated with the two feature vectors to form a 1×2049-dimensional pair feature, which is passed to GART. The GART performs a coarse-to-fine procedure: it first classifies the pair feature into groups and then regresses the score

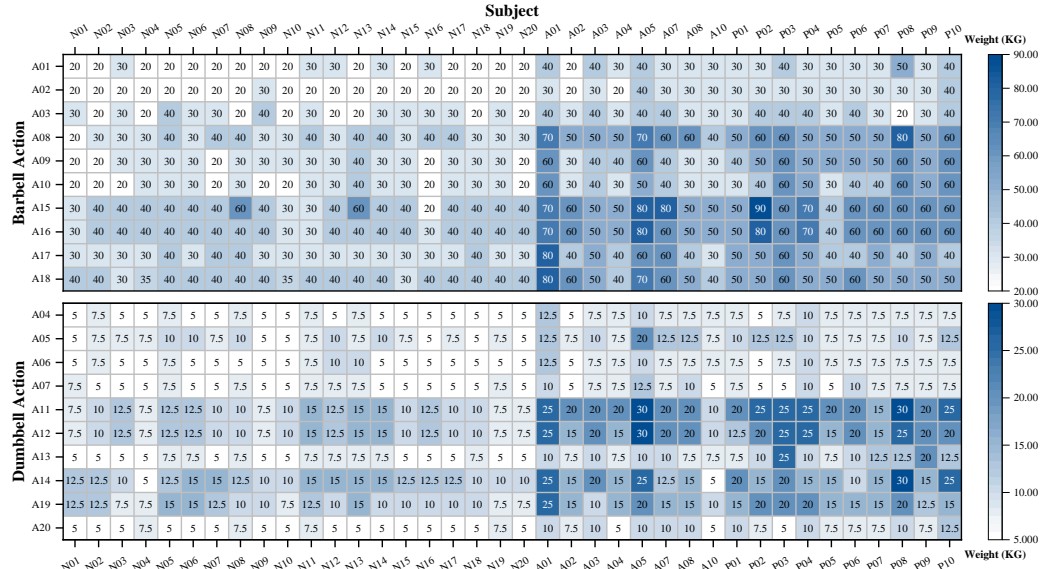

Figure 10: **Weight of subjects loaded in fitness actions.** FLEX comprises 20 weight-loaded fitness actions evenly divided between barbells and dumbbells. For barbells, the intrinsic weight of the bar (20 kg) is included in the calculation, whereas for dumbbells, only the single-sided weight is considered. In the figure, the X-axis denotes different subjects, and the Y-axis indicates the various actions. The color intensity reflects differences in weight magnitude, with the darker one corresponding to heavier weights.

difference between the exemplar and the target. The predicted quality score for the target is obtained by adding this difference to the exemplar's known score.

TPT Bai et al. (2022) follows the basic idea of CoRe Yu et al. (2021), which estimates the quality score of a target video by pairwise comparison with an exemplar and regressing their score difference. Unlike CoRe, which directly concatenates global features and feeds them into GART, TPT introduces a Temporal Parsing Transformer that uses learnable queries to decompose the entire video into a sequence of ordered part-level representations and align them between exemplar and target. The aligned part representations are then concatenated and passed through an MLP to form a pair-wise representation, which is further processed by Group-Aware Comparative Regression (GACR) to predict the score difference, yielding the final quality score of the target video.

### C.1.1 IMPLEMENTATION

To assess the contribution of multimodal inputs in action-quality assessment, we adopt CoRe Yu et al. (2021) as our primary baseline. Although its variant TPT Bai et al. (2022) reports stronger performance, it comes with substantially higher complexity and reduced flexibility for modification. Since the goal of this work is to benchmark the FLEX dataset under a transparent and reproducible setting, CoRe provides a suitable and representative choice, while remaining extensible to TPT. In the following experiments, we preserve the original GART architecture, loss function, and training strategy, modifying only the pairwise feature input to derive 8 CoRe variants.

Following CoRe's protocol, we first segment each full action video into a single-action clip based on keyframes, then sample 103 frames per clip. Because skeletal and sEMG signals are synchronized with the video, we segment them using the same keyframe indices: each skeleton sample is represented by a sequence of 103 frames, while sEMG retains its native sequence length. We set the learning rate of the pre-trained feature extractors (I3D, ST-GCN or SkateFormer) to $1 \times 10^{-4}$ and the GART learning rate to $1 \times 10^{-3}$, optimizing with Adam and no weight decay. Unlike the original CoRe, which uses 10 voters on MTL-AQA and AQA-7, we employ 3 on FLEX. Experiments were conducted on multiple NVIDIA RTX 4090 and A40 GPUs; each model was trained for 200 epochs per action. AQA experiments consumed approximately 5600 GPU hours in total.

### C.1.2 METRICS

In our AQA experiments, we use Spearman's rank correlation ($\rho$) and Relative $L2$-Distance ($R\text{-}l2$) as evaluation metrics. $\rho$, introduced by Pirsiavash et al. Pirsiavash et al. (2014), is the most widely adopted performance indicator in the AQA field; it quantifies the strength and direction of the monotonic relationship between ground-truth and predicted rankings but does not capture the absolute differences in their scores. The $\rho$ formula is as follows:

$$\rho = 1 - \frac{6 \sum_{i=1}^{n} (R_i - \widehat{R_i})^2}{n(n^2 - 1)} \tag{2}$$

where $R_i$ and $\widehat{R}_i$ represent the ground truth and predicted rankings of the $i$-th sample, respectively. $n$ is the total number of samples. The $\rho$ range is $[-1, 1]$, with values closer to 1 indicating better performance. SRC is currently the most widely used performance metric in AQA. However, SRC can only measure the strength and direction of the monotonic relationship between ground truth and predicted rankings without measuring the differences between ground truth and predicted scores.

To address this limitation, Yu et al. Yu et al. (2021)proposed $R\text{-}l2$, an $L2$-based metric that, unlike $\rho$, emphasizes the numerical discrepancy between ground-truth and predicted scores. Furthermore, by normalizing for the score ranges of different action categories, $R\text{-}l2$ facilitates cross-category training in a way that traditional $L2$ distance cannot. The $R - l2$ formula is as follows:

$$R - l2 = \frac{1}{n} \sum_{i=1}^{n} \left( \frac{|s_i - \widehat{s_i}|}{s_{max} - s_{min}} \right)^2 \tag{3}$$

where $s_i$ and $\widehat{s_i}$ represent the ground truth and predicted scores of the $i$-th sample. $s_{max}$ and $s_{min}$ represent the maximum and minimum scores of this action category. $n$ is the total number of samples. The $R - l2$ range is $[0, 1]$, closer to 0 indicating better performance.

### C.1.3 RESULTS

**3D Pose Analysis.** From Table 3, we can see that ST-GCN consistently outperforms SkateFormer under the same dataset and training settings. We attribute this to two main factors: 1) The FLEX scoring system is based on cumulative weighted penalties for fine-grained joint errors (ET), emphasizing localized constraints rather than global motion patterns. ST-GCN leverages fixed skeleton graph convolutions, encoding strong structural priors that make it naturally sensitive to joint-level errors. In contrast, SkateFormer's partitioned attention simultaneously captures global and local context, which may dilute the impact of strict local constraints central to our task. 2) SkateFormer was originally pre-trained and evaluated on the large-scale NTU RGB+D dataset with extensive augmentation. FLEX, by comparison, provides only 7,512 skeleton samples for a smaller, fine-grained task. Transformers typically excel with large datasets and heavy augmentation, whereas GCNs offer more stable performance in small-data regimes. Taken together, these factors indicate that ST-GCN remains the more suitable and effective model for evaluation on the FLEX dataset.

**Generalization Analysis.** We further evaluate the model's robustness under disjoint protocols, as shown in the bottom rows of Table 3. Compared to the standard random split ($\rho = 0.8069$), the performance drops significantly in disjoint settings, revealing the challenge of domain generalization in AQA. Specifically, the *Cross-subject* setting yields a correlation of $\rho = 0.4288$, indicating that individual variations in motion execution pose a substantial barrier to transferring learned features to unseen subjects. The viewpoint-related protocols present even greater challenges: *Cross-view* and *Mix-view* achieve $\rho$ of 0.3641 and 0.3897, respectively. This suggests that drastic perspective shifts introduce severe distribution gaps that standard visual backbones struggle to bridge without explicit alignment. Overall, these results underscore that while CoRe achieves high accuracy on seen data distributions, the proposed disjoint splits serve as a rigorous benchmark for assessing true action understanding and generalization capabilities.

**sEMG Contribution.** While the quantitative improvement from EMG is modest, the qualitative information it provides is fundamentally distinct from visual or kinematic modalities. EMG captures muscle-specific activation patterns—such as co-contraction, fatigue signatures, imbalance, or compensatory muscle usage—that cannot be inferred reliably from video or skeletal data, even when those

modalities achieve higher numerical performance. Thus, a modality's value should not be judged solely by its effect on aggregate accuracy scores; it also lies in the unique physiological insights it contributes. EMG plays a complementary rather than dominant role. Consistent with this view, our results show small but consistent performance gains from EMG, supporting the hypothesis that subtle neuromuscular cues complement rather than replace visual information.

## C.2 Action Quality Understanding

### C.2.1 Implementation

For Qwen2.5-VL-SFT, supervised fine-tuning was conducted using llama-factory Zheng et al. (2024). We used bfloat16 precision, LoRA with rank = 8, alpha = 16, and a dropout rate of 0.05. The optimizer was AdamW with an initial learning rate of $5 \times 10^{-5}$, cosine-decay scheduling, and gradient clipping at 1.0. The batch size was 2 with gradient accumulation over 4 steps, and training spanned 2 epochs for 12 hours on a single NVIDIA H20.

### C.2.2 Metrics

To strictly evaluate the quality of the generated action feedback against ground-truth expert annotations, we employ a multi-granular evaluation framework that spans from surface-level lexical fidelity (BLEU Papineni et al. (2002), ROUGE-L Lin & Och (2004)) and morphological robustness (METEOR Banerjee & Lavie (2005), CHRF++ Popović (2017)) to deep semantic consistency (BERTScore-F1 Zhang et al.).

Specifically, BLEU and ROUGE-L ensure the accurate capture of anatomical keywords via n-gram precision and recall, while METEOR and CHRF++ account for synonymy and character-level variations to prevent rigid matching. Crucially, to address the limitations of surface-level overlap, BF1 leverages pre-trained contextual embeddings to measure semantic similarity. This ensures the evaluation aligns with human judgment by validating the intrinsic meaning of the advice rather than just its phrasing. For all metrics, higher values indicate superior performance.

### C.2.3 Results

In Sec. 4.2 and Table 4, we briefly introduced the performance and overall trends of various VLMs on the FLEX-VideoQA dataset under different strategies. To provide a more detailed qualitative performance comparison and analysis of the underlying reasons, we elaborate further below.

**Part 1: Performance study**

**Part 1.1: Evaluation regimes**

We adopted three complementary evaluation regimes: (i) an off-the-shelf, in which the original pretrained checkpoints are applied directly to FLEX-VideoQA without additional conditioning; (ii) a rules-based prompting configuration, where domain-specific action guidelines and scoring rubrics are supplied as system-level prompts at inference time; and (iii) a lightweight supervised fine-tuning configuration that adds a LoRA adapter to the vision–language backbone and is trained on the same annotation schema, thereby allowing the model to internalise evaluation heuristics rather than rely on explicit prompts.

**Part 1.2: Dataset**

All experiments are carried out on our newly curated FLEX-VideoQA benchmark, whose dialogue instances are produced by a four-stage templating pipeline—action recognition, standard-of-performance retrieval, error diagnosis, and quantitative scoring—so that both questions and reference answers are automatically synthesised from structured annotations.

**Part 1.3: Case study**

To provide a more detailed presentation of model performance, we selected samples with average scores (P07-A17-01) from the same test set to showcase text generation quality, and we also included the highest and lowest scores within the test set to illustrate score prediction quality. From Table 7, we can easily get:

On sample P07-A17-01, the SFT-tuned Qwen2.5-VL 3B delivers the most diagnostically valuable narrative: it correctly identifies the barbell bent-over row, pinpoints multiple biomechanical faults—spinal curvature, shoulder elevation, eccentric tempo—and does so with succinct, discipline-specific diction that minimises redundancy while preserving clarity. The same model under prompt-only control achieves complete alignment with the reference checklist but adopts a largely affirmative stance devoid of critical feedback; this exhaustive yet un-curated recitation inflates textual volume without enriching actionable insight.

MiniCPM-o 2.6 8B and InternVL 3 2B likewise recognise the exercise and reproduce core set-up cues, yet their descriptions verge on generic rehearsal manuals: they either confine themselves to superficial form validation or omit fault identification altogether, limiting their utility for quantitative or coaching-oriented evaluation.

On sample P07-A17-01 of the FLEX-VideoQA benchmark, all evaluated vision–language models correctly categorise the scene as a barbell bent-over row, indicating robust coarse-grained action recognition. Yet when the task shifts to the dataset's distinctive fine-grained requirements—joint-level motion description and execution scoring—their capabilities diverge. The SFT-tuned Qwen2.5-VL 3B captures several biomechanically salient cues such as spinal alignment, scapular control, and eccentric tempo, providing a relatively deep diagnostic narrative, whereas its prompt-only counterpart and the MiniCPM-o 2.6 8B and InternVL 3 2B baselines largely confine themselves to generic procedural summaries that overlook quantitative joint trajectories, range of motion, and temporal rhythm. This contrast underscores FLEX-VideoQA's departure from earlier VideoQA corpora: while current VLMs have largely mastered action-type classification, modelling kinematic nuance and delivering fine-grained quality assessment remain open challenges.

Overall, Qwen2.5-VL 3B's SFT configuration stands out for blending technical precision with error-focused depth, whereas the other configurations illustrate a trade-off between checklist completeness, verbosity, and analytic substance.

Table 7: **Qualitative comparison of models in natural language feedback and error analysis generation.**

| Dimension | Qwen2.5-VL 3B (SFT) | Qwen2.5-VL 3B (Prompt) | MiniCPM-o 2.6 8B (Prompt) | InternVL 3 2B (Prompt) |
|---|---|---|---|---|
| Coarse Action Recognition | ✓ | ✓ | ✓ | ✓ |
| Fine-grained Analysis | **Better** | Good | Limited | Limited |
| Error Diagnosis | 5 faults flagged | 2 faults | None | None |
| Key-point Coverage (5 total) | 5 | 4 | 3 | 3 |
| Wording | Concise, technical | Verbose, exhaustive | Moderate | Direct, brief |

**Part 2: Technical analysis**

**Part 2.1: Reasons for different performance between models**

MiniCPM-o 2.6 8B, equipped with a larger language backbone, excels on semantic-similarity metrics (METEOR, BERTScore) after prompting; its extensive textual prior lets it quickly internalize "action–terminology" alignments under either rule-based prompting, yet its limited video pre-training maybe constraining its temporal reasoning.

Qwen 2.5-VL 3B, although smaller, benefits from the field's largest multi-source video corpus, endowing it with rich motion priors that yield consistently balanced scores on syntax-and-semantics metrics such as BLEU and CHRF++, even in an off-the-shelf setting; LoRA-SFT further consolidates and significantly boosts this multimodal (visual–linguistic) strength.

InternVL-3 2B, designed for lightness and trained on comparatively fewer videos, remains disadvantaged under all two evaluation modes: rule prompting still produces noticeable relative gains, yet absolute performance is capped by simultaneous bottlenecks in visual perception and language capacity.

**Part 2.2: Reasons for the different performance improvement between text generation and AQA-scoring**

During our experiments (see in Table 4 and Table 8), we observed that although text generation quality improved markedly, scoring accuracy did not improve accordingly. We believe this is primarily due to the inherent limitations of large pretrained language models in handling numerical data. These

models, during pretraining, rely on autoregressive prediction over discrete tokens—splitting numbers into subword units and treating them like any other vocabulary—so they lack distance constraints along the continuous number axis and do not employ specialized regression losses to reinforce gradient signals for magnitude differences. Furthermore, their training objective is to maximize likelihood rather than achieve precise counting or numerical regression, with optimization favoring overall syntactic and semantic coherence. Consequently, they struggle to represent fine-grained visual details or score distinctions, yielding approximate rather than accurately calibrated scoring outputs.

Future work could begin with a unified modeling framework that integrates discrete generation and continuous regression. By leveraging numerically aware representation learning and multi-scale supervisory signals, numerical values would be mapped to continuous vectors endowed with dimensional and ordinal constraints. An explicit regression head or score-distance regularization could then be employed to simultaneously minimize language reconstruction error and numerical deviation during autoregressive generation. At the same time, introducing cross-modal alignment through pose, temporal, or physical constraints would enforce a consistent metric structure among vision, language, and numeric modalities in the latent space, thereby enhancing the model's sensitivity to fine-grained quantitative differences and its generalization ability.

Table 8: **Score prediction.**

| Sample ID | Reference/GT Score | CoRe | Qwen2.5-VL 3B (SFT) | Qwen2.5-VL 3B (Prompt) | MiniCPM-o 2.6 8B (Prompt) | InternVL 3 2B (Prompt) |
|---|---|---|---|---|---|---|
| P08-A09-09 | 100 | 104.7985 | 65 | 65 | 95 | 85 |
| N04-A15-01 | 22.16 | 21.8366 | 65 | 65 | 95 | 75 |
| A05-A20-04 | 68.15 | 69.4700 | 55 | 65 | 75 | 75 |

## C.3 VIDEO2EMG

### C.3.1 IMPLEMENTATION

Image data were prepared according to CoRe's specifications; the key difference is that we applied a sliding-window average to the sEMG signals to be temporally aligned per frame with the video. We evaluated four Video2EMG models by combining two backbone feature extractors (ResNet-50, ViT-S/16) with two regression heads (LSTM, SVR). In the neural variants, frame-level features extracted by the backbone were processed by a single-layer LSTM (hidden size 256), and the final hidden state was passed through fully connected layers for prediction. In the kernel-based variants, features from 16 frames were concatenated into high-dimensional vectors and regressed to the EMG space using SVRs with RBF kernels. The models were trained and evaluated on 18 muscles according to the collected sEMG signals (see Table 9). Training used mean squared error (MSE) loss with the Adam optimizer (initial learning rate $1 \times 10^{-3}$), a batch size of 32, and 150 epochs, requiring about 6 hours on a single NVIDIA RTX4090. The Video2EMG experiments in total consumed approximately 72 GPU hours.

### C.3.2 RESULTS

In Sec. 4.3 and Table 5, we briefly introduced the performance of various Video2EMG. To provide a more detailed analysis of their performance, we elaborate further below.

**Backbones Comparison.** From Table 5 results, we observe that although the models adopt different backbone networks as feature extractors, their performance on the two evaluation metrics is overall quite similar. This is mainly due to the specific model configuration used in our experiments. In the Video2EMG experiments, we freeze the parameters of ViT and ResNet and use them solely as feature extraction backbones, selecting medium-sized models such as ResNet-50 and ViT-S/16. Under this setting, the representational capacity of both types of visual backbones is relatively mature and stable, and the quality of the high-level semantic features they extract is comparable. Therefore, given identical training data and supervision signals, it is expected that their performance in terms of MAE would be similar.

On this basis, we focus our training and analysis on the design and comparison of the regression heads. As shown in the results, although the absolute MAE values across different configurations are relatively close, SVR consistently outperforms LSTM when using the same visual features, with a

relative error reduction of around 10%. This indicates that, under the current experimental setup, the choice of regression head still has a substantial impact on the final performance.

**Robustness Analysis.** We further evaluated the robustness of the model under non-overlapping split protocols, and the corresponding results are reported in the bottom rows of Table 5. In addition, we introduce cross correlation peak (CCP) as an evaluation metric to measure model's EMG waveform predictions.

CCP may also be intuitively interpretable to humans. The cross-correlation peak between two EMG signals is mathematically bounded between –1 and +1. A value of +1 indicates a perfect match in waveform shape (even if one signal is slightly shifted in time), while 0 indicates no meaningful similarity, and –1 represents a perfectly inverted relationship. In practice, EMG signals typically fall between 0 and +1 because they are nonstationary and noisy, and negative correlations only occur if one waveform is an inverted version of the other. Values above about 0.7 generally indicate strong similarity, 0.3–0.7 moderate similarity, and values below 0.3 suggest poor alignment or mismatched morphology.

We observe that the methods are performing moderately well on the Vanilla Split, and poorly on the Cross-Subject/View splits. This clearly indicates that the dataset and the Video2EMG prediction task are challenging, far from being solved—there is a lot of room for improvements for future research.

Table 9: Muscles Recorded by the sEMG Device in the FLEX Dataset.

| Action | Muscle | | | |
|---|---|---|---|---|
| Standing Barbell Overhead Press | Left anterior deltoid | Right anterior deltoid | Left medial deltoid | Right medial deltoid |
| Barbell Biceps Curl | Left biceps brachii | Right biceps brachii | Left triceps brachii | Right triceps brachii |
| Barbell Upright Row | Left medial deltoid | Right medial deltoid | Left anterior deltoid | Right anterior deltoid |
| Dumbbell Front Raise | Left anterior deltoid | Right anterior deltoid | Left medial deltoid | Right medial deltoid |
| Dumbbell Biceps Curl | Left biceps brachii | Right biceps brachii | Left triceps brachii | Right triceps brachii |
| Dumbbell Lateral Raise | Left medial deltoid | Right medial deltoid | Left anterior deltoid | Right anterior deltoid |
| Bent-Over Dumbbell Reverse Fly | Left posterior deltoid | Right posterior deltoid | | |
| Flat Barbell Bench Press | Left pectoralis major | Right pectoralis major | Left pectoralis major | Right pectoralis major |
| Incline Barbell Bench Press | Left pectoralis major | Right pectoralis major | Left pectoralis major | Right pectoralis major |
| Decline Barbell Bench Press | Left pectoralis major | Right pectoralis major | Left pectoralis major | Right pectoralis major |
| Flat Dumbbell Bench Press | Left pectoralis major | Right pectoralis major | Left pectoralis major | Right pectoralis major |
| Incline Dumbbell Bench Press | Left pectoralis major | Right pectoralis major | Left pectoralis major | Right pectoralis major |
| Dumbbell Fly | Left pectoralis major | Right pectoralis major | Left pectoralis major | Right pectoralis major |
| Seated Dumbbell Shoulder Press | Left anterior deltoid | Right anterior deltoid | Left medial deltoid | Right medial deltoid |
| Deadlift | Left biceps femoris | Right biceps femoris | Left vastus lateralis | Right vastus lateralis |
| Squat | Left vastus lateralis | Right vastus lateralis | Left biceps femoris | Right biceps femoris |
| Barbell Bent-Over Row | Left latissimus dorsi | Right latissimus dorsi | | |
| T-Bar Row | Left latissimus dorsi | Right latissimus dorsi | | |
| Dumbbell Bent-Over Row | Left latissimus dorsi | Right latissimus dorsi | | |
| Dumbbell Bent-Over Triceps Extension | Left triceps brachii | Right triceps brachii | Left biceps brachii | Right biceps brachii |

## D    FUTURE DIRECTION

FLEX is a dataset-centric work that is semantically grounded in gym-based fitness training. Under clearly defined technical standards and target muscle groups, we collect high-quality data and provide fine-grained annotations, thereby offering a high-difficulty, high-fidelity benchmark for multimodal action understanding methods. However, this semantic focus does not imply that the application scope or underlying scientific questions are narrow. The key challenges embodied in FLEX—such as temporal reasoning over human motion, cross-modal alignment, multimodal information fusion, and robust modeling of variations in execution quality—are not unique to fitness, but are also central to broader domains including rehabilitation training, elite sports coaching, and human–robot interaction.

Furthermore, the multimodal and fine-grained annotations in FLEX not only provide a high-difficulty, high-fidelity evaluation platform for multimodal action understanding methods, but also help support research directions such as error detection, self-supervised learning, vision–language modeling, and biomechanical analysis. To facilitate future research and inspire further exploration, we therefore systematically outline below the potential research topics and application paths that can be pursued based on the FLEX dataset.

## D.1 Physiology-Aware Research

During the construction of the FLEX dataset, in addition to recording videos and surface EMG signals, we also used a customized wearable vest to synchronously capture participants' heart rate and respiratory rate while they performed the exercises. As important physiological feedback during physical activity, heart rate and breathing provide complementary information about movement quality and training load. We believe that jointly modeling these physiological signals with action videos holds substantial research potential, particularly for biomechanics and fatigue analysis, and we elaborate on this in more detail below.

**Heart Rate.** Heart-rate data adds a powerful physiological dimension to action quality assessment because it reveals how the body responds internally to a movement, not just how the movement appears externally. Like in our dataset, when synchronized with fine-grained motion and muscle-activation data, heart rate provides insight into movement efficiency: for example, if two people perform an exercise with similar form, the one whose heart rate rises more sharply is likely compensating, bracing inefficiently, or expending unnecessarily high effort. This makes heart-rate curves a way to detect subtle technique problems that video and pose estimation alone may not capture. Heart rate also reflects fatigue progression within a set. Long before form visibly deteriorates, the cardiovascular system responds with accelerated rise patterns, so monitoring intra-set heart-rate drift allows a model to predict imminent form breakdown and adjust load, tempo, or stop the set entirely for safety. Even in the absence of formal HRV measurements, shorter-term heart-rate fluctuations can signal readiness and recovery, enabling the system to anticipate when a user is physiologically "off" before the workout begins and then adjust expectations for movement quality accordingly.

Heart rate becomes even more valuable when fused with other modalities like sEMG. It can help interpret whether muscle activation patterns are correct or compensatory, distinguishing between movement that is technically flawed and movement that is simply difficult. For instance, if the heart rate rises sharply during a curl while shoulder muscles show excessive activation, the system can infer that the user is swinging the weight. This combination of physiological and biomechanical information makes it possible to move beyond simple pose-based scoring toward a richer model of movement quality under load. Heart rate also enables scoring of the balance between effort and form: when a user is working hard but keeping technique stable, the system can label a rep as high quality, whereas minimal effort paired with large deviations indicates poor technique. Because injury risk typically emerges when high physiological strain overlaps with joint-angle deviations, heart rate adds an early-warning mechanism that highlights dangerous reps before they become harmful.

From a modeling perspective, heart-rate sequences can be integrated into a multi-task learning framework where the model simultaneously predicts rep boundaries, form scores, fatigue state, and the consistency between physiological effort and biomechanical movement patterns. Since heart-rate signals are smooth and low-frequency, they stabilize predictions and reduce noise in pose-only systems. With such capabilities, we aim to build applications that go far beyond typical video-only fitness feedback: systems that personalize load recommendations based on physiological response, detect hidden fatigue before form collapses, adjust difficulty automatically, or diagnose whether performance issues stem from poor technique, insufficient strength, or inadequate endurance. Heart rate gives access to the "why" behind movement quality—something that purely visual systems cannot infer—and turns action quality assessment into a truly holistic understanding of how well the user is performing.

**Breath Rate.** In weight-loaded fitness scenarios, breath signals constitute a critical physiological channel for action quality assessment, yet one that is often difficult to reliably infer from video alone. Proper breathing rhythm is directly linked to intra-abdominal pressure, trunk stability, and force production efficiency: in exercises such as squats, deadlifts, and bench presses, the timing of inhalation, breath-holding, and exhalation is tightly coupled with spinal protection and torque transmission. By synchronously recording and aligning breathing frequency with video, 3D skeletons, and sEMG signals, we can more accurately determine whether a participant is adopting an appropriate breathing strategy and identify high-risk cases where the movement appears acceptable on the surface but the breathing pattern is completely misaligned—risks that are often hard to detect from appearance alone.

From the perspectives of annotation and modeling, breath signals provide an important basis for fine-grained error types and phase-wise assessment (for example, ET014 and ET017). Many core-stability-related errors—such as exhaling too early at the bottom of a squat, frequent breathing changes during high-load eccentric phases, or loss of breath-holding during critical force-production windows—may manifest in video only as slight wobbling or a mildly strained appearance, but are very salient in the breathing rhythm. Once breathing patterns are aligned with action phases, annotators can label "breathing errors" with greater confidence, rather than relying solely on subjective impressions. In future model training, breath sequences can also serve as a low-frequency, structurally clear temporal signal that helps the model learn the alignment between "ideal breathing" and action phases: when the model detects that external form has not yet visibly collapsed but the breathing rhythm is already severely disrupted, it can issue early risk alerts or recommend reducing the load, thereby providing earlier safety warnings than systems based only on video or skeleton sequences.

Moreover, when combined with heart rate and sEMG, breath signals help distinguish between "technical issues" and "physiological capacity limitations." For example, under the same load, if a participant's movement form appears acceptable but breathing is clearly labored and uncoordinated, and heart rate and muscle activation patterns indicate excessive strain, this more likely suggests that the training intensity exceeds their cardiopulmonary or core capacity. Conversely, if breathing remains steady and breath-holding is appropriate during force production, while most errors concentrate on joint trajectories or muscle recruitment patterns, the primary issue is more likely technical. Such multimodal information supports more targeted feedback—e.g., recommending optimization of breathing strategy, reducing load, shortening set duration, or adjusting tempo—so that action quality assessment goes beyond a binary judgment of "form correctness" and evolves into an integrated interpretation of technique, fatigue, and physiological load.

**sEMG.** EMG's qualitative value also makes it a key modality for precision healthcare applications—such as rehabilitation, motor-control analysis, and ergonomics—where understanding which muscles are activated, and whether they are activated correctly, is more critical than aggregate performance metrics. FLEX supports such research by providing synchronized muscle-level signals aligned with high-fidelity motion data.

By offering a multimodal dataset that includes both dominant and subtle modalities, FLEX also enables future exploration of advanced fusion strategies, robustness under occlusion, and muscle-aware AQA—directions that are difficult to study with existing datasets and that open up meaningful opportunities for the community.

## D.2 Cross Domain Research

Thanks to its design along multiple dimensions—multimodality, multi-view recording, diverse action types, multiple participants, varying skill levels, and repeated trials—FLEX can support a much broader range of research paradigms than those showcased in the main experiments. It is by no means limited to the specific setups we present. For example, researchers can systematically study and evaluate cross-subject, cross-view, cross-action, cross–ability level, and cross-repetition generalization within a single unified dataset.

To further facilitate cross-scenario and cross-domain research on FLEX, we build on the split strategies commonly used in existing AQA datasets and propose a new family of evaluation protocols as references and examples for future work. Unlike the random split adopted in the main paper—following standard AQA practice (e.g., AQA-7, FineDiving), where for each action, samples from different subjects and repetitions are mixed and then randomly divided into training and test sets—we additionally define stricter and more realistic protocols, including:

- *Cross-subject*: train and test on disjoint subject sets to assess cross-person generalization.
- *Cross-view*: train on 1 view and test on a different unseen view to simulate viewpoint shifts.
- *Mix-view*: train on 3 views and test on the remaining unseen view.

We have added baseline results for these splits in Table 3. Compared with the random split, performance drops noticeably under cross-subject and cross-view settings, showing that FLEX remains substantially challenging when evaluated under stricter protocols. This also offers a natural difficulty ladder: methods can first be validated under the original split, then further tested for generalization and robustness under the harder splits.

### D.3 PRACTICAL LIGHTWEIGHT DEPLOYMENT

The potential mobile and edge deployment was carefully considered from the outset: we deliberately included a viewpoint captured from a mobile device (see Fig. 1 main paper), so that the community can easily explore and validate lightweight models on top of FLEX. Furthermore, from a practical deployment perspective, there are multiple well-established paths, such as reducing the number of input modalities, lowering input resolution, applying knowledge distillation, and performing online detection-and-evaluation schemes. All of these strategies can substantially reduce computational overhead without changing the FLEX annotation protocol or evaluation settings. Since FLEX is a dataset paper, in this study, we have focussed to validate the effectiveness and complementarity of its multimodal data rather than to aggressively optimize model efficiency. Accordingly, we adopt full multimodal configurations and relatively large-scale, long training and evaluation to probe the upper-bound performance of FLEX in research settings, which does not directly reflect the resource requirements of real-world deployment.

### D.4 DOMAIN GENERALIZATION

To further illustrate the generalizability of our pipeline, we provide a concrete example in a domain that is structurally different from fitness: surgical skill assessment, a key area in medical training. Surgical training involves evaluating fine-grained motor actions, adherence to procedural steps, ergonomic efficiency, and error patterns—challenges that closely parallel those addressed by our framework. Applying our pipeline to this domain requires only substituting the domain-specific knowledge sources, while all technical components remain unchanged:

- *Domain Knowledge Source:* replace the fitness-oriented knowledge graph with well-established surgical standards such as the **Objective Structured Assessment of Technical Skills (OSATS)**, **Global Operative Assessment of Laparoscopic Skills (GOALS)**, or other procedure-specific surgical competency rubrics.
- *Action Ontology & Error Taxonomy:* construct a surgical action graph describing key procedural steps (e.g., incision, tissue handling, knot tying) and their associated error types (e.g., excessive force, imprecise instrument motion, improper angle). These map directly onto our method for building action hierarchies and error annotation rules.
- *Compositional Scoring:* surgical scoring systems decompose performance into factors such as economy of motion, precision, and smoothness—an ideal fit for our hierarchical and compositional scoring design. The same rule-based scoring logic applies seamlessly.
- *Multimodal Alignment:* surgical settings often use video, tool-motion trajectories (from robotic systems or instrument trackers), motion sensors, and even EMG for tremor detection. These modalities align directly with the multimodal inputs modeled in FLEX and can be fused using the same alignment and representation learning framework.
- *Modeling Pipeline:* our modules do not assume any fitness-specific semantics, so they can be transferred to surgical tasks without modification.

We hope this example demonstrates concretely that the pipeline's generalization: the knowledge-driven, modular, and multimodal design naturally extends to domains such as surgery, where fine-grained evaluation is critical and multimodal signals are readily available.

## E   FITNESS KNOWLEDGE GRAPH

In this section, we present the role of the fitness knowledge graph in annotation, encompassing actions, action keysteps, error types, and their associated weights (see in Table 10 and Table 11). Together, these elements provide annotators with a unified reference standard that enhances data consistency and reliability. Beyond annotation, the knowledge graph also underpins intelligent movement analysis, personalized coaching, and rehabilitation monitoring, while establishing a generalizable framework for both academic research and industrial applications.

Table 10: Full list of the FLEX annotation rules.

| Standing Barbell Overhead Press | Action Keystep | Error Type |
|---|---|---|
| **Preparation** | AK01: Stand with feet shoulder-width apart and toes slightly turned outward. Keep the chest lifted and the core engaged. Maintain a neutral spine position with the scapulae depressed and retracted, ensuring trunk stability. | ET001: Feet positioned too narrowly or too widely. |
| | | ET002: Excessive arching or collapsing of the lower back. |
| | | ET003: Excessive forward flexion or excessive backward curvature of the spine. |
| | | ET004: Shoulder elevation, lack of scapular retraction, or failure to depress the shoulders. |
| | AK02: Grip the barbell with a pronated (overhand) closed grip, with hands slightly wider than shoulder-width apart. Keep the elbows slightly flexed and maintain a neutral wrist position. | ET006: Open grip. |
| | | ET007: Grip width is too narrow or too wide. |
| | | ET008: Excessive elbow flexion or hyperextension. |
| | | ET009: Wrist excessively extended or tilted to the side. |
| | AK03: Lift the barbell to rest above the clavicles. Forearms remain perpendicular to the floor, palms facing up, elbows lower than the shoulders for stable support. | ET010: The initial barbell position is set too high or too low. |
| | | ET011: Forearms are not perpendicular to the ground. |
| | | ET012: Palms facing inward or downward. |
| | | ET013: Excessive elbow adduction or abduction. |
| **Concentric** | AK04: Exhale. Primarily activate the anterior deltoid, assisted by the triceps, with elbows pointing forward. Press the barbell overhead until the arms are straight. Maintain a neutral spine and braced core throughout. | ET014: Inhalation. |
| | | ET003: Excessive forward flexion or excessive backward curvature of the spine. |
| | | ET005: Trunk swaying forward and backward, indicating instability. |
| | | ET008: Excessive elbow flexion or hyperextension. |
| | | ET013: Excessive elbow adduction or abduction. |
| | | ET015: Using a knee drive or leaning the torso backward to assist the movement. |
| | | ET016: Excessive forward head movement, obstructing the barbell's vertical path. |
| **Eccentric** | AK05: Inhale. Control the barbell along the same path, lowering it smoothly to the clavicles. Keep the scapulae stable, the deltoids resisting gravity, and forearms vertical. Engage the core and maintain a neutral lumbar spine. | ET017: Exhalation. |
| | | ET003: Excessive forward flexion or excessive backward curvature of the spine. |
| | | ET011: Forearms are not perpendicular to the ground. |
| | | ET013: Excessive elbow adduction or abduction. |
| | | ET018: Lowering too quickly. |
| | | ET019: Ending position of the barbell is too high or too low. |

| Barbell Biceps Curl | Action Keystep | Error Type |
|---|---|---|
| **Preparation** | AK01: Stand with feet shoulder-width apart and toes slightly turned outward. Keep the chest lifted and the core engaged. Maintain a neutral spine position with the scapulae depressed and retracted, ensuring trunk stability. | ET001: Feet positioned too narrowly or too widely. |
| | | ET002: Excessive arching or collapsing of the lower back. |
| | | ET003: Excessive forward flexion or excessive backward curvature of the spine. |
| | | ET004: Shoulder elevation, lack of scapular retraction, or failure to depress the shoulders. |
| | AK06: Grip the barbell with an underhand, closed grip, slightly wider than shoulder-width. Keep the wrists in a neutral position. | ET006: Open grip. |
| | | ET007: Grip width is too narrow or too wide. |
| | | ET009: Wrist excessively extended or tilted to the side. |
| | AK07: Arms hang naturally with elbows slightly bent. Elbows remain tucked in, upper arms fixed. The barbell starts at the front of the thighs. | ET008: Excessive elbow flexion or hyperextension. |
| | | ET020: Elbows move forward or backward, upper arms are not stabilized. |
| | | ET021: Excessive elbow abduction, moving away from the torso. |
| **Concentric** | AK08: Exhale. Actively contract the biceps to curl the forearms along a semicircular path up to the front of the shoulders. Apply symmetrical force with both hands to maintain balance. Keep the upper arms fixed and spine neutral throughout. | ET014: Inhalation. |
| | | ET003: Excessive forward flexion or excessive backward curvature of the spine. |
| | | ET005: Trunk swaying forward and backward, indicating instability. |
| | | ET009: Wrist excessively extended or tilted to the side. |
| | | ET015: Using a knee drive or leaning the torso backward to assist the movement. |
| | | ET019: Ending position of the barbell is too high or too low. |
| | | ET022: Using forward or upward shoulder movement to assist the action. |
| | | ET023: Elbows move forward or backward, causing the barbell path to deviate from an arc. |
| **Eccentric** | AK09: Inhale. The biceps control the barbell on the slow descent back to the starting position at the front of the thighs. Keep the elbows slightly bent (not locked) and the upper arms tight against the torso. | ET017: Exhalation. |
| | | ET008: Excessive elbow flexion or hyperextension. |
| | | ET018: Lowering too quickly. |
| | | ET020: Elbows move forward or backward, upper arms are not stabilized. |

| Barbell Upright Row | Action Keystep | Error Type |
|---|---|---|
| **Preparation** | AK01: Stand with feet shoulder-width apart and toes slightly turned outward. Keep the chest lifted and the core engaged. Maintain a neutral spine position with the scapulae depressed and retracted, ensuring trunk stability. | ET001: Feet positioned too narrowly or too widely. |
| | | ET002: Excessive arching or collapsing of the lower back. |
| | | ET003: Excessive forward flexion or excessive backward curvature of the spine. |
| | | ET004: Shoulder elevation, lack of scapular retraction, or failure to depress the shoulders. |
| | AK02: Grip the barbell with a pronated (overhand) closed grip, with hands slightly wider than shoulder-width apart. Keep the elbows slightly flexed and maintain a neutral wrist position. | ET006: Open grip. |
| | | ET007: Grip width is too narrow or too wide. |
| | | ET008: Excessive elbow flexion or hyperextension. |
| | | ET009: Wrist excessively extended or tilted to the side. |
| | AK10: Position the barbell in front of the thighs with a firm grip, ensuring the web of the hands is securely locked around the bar. | ET010: The initial barbell position is set too high or too low. |
| **Concentric** | AK11: Exhale. Maintain a stable torso. Focus on contracting the middle deltoid. Abduct the elbows out to the sides and lift the barbell vertically along the sternum until it reaches chest height. | ET014: Inhalation. |
| | | ET005: Trunk swaying forward and backward, indicating instability. |
| | | ET015: Using a knee drive or leaning the torso backward to assist the movement. |
| | | ET019: Ending position of the barbell is too high or too low. |
| | | ET024: Elbows rising much lower than the shoulder joints. |
| | | ET025: The barbell does not travel vertically relative to the ground. |
| **Eccentric** | AK12: Inhale. Use the deltoids' eccentric strength to control the barbell's descent along the same path back to the start. Keep the lumbar spine neutral and core braced, with the elbows slightly bent at the bottom. | ET017: Exhalation. |
| | | ET003: Excessive forward flexion or excessive backward curvature of the spine. |
| | | ET008: Excessive elbow flexion or hyperextension. |
| | | ET018: Lowering too quickly. |

| Dumbbell Front Raise | Action Keystep | Error Type |
|---|---|---|
| **Preparation** | AK01: Stand with feet shoulder-width apart and toes slightly turned outward. Keep the chest lifted and the core engaged. Maintain a neutral spine position with the scapulae depressed and retracted, ensuring trunk stability. | ET001: Feet positioned too narrowly or too widely. |
| | | ET002: Excessive arching or collapsing of the lower back. |
| | | ET003: Excessive forward flexion or excessive backward curvature of the spine. |
| | | ET004: Shoulder elevation, lack of scapular retraction, or failure to depress the shoulders. |
| | AK13: Arms fully extended. Hold the dumbbells in an overhand, closed grip at the front of the thighs, palms facing the legs, elbows slightly bent. | ET006: Open grip. |
| | | ET008: Excessive elbow flexion or hyperextension. |
| | | ET026: Palms facing forward or backward. |
| **Concentric** | AK14: Exhale. Focus on the anterior deltoid as the primary mover. Using the shoulder joint as the pivot, smoothly raise the dumbbells forward until the elbows are level with the shoulders. Keep the core stable, shoulders depressed, and elbows slightly bent. | ET014: Inhalation. |
| | | ET005: Trunk swaying forward and backward, indicating instability. |
| | | ET008: Excessive elbow flexion or hyperextension. |
| | | ET015: Using a knee drive or leaning the torso backward to assist the movement. |
| | | ET022: Using forward or upward shoulder movement to assist the action. |
| | | ET027: Changing the elbow angle, causing the dumbbell path to deviate from an arc. |
| | | ET028: Ending position of the dumbbell is too high or too low. |

| | Action Keystep | Error Type |
|---|---|---|
| Eccentric | AK15: Inhale. Maintain tension in the deltoids to resist gravity. Control the dumbbells as you lower them along the same path back to the starting position. Keep the core stable and elbows slightly bent. | ET017: Exhalation.
ET008: Excessive elbow flexion or hyperextension.
ET018: Lowering too quickly. |

| Dumbbell Biceps Curl | Action Keystep | Error Type |
|---|---|---|
| Preparation | AK01: Stand with feet shoulder-width apart and toes slightly turned outward. Keep the chest lifted and the core engaged. Maintain a neutral spine position with the scapulae depressed and retracted, ensuring trunk stability. | ET001: Feet positioned too narrowly or too widely.
ET002: Excessive arching or collapsing of the lower back.
ET003: Excessive forward flexion or excessive backward curvature of the spine.
ET004: Shoulder elevation, lack of scapular retraction, or failure to depress the shoulders. |
| | AK16: Grip the dumbbells with a neutral, closed grip. | ET006: Open grip. |
| | AK07: Arms hang naturally with elbows slightly bent. Elbows remain tucked in, upper arms fixed. The barbell starts at the front of the thighs. | ET008: Excessive elbow flexion or hyperextension.
ET020: Elbows move forward or backward, upper arms are not stabilized.
ET021: Excessive elbow abduction, moving away from the torso. |
| Concentric | AK17: Exhale. Actively contract the biceps, curling the forearms in a semicircle while externally rotating the forearms, until the dumbbells reach shoulder level. Keep the upper arms fixed and the spine neutral. | ET014: Inhalation.
ET003: Excessive forward flexion or excessive backward curvature of the spine.
ET005: Trunk swaying forward and backward, indicating instability.
ET015: Using a knee drive or leaning the torso backward to assist the movement.
ET022: Using forward or upward shoulder movement to assist the action.
ET028: Ending position of the dumbbell is too high or too low.
ET029: Elbows move forward or backward, causing the dumbbell path to deviate from an arc.
ET030: Excessive wrist supination over-rotating the forearms. |
| Eccentric | AK18: Inhale. The biceps control the dumbbells on the slow descent back to the starting position in front of the thighs. Keep elbows slightly bent and upper arms close to the torso. | ET017: Exhalation.
ET008: Excessive elbow flexion or hyperextension.
ET018: Lowering too quickly.
ET020: Elbows move forward or backward, upper arms are not stabilized. |

| Dumbbell Lateral Raise | Action Keystep | Error Type |
|---|---|---|
| Preparation | AK01: Stand with feet shoulder-width apart and toes slightly turned outward. Keep the chest lifted and the core engaged. Maintain a neutral spine position with the scapulae depressed and retracted, ensuring trunk stability. | ET001: Feet positioned too narrowly or too widely.
ET002: Excessive arching or collapsing of the lower back.
ET003: Excessive forward flexion or excessive backward curvature of the spine.
ET004: Shoulder elevation, lack of scapular retraction, or failure to depress the shoulders. |
| | AK19: Hold the dumbbells in a neutral, closed grip, positioning them in front of the body with arms fully extended. | ET006: Open grip.
ET008: Excessive elbow flexion or hyperextension.
ET031: Initial position of the dumbbell is too far forward or backward. |
| Concentric | AK20: Exhale. Focus on the anterior and middle deltoid. Using the shoulder joint as the pivot, lift the dumbbells laterally to shoulder height. Keep elbows slightly bent, core stable, and shoulders depressed throughout. | ET014: Inhalation.
ET005: Trunk swaying forward and backward, indicating instability.
ET008: Excessive elbow flexion or hyperextension.
ET015: Using a knee drive or leaning the torso backward to assist the movement.
ET022: Using forward or upward shoulder movement to assist the action.
ET027: Changing the elbow angle, causing the dumbbell path to deviate from an arc.
ET028: Ending position of the dumbbell is too high or too low. |
| Eccentric | AK15: Inhale. Maintain tension in the deltoids to resist gravity. Control the dumbbells as you lower them along the same path back to the starting position. Keep the core stable and elbows slightly bent. | ET017: Exhalation.
ET008: Excessive elbow flexion or hyperextension.
ET018: Lowering too quickly. |

| Bent-Over Dumbbell Reverse Fly | Action Keystep | Error Type |
|---|---|---|
| Preparation | AK21: Stand with feet shoulder-width apart, toes slightly turned outward, and knees slightly flexed. Keep the back straight and bend forward to approximately a 40° angle with the ground. | ET001: Feet positioned too narrowly or too widely.
ET002: Excessive arching or collapsing of the lower back.
ET032: Excessive knee flexion or hyperextension.
ET033: Torso angle too shallow or too steep. |
| | AK22: Let the arms hang naturally, holding the dumbbells with a neutral, closed grip just below the knee level. Keep the elbows slightly flexed. | ET006: Open grip.
ET008: Excessive elbow flexion or hyperextension. |
| Concentric | AK23: Exhale. Focus on the posterior deltoid. Using the shoulder joint as the pivot, lift the dumbbells laterally until they reach shoulder height. Keep elbows slightly bent, core stable, and shoulders depressed. | ET014: Inhalation.
ET008: Excessive elbow flexion or hyperextension.
ET027: Changing the elbow angle, causing the dumbbell path to deviate from an arc.
ET034: During abduction, the scapula elevates and the neck protrudes forward.
ET035: Arms extending diagonally backward, lifting beyond the shoulder joint's horizontal level. |
| Eccentric | AK15: Inhale. Maintain tension in the deltoids to resist gravity. Control the dumbbells as you lower them along the same path back to the starting position. Keep the core stable and elbows slightly bent. | ET017: Exhalation.
ET008: Excessive elbow flexion or hyperextension.
ET018: Lowering too quickly. |

| Flat Barbell Bench Press | Action Keystep | Error Type |
|---|---|---|
| Preparation | AK24: Lie supine on a flat bench, feet flat on the floor. Keep the waist and abdomen naturally braced, shoulders depressed, and ensure the head, upper back, and hips remain on the bench. Lift the chest and engage the abdomen. | ET036: Feet not firmly planted on the ground.
ET002: Excessive arching or collapsing of the lower back.
ET004: Shoulder elevation, lack of scapular retraction, or failure to depress the shoulders.
ET037: Head, upper back, or glutes lifting off the bench. |
| | AK25: Fully extend the arms, gripping the barbell with an overhand, closed grip, slightly wider than shoulder-width. | ET006: Open grip.
ET007: Grip width is too narrow or too wide.
ET009: Wrist excessively extended or tilted to the side. |
| Eccentric | AK26: Inhale. Maintain torso stability, allow the shoulders to naturally depress, and bend the elbows downward so the barbell lowers slowly and smoothly to the chest or just above it. | ET017: Exhalation.
ET018: Lowering too quickly.
ET019: Ending position of the barbell is too high or too low.
ET038: The angle between the upper arm and torso is too large or too small.
ET039: Insufficient barbell descent, resulting a shortened range of motion. |
| Concentric | AK27: Exhale. Focus on pectoralis major contraction. Using the shoulder joint as the pivot, apply symmetrical force with both hands to press the barbell upward. Keep the elbows slightly bent at the top. | ET014: Inhalation.
ET008: Excessive elbow flexion or hyperextension.
ET040: Hips rise or legs drive off the floor to assist during the press.
ET041: Raising the head or pushing the neck forward to assist during the press.
ET042: Shrugging the shoulders or shifting forward during the press.
ET043: The barbell sways left or right, or one arm dominates the force output.
ET044: Insufficient barbell elevation, resulting a shortened range of motion. |

| Incline Barbell Bench Press | Action Keystep | Error Type |
|---|---|---|
| Preparation | AK28: Lie supine on an incline bench at approximately 30° to the floor, feet flat. Keep the waist and abdomen braced, shoulders depressed, and the head, upper back, and hips on the bench. Chest lifted and abdomen engaged. | ET002: Excessive arching or collapsing of the lower back. |
| | | ET004: Shoulder elevation, lack of scapular retraction, or failure to depress the shoulders. |
| | | ET036: Feet not firmly planted on the ground. |
| | | ET037: Head, upper back, or glutes lifting off the bench. |
| | AK25: Fully extend the arms, gripping the barbell with an overhand, closed grip, slightly wider than shoulder-width. | ET006: Open grip. |
| | | ET007: Grip width is too narrow or too wide. |
| | | ET009: Wrist excessively extended or tilted to the side. |
| Eccentric | AK26: Inhale. Maintain torso stability, allow the shoulders to naturally depress, and bend the elbows downward so the barbell lowers slowly and smoothly to the chest or just above it. | ET017: Exhalation. |
| | | ET018: Lowering too quickly. |
| | | ET019: Ending position of the barbell is too high or too low. |
| | | ET038: The angle between the upper arm and torso is too large or too small. |
| | | ET039: Insufficient barbell descent, resulting a shortened range of motion. |
| Concentric | AK27: Exhale. Focus on pectoralis major contraction. Using the shoulder joint as the pivot, apply symmetrical force with both hands to press the barbell upward. Keep the elbows slightly bent at the top. | ET014: Inhalation. |
| | | ET008: Excessive elbow flexion or hyperextension. |
| | | ET040: Hips rise or legs drive off the floor to assist during the press. |
| | | ET041: Raising the head or pushing the neck forward to assist during the press. |
| | | ET042: Shrugging the shoulders or shifting forward during the press. |
| | | ET043: The barbell sways left or right, or one arm dominates the force output. |
| | | ET044: Insufficient barbell elevation, resulting a shortened range of motion. |

| Decline Barbell Bench Press | Action Keystep | Error Type |
|---|---|---|
| Preparation | AK29: Lie supine on a decline bench at approximately 15° to the floor, feet flat. Keep the waist and abdomen braced, shoulders depressed, and the head, upper back, and hips on the bench. Chest lifted and abdomen engaged. | ET002: Excessive arching or collapsing of the lower back. |
| | | ET004: Shoulder elevation, lack of scapular retraction, or failure to depress the shoulders. |
| | | ET036: Feet not firmly planted on the ground. |
| | | ET037: Head, upper back, or glutes lifting off the bench. |
| | AK25: Fully extend the arms, gripping the barbell with an overhand, closed grip, slightly wider than shoulder-width. | ET006: Open grip. |
| | | ET007: Grip width is too narrow or too wide. |
| | | ET009: Wrist excessively extended or tilted to the side. |
| Eccentric | AK26: Inhale. Maintain torso stability, allow the shoulders to naturally depress, and bend the elbows downward so the barbell lowers slowly and smoothly to the chest or just above it. | ET017: Exhalation. |
| | | ET018: Lowering too quickly. |
| | | ET019: Ending position of the barbell is too high or too low. |
| | | ET038: The angle between the upper arm and torso is too large or too small. |
| | | ET039: Insufficient barbell descent, resulting a shortened range of motion. |
| Concentric | AK27: Exhale. Focus on pectoralis major contraction. Using the shoulder joint as the pivot, apply symmetrical force with both hands to press the barbell upward. Keep the elbows slightly bent at the top. | ET014: Inhalation. |
| | | ET008: Excessive elbow flexion or hyperextension. |
| | | ET040: Hips rise or legs drive off the floor to assist during the press. |
| | | ET041: Raising the head or pushing the neck forward to assist during the press. |
| | | ET042: Shrugging the shoulders or shifting forward during the press. |
| | | ET043: The barbell sways left or right, or one arm dominates the force output. |
| | | ET044: Insufficient barbell elevation, resulting a shortened range of motion. |

| Flat Dumbbell Bench Press | Action Keystep | Error Type |
|---|---|---|
| Preparation | AK24: Lie supine on a flat bench, feet flat on the floor. Keep the waist and abdomen naturally braced, shoulders depressed, and ensure the head, upper back, and hips remain on the bench. Lift the chest and engage the abdomen. | ET036: Feet not firmly planted on the ground. |
| | | ET002: Excessive arching or collapsing of the lower back. |
| | | ET004: Shoulder elevation, lack of scapular retraction, or failure to depress the shoulders. |
| | | ET037: Head, upper back, or glutes lifting off the bench. |
| | AK30: Hold the dumbbells with a neutral, closed grip. Bend the elbows, keeping the palms facing forward and forearms perpendicular to the ground, supporting the dumbbells on either side of the pectoralis major. | ET006: Open grip. |
| | | ET009: Wrist excessively extended or tilted to the side. |
| | | ET011: Forearms are not perpendicular to the ground. |
| | | ET045: Initial position of the dumbbell is too high or too low. |
| Concentric | AK31: Exhale. Maintain torso stability. Focus on contracting the pectoralis major to press both dumbbells inward and upward until the arms are extended above the chest. | ET014: Inhalation. |
| | | ET008: Excessive elbow flexion or hyperextension. |
| | | ET040: Hips rise or legs drive off the floor to assist during the press. |
| | | ET041: Raising the head or pushing the neck forward to assist during the press. |
| | | ET042: Shrugging the shoulders or shifting forward during the press. |
| | | ET046: Dumbbells collide or deviate outward. |
| | | ET047: Uneven elevation of the dumbbells on both sides. |
| | | ET048: Insufficient dumbbell elevation, resulting a shortened range of motion. |
| Eccentric | AK32: Inhale. Let the shoulders depress naturally, forearms remain vertical. Bend the elbows downward in a controlled manner until the dumbbells are at the sides of the shoulders. | ET017: Exhalation. |
| | | ET004: Shoulder elevation, lack of scapular retraction, or failure to depress the shoulders. |
| | | ET011: Forearms are not perpendicular to the ground. |
| | | ET018: Lowering too quickly. |
| | | ET038: The angle between the upper arm and torso is too large or too small. |
| | | ET049: Insufficient dumbbell descent, resulting a shortened range of motion. |

| Incline Dumbbell Bench Press | Action Keystep | Error Type |
|---|---|---|
| Preparation | AK28: Lie supine on an incline bench at approximately 30° to the floor, feet flat. Keep the waist and abdomen braced, shoulders depressed, and the head, upper back, and hips on the bench. Chest lifted and abdomen engaged. | ET002: Excessive arching or collapsing of the lower back. |
| | | ET004: Shoulder elevation, lack of scapular retraction, or failure to depress the shoulders. |
| | | ET036: Feet not firmly planted on the ground. |
| | | ET037: Head, upper back, or glutes lifting off the bench. |
| | AK30: Hold the dumbbells with a neutral, closed grip. Bend the elbows, keeping the palms facing forward and forearms perpendicular to the ground, supporting the dumbbells on either side of the pectoralis major. | ET006: Open grip. |
| | | ET009: Wrist excessively extended or tilted to the side. |
| | | ET011: Forearms are not perpendicular to the ground. |
| | | ET045: Initial position of the dumbbell is too high or too low. |
| Concentric | AK31: Exhale. Maintain torso stability. Focus on contracting the pectoralis major to press both dumbbells inward and upward until the arms are extended above the chest. | ET014: Inhalation. |
| | | ET008: Excessive elbow flexion or hyperextension. |
| | | ET040: Hips rise or legs drive off the floor to assist during the press. |
| | | ET041: Raising the head or pushing the neck forward to assist during the press. |
| | | ET042: Shrugging the shoulders or shifting forward during the press. |
| | | ET046: Dumbbells collide or deviate outward. |
| | | ET047: Uneven elevation of the dumbbells on both sides. |
| | | ET048: Insufficient dumbbell elevation, resulting a shortened range of motion. |
| Eccentric | AK32: Inhale. Let the shoulders depress naturally, forearms remain vertical. Bend the elbows downward in a controlled manner until the dumbbells are at the sides of the shoulders. | ET017: Exhalation. |
| | | ET004: Shoulder elevation, lack of scapular retraction, or failure to depress the shoulders. |
| | | ET011: Forearms are not perpendicular to the ground. |
| | | ET018: Lowering too quickly. |
| | | ET038: The angle between the upper arm and torso is too large or too small. |
| | | ET049: Insufficient dumbbell descent, resulting a shortened range of motion. |

| Dumbbell Fly | Action Keystep | Error Type |
|---|---|---|
| **Preparation** | AK24: Lie supine on a flat bench, feet flat on the floor. Keep the waist and abdomen naturally braced, shoulders depressed, and ensure the head, upper back, and hips remain on the bench. Lift the chest and engage the abdomen. | ET036: Feet not firmly planted on the ground. |
| | | ET002: Excessive arching or collapsing of the lower back. |
| | | ET004: Shoulder elevation, lack of scapular retraction, or failure to depress the shoulders. |
| | | ET037: Head, upper back, or glutes lifting off the bench. |
| | AK33: Hold the dumbbells with a neutral, closed grip, fully extending the arms to support the dumbbells above the chest. Keep the palms facing each other and elbows slightly flexed. | ET006: Open grip. |
| | | ET008: Excessive elbow flexion or hyperextension. |
| | | ET045: Initial position of the dumbbell is too high or too low. |
| **Eccentric** | AK34: Inhale. Using the pectoralis major's strength, control the arms as they open to the sides. The lowest point is when the dumbbells align with shoulder height. | ET017: Exhalation. |
| | | ET008: Excessive elbow flexion or hyperextension. |
| | | ET009: Wrist excessively extended or tilted to the side. |
| | | ET018: Lowering too quickly. |
| | | ET027: Changing the elbow angle, causing the dumbbell path to deviate from an arc. |
| | | ET049: Insufficient dumbbell descent, resulting a shortened range of motion. |
| | | ET050: Uneven descent of the dumbbells on both sides. |
| **Concentric** | AK35: Exhale. Keep the shoulders depressed. Focus on the pectoralis major contraction to bring the arms smoothly back together until they are extended, with elbows slightly bent. | ET014: Inhalation. |
| | | ET008: Excessive elbow flexion or hyperextension. |
| | | ET040: Hips rise or legs drive off the floor to assist during the press. |
| | | ET041: Raising the head or pushing the neck forward to assist during the press. |
| | | ET042: Shrugging the shoulders or shifting forward during the press. |
| | | ET046: Dumbbells collide or deviate outward. |
| | | ET047: Uneven elevation of the dumbbells on both sides. |
| | | ET048: Insufficient dumbbell elevation, resulting a shortened range of motion. |

| Seated Dumbbell Shoulder Press | Action Keystep | Error Type |
|---|---|---|
| **Preparation** | AK36: Feet flat on the floor, upper back and hips pressed against the seat. Chest lifted, abdomen braced, maintaining a normal lumbar curve. | ET002: Excessive arching or collapsing of the lower back. |
| | | ET004: Shoulder elevation, lack of scapular retraction, or failure to depress the shoulders. |
| | | ET036: Feet not firmly planted on the ground. |
| | | ET037: Head, upper back, or glutes lifting off the bench. |
| | AK37: Hold the dumbbells with an overhand, closed grip at shoulder level, keeping the wrists neutral and forearms perpendicular to the ground. Palms should be slightly forward, and elbows positioned in front of the shoulder joints. | ET006: Open grip. |
| | | ET009: Wrist excessively extended or tilted to the side. |
| | | ET011: Forearms are not perpendicular to the ground. |
| | | ET012: Palms facing inward or downward. |
| | | ET013: Excessive elbow adduction or abduction. |
| **Concentric** | AK38: Exhale. Engage the deltoids, using the shoulder joint as the pivot to press the dumbbells overhead until arms are nearly straight, elbows slightly bent at the top. | ET014: Inhalation. |
| | | ET008: Excessive elbow flexion or hyperextension. |
| | | ET040: Hips rise or legs drive off the floor to assist during the press. |
| | | ET041: Raising the head or pushing the neck forward to assist during the press. |
| | | ET042: Shrugging the shoulders or shifting forward during the press. |
| | | ET046: Dumbbells collide or deviate outward. |
| | | ET047: Uneven elevation of the dumbbells on both sides. |
| | | ET048: Insufficient dumbbell elevation, resulting a shortened range of motion. |
| **Eccentric** | AK39: Inhale. Bend the elbows in a controlled manner, lowering the dumbbells along the sides of the shoulders to the starting position, elbows slightly below shoulder level. | ET017: Exhalation. |
| | | ET018: Lowering too quickly. |
| | | ET049: Insufficient dumbbell descent, resulting a shortened range of motion. |
| | | ET050: Uneven descent of the dumbbells on both sides. |

| Deadlift | Action Keystep | Error Type |
|---|---|---|
| **Preparation** | AK01: Stand with feet shoulder-width apart and toes slightly turned outward. Keep the chest lifted and the core engaged. Maintain a neutral spine position with the scapulae depressed and retracted, ensuring trunk stability. | ET001: Feet positioned too narrowly or too widely. |
| | | ET002: Excessive arching or collapsing of the lower back. |
| | | ET003: Excessive forward flexion or excessive backward curvature of the spine. |
| | | ET004: Shoulder elevation, lack of scapular retraction, or failure to depress the shoulders. |
| | AK40: Keep the back straight and bend the knees to squat down. Position the shins close to the barbell and grip it with an overhand, closed grip, slightly wider than shoulder-width. Ensure the knees are aligned with the direction of the toes. | ET006: Open grip. |
| | | ET007: Grip width is too narrow or too wide. |
| | | ET051: Initial position of the barbell is too far forward or too far back. |
| | | ET058: The hip position is set too high or too low. |
| **Concentric** | AK41: Exhale. Engage the gluteus maximus and hamstrings to extend the knees and hips, lifting the barbell off the ground. Move the hips forward until standing fully upright. | ET014: Inhalation. |
| | | ET002: Excessive arching or collapsing of the lower back. |
| | | ET005: Trunk swaying forward and backward, indicating instability. |
| | | ET025: The barbell does not travel vertically relative to the ground. |
| | | ET044: Insufficient barbell elevation, resulting a shortened range of motion. |
| | | ET052: Hips rising faster than barbell, causing asynchronous force distribution between legs and hips. |
| | | ET053: Excessive forward or backward tilt of the shoulder joint. |
| | | ET062: Barbell deviating from the legs. |
| | | ET063: The knees lock out, with minimal extension in the hip and knee joints. |
| **Eccentric** | AK42: Inhale. Keep the back straight, hinge at the hips by leaning forward and shifting the hips back under control to full extension range. Hips remain higher than the knees and lower than the shoulders. | ET017: Exhalation. |
| | | ET003: Excessive forward flexion or excessive backward curvature of the spine. |
| | | ET018: Lowering too quickly. |
| | | ET054: Knees bend prematurely, causing the barbell to circle around them. |
| | | ET062: Barbell deviating from the legs. |
| | | ET063: The knees lock out, with minimal extension in the hip and knee joints. |

| Squat | Action Keystep | Error Type |
|---|---|---|
| **Preparation** | AK56: Stand with feet slightly wider than shoulder-width, toes slightly turned outward. Keep the chest lifted, engage the core, maintain a neutral spine, depress and retract the scapulae, and ensure torso stability. | ET061: Feet placed too narrowly or too widely during the squat. |
| | | ET002: Excessive arching or collapsing of the lower back. |
| | | ET003: Excessive forward flexion or excessive backward curvature of the spine. |
| | | ET004: Shoulder elevation, lack of scapular retraction, or failure to depress the shoulders. |
| | AK43: Grip the barbell with an overhand, closed grip, wider than shoulder-width, and position it on the trapezius and deltoids. | ET006: Open grip. |
| | | ET007: Grip width is too narrow or too wide. |
| | | ET010: The initial barbell position is set too high or too low. |
| **Eccentric** | AK44: Inhale. Look forward, keep the back straight, flex hips and knees to descend smoothly until the angle between thighs and calves is slightly less than 90°. Ensure the knees track the direction of the toes. | ET017: Exhalation. |
| | | ET002: Excessive arching or collapsing of the lower back. |
| | | ET005: Trunk swaying forward and backward, indicating instability. |
| | | ET033: Torso angle too shallow or too steep. |
| | | ET039: Insufficient barbell descent, resulting a shortened range of motion. |
| | | ET055: Excessive forward movement of the knees. |
| | | ET056: Knees caving inward. |
| **Concentric** | AK45: Exhale. Keep the back straight. Primarily drive with the quadriceps to extend the hips and knees, moving the hips forward until the knees are straight or slightly bent. | ET014: Inhalation. |
| | | ET002: Excessive arching or collapsing of the lower back. |
| | | ET032: Excessive knee flexion or hyperextension. |
| | | ET052: Hips rising faster than barbell, causing asynchronous force distribution between legs and hips. |

| Barbell Bent-Over Row | Action Keystep | Error Type |
|---|---|---|
| **Preparation** | AK01: Stand with feet shoulder-width apart and toes slightly turned outward. Keep the chest lifted and the core engaged. Maintain a neutral spine position with the scapulae depressed and retracted, ensuring trunk stability. | ET001: Feet positioned too narrowly or too widely. |
| | | ET002: Excessive arching or collapsing of the lower back. |
| | | ET003: Excessive forward flexion or excessive backward curvature of the spine. |
| | | ET004: Shoulder elevation, lack of scapular retraction, or failure to depress the shoulders. |
| | AK46: Keep the back straight and knees slightly flexed. Lean forward to a 30°–45° angle relative to the ground. Grip the barbell with an overhand, closed grip, slightly wider than shoulder-width, while keeping the wrists neutral. | ET006: Open grip. |
| | | ET007: Grip width is too narrow or too wide. |
| | | ET009: Wrist excessively extended or tilted to the side. |
| | | ET032: Excessive knee flexion or hyperextension. |
| | | ET033: Torso angle too shallow or too steep. |
| | AK47: Shift the hips backward and lift the barbell with straight arms to just below the knee level, keeping the elbows extended. | ET008: Excessive elbow flexion or hyperextension. |
| | | ET010: The initial barbell position is set too high or too low. |
| | | ET013: Excessive elbow adduction or abduction. |
| **Concentric** | AK48: Exhale. Focus on contracting the latissimus dorsi, pulling the upper arms back and elbows toward the midline of the back, drawing the barbell toward the lower abdomen. | ET014: Inhalation. |
| | | ET004: Shoulder elevation, lack of scapular retraction, or failure to depress the shoulders. |
| | | ET005: Trunk swaying forward and backward, indicating instability. |
| | | ET038: The angle between the upper arm and torso is too large or too small. |
| | | ET043: The barbell sways left or right, or one arm dominates the force output. |
| | | ET044: Insufficient barbell elevation, resulting a shortened range of motion. |
| | | ET057: Elbows flex before the scapula retracts. |
| **Eccentric** | AK49: Inhale. Use the latissimus dorsi tension to control the barbell's smooth descent until the arms are fully extended. | ET017: Exhalation. |
| | | ET002: Excessive arching or collapsing of the lower back. |
| | | ET018: Lowering too quickly. |
| | | ET039: Insufficient barbell descent, resulting a shortened range of motion. |

| T-Bar Row | Action Keystep | Error Type |
|---|---|---|
| **Preparation** | AK01: Stand with feet shoulder-width apart and toes slightly turned outward. Keep the chest lifted and the core engaged. Maintain a neutral spine position with the scapulae depressed and retracted, ensuring trunk stability. | ET001: Feet positioned too narrowly or too widely. |
| | | ET002: Excessive arching or collapsing of the lower back. |
| | | ET003: Excessive forward flexion or excessive backward curvature of the spine. |
| | | ET004: Shoulder elevation, lack of scapular retraction, or failure to depress the shoulders. |
| | AK46: Keep the back straight and knees slightly flexed. Lean forward to a 30°–45° angle relative to the ground. Grip the barbell with an overhand, closed grip, slightly wider than shoulder-width, while keeping the wrists neutral. | ET006: Open grip. |
| | | ET007: Grip width is too narrow or too wide. |
| | | ET009: Wrist excessively extended or tilted to the side. |
| | | ET032: Excessive knee flexion or hyperextension. |
| | | ET033: Torso angle too shallow or too steep. |
| | AK47: Shift the hips backward and lift the barbell with straight arms to just below the knee level, keeping the elbows extended. | ET008: Excessive elbow flexion or hyperextension. |
| | | ET010: The initial barbell position is set too high or too low. |
| | | ET013: Excessive elbow adduction or abduction. |
| **Concentric** | AK48: Exhale. Focus on contracting the latissimus dorsi, pulling the upper arms back and elbows toward the midline of the back, drawing the barbell toward the lower abdomen. | ET014: Inhalation. |
| | | ET004: Shoulder elevation, lack of scapular retraction, or failure to depress the shoulders. |
| | | ET005: Trunk swaying forward and backward, indicating instability. |
| | | ET038: The angle between the upper arm and torso is too large or too small. |
| | | ET043: The barbell sways left or right, or one arm dominates the force output. |
| | | ET044: Insufficient barbell elevation, resulting a shortened range of motion. |
| | | ET057: Elbows flex before the scapula retracts. |
| **Eccentric** | AK49: Inhale. Use the latissimus dorsi tension to control the barbell's smooth descent until the arms are fully extended. | ET017: Exhalation. |
| | | ET002: Excessive arching or collapsing of the lower back. |
| | | ET018: Lowering too quickly. |
| | | ET039: Insufficient barbell descent, resulting a shortened range of motion. |

| Dumbbell Bent-Over Row | Action Keystep | Error Type |
|---|---|---|
| **Preparation** | AK01: Stand with feet shoulder-width apart and toes slightly turned outward. Keep the chest lifted and the core engaged. Maintain a neutral spine position with the scapulae depressed and retracted, ensuring trunk stability. | ET001: Feet positioned too narrowly or too widely. |
| | | ET002: Excessive arching or collapsing of the lower back. |
| | | ET003: Excessive forward flexion or excessive backward curvature of the spine. |
| | | ET004: Shoulder elevation, lack of scapular retraction, or failure to depress the shoulders. |
| | AK50: Keep the back straight and knees slightly flexed. Lean forward to a 30°–45° angle relative to the ground. Hold the dumbbells with a neutral, closed grip while keeping the wrists neutral. | ET006: Open grip. |
| | | ET007: Grip width is too narrow or too wide. |
| | | ET009: Wrist excessively extended or tilted to the side. |
| | | ET013: Excessive elbow adduction or abduction. |
| | | ET032: Excessive knee flexion or hyperextension. |
| | | ET033: Torso angle too shallow or too steep. |
| **Concentric** | AK51: Exhale. Focus on the back musculature, bending the elbows to pull the dumbbells close along the sides of the body until the upper arms are parallel to the floor. | ET014: Inhalation. |
| | | ET005: Trunk swaying forward and backward, indicating instability. |
| | | ET015: Using a knee drive or leaning the torso backward to assist the movement. |
| | | ET021: Excessive elbow abduction, moving away from the torso. |
| | | ET047: Uneven elevation of the dumbbells on both sides. |
| | | ET048: Insufficient dumbbell elevation, resulting a shortened range of motion. |
| | | ET057: Elbows flex before the scapula retracts. |
| | | ET058: The hip position is set too high or too low. |
| **Eccentric** | AK52: Inhale. Brace the core, using the back musculature to control the dumbbells' slow, steady descent until the arms are fully extended. | ET017: Exhalation. |
| | | ET002: Excessive arching or collapsing of the lower back. |
| | | ET003: Excessive forward flexion or excessive backward curvature of the spine. |
| | | ET008: Excessive elbow flexion or hyperextension. |
| | | ET018: Lowering too quickly. |
| | | ET049: Insufficient dumbbell descent, resulting a shortened range of motion. |
| | | ET050: Uneven descent of the dumbbells on both sides. |

| Dumbbell Bent-Over Triceps Extension | Action Keystep | Error Type |
|---|---|---|
| **Preparation** | AK01: Stand with feet shoulder-width apart and toes slightly turned outward. Keep the chest lifted and the core engaged. Maintain a neutral spine position with the scapulae depressed and retracted, ensuring trunk stability. | ET001: Feet positioned too narrowly or too widely. |
| | | ET002: Excessive arching or collapsing of the lower back. |
| | | ET003: Excessive forward flexion or excessive backward curvature of the spine. |
| | | ET004: Shoulder elevation, lack of scapular retraction, or failure to depress the shoulders. |
| | AK50: Keep the back straight and knees slightly flexed. Lean forward to a 30°–45° angle relative to the ground. Hold the dumbbells with a neutral, closed grip while keeping the wrists neutral. | ET006: Open grip. |
| | | ET007: Grip width is too narrow or too wide. |
| | | ET009: Wrist excessively extended or tilted to the side. |
| | | ET013: Excessive elbow adduction or abduction. |
| | | ET032: Excessive knee flexion or hyperextension. |
| | | ET033: Torso angle too shallow or too steep. |
| | AK53: Keep the upper arms close to the sides of the body while allowing the forearms to hang down. Maintain an approximately 90° angle between the upper and lower arms. | ET011: Forearms are not perpendicular to the ground. |
| | | ET021: Excessive elbow abduction, moving away from the torso. |
| | | ET059: The angle between the upper arm and forearm is too small or too large. |

| | | |
|---|---|---|
| **Concentric** | AK54: Exhale. Keep the upper arms still, elbows close to the torso. Concentrate on the triceps to bend at the elbows, lifting the forearms until they align in a straight line with the upper arms, parallel to the floor. | ET014: Inhalation.
ET005: Trunk swaying forward and backward, indicating instability.
ET015: Using a knee drive or leaning the torso backward to assist the movement.
ET020: Elbows move forward or backward, upper arms are not stabilized.
ET021: Excessive elbow abduction, moving away from the torso.
ET047: Uneven elevation of the dumbbells on both sides.
ET048: Insufficient dumbbell elevation, resulting a shortened range of motion.
ET060: Forearms not aligned in a straight line with the upper arms. |
| **Eccentric** | AK55: Inhale. Use controlled tension in the triceps to lower the dumbbells slowly until the forearms are vertical to the floor. | ET017: Exhalation.
ET018: Lowering too quickly.
ET049: Insufficient dumbbell descent, resulting a shortened range of motion.
ET050: Uneven descent of the dumbbells on both sides.
ET059: The angle between the upper arm and forearm is too small or too large. |

Table 11: Full list of the FLEX error types' weight.

| Error Code | Error Description | Weight |
|---|---|---|
| ET001 | Feet positioned too narrowly or too widely. | 6 |
| ET002 | Excessive arching or collapsing of the lower back. | 9 |
| ET003 | Excessive forward flexion or excessive backward curvature of the spine. | 9 |
| ET004 | Shoulder elevation, lack of scapular retraction, or failure to depress the shoulders. | 8 |
| ET005 | Trunk swaying forward and backward, indicating instability. | 5 |
| ET006 | Open grip. | 6 |
| ET007 | Grip width is too narrow or too wide. | 6 |
| ET008 | Excessive elbow flexion or hyperextension. | 6 |
| ET009 | Wrist excessively extended or tilted to the side. | 9 |
| ET010 | The initial barbell position is set too high or too low. | 5 |
| ET011 | Forearms are not perpendicular to the ground. | 6 |
| ET012 | Palms facing inward or downward. | 4 |
| ET013 | Excessive elbow adduction or abduction. | 7 |
| ET014 | Inhalation. | 2 |
| ET015 | Using a knee drive or leaning the torso backward to assist the movement. | 8 |
| ET016 | Excessive forward head movement, obstructing the barbell's vertical path. | 7 |
| ET017 | Exhalation. | 2 |
| ET018 | Lowering too quickly. | 5 |
| ET019 | Ending position of the barbell is too high or too low. | 7 |
| ET020 | Elbows move forward or backward, upper arms are not stabilized. | 8 |
| ET021 | Excessive elbow abduction, moving away from the torso. | 8 |
| ET022 | Using forward or upward shoulder movement to assist the action. | 8 |
| ET023 | Elbows move forward or backward, causing the barbell path to deviate from an arc. | 7 |
| ET024 | Elbows rising much lower than the shoulder joints. | 8 |
| ET025 | The barbell does not travel vertically relative to the ground. | 9 |
| ET026 | Palms facing forward or backward. | 4 |
| ET027 | Changing the elbow angle, causing the dumbbell path to deviate from an arc. | 8 |
| ET028 | Ending position of the dumbbell is too high or too low. | 7 |
| ET029 | Elbows move forward or backward, causing the dumbbell path to deviate from an arc. | 8 |
| ET030 | Excessive wrist supination over-rotating the forearms. | 5 |
| ET031 | Initial position of the dumbbell is too far forward or backward. | 5 |
| ET032 | Excessive knee flexion or hyperextension. | 9 |
| ET033 | Torso angle too shallow or too steep. | 7 |
| ET034 | During abduction, the scapula elevates and the neck protrudes forward. | 6 |
| ET035 | Arms extending diagonally backward, lifting beyond the shoulder joint's horizontal level. | 5 |
| ET036 | Feet not firmly planted on the ground. | 8 |
| ET037 | Head, upper back, or glutes lifting off the bench. | 9 |
| ET038 | The angle between the upper arm and torso is too large or too small. | 8 |
| ET039 | Insufficient barbell descent, resulting a shortened range of motion. | 7 |
| ET040 | Hips rise or legs drive off the floor to assist during the press. | 7 |
| ET041 | Raising the head or pushing the neck forward to assist during the press. | 7 |
| ET042 | Shrugging the shoulders or shifting forward during the press. | 9 |
| ET043 | The barbell sways left or right, one arm dominates the force output. | 9 |
| ET044 | Insufficient barbell elevation, resulting a shortened range of motion. | 7 |
| ET045 | Initial position of the dumbbell is too high or too low. | 5 |
| ET046 | Dumbbells collide or deviate outward. | 5 |
| ET047 | Uneven elevation of the dumbbells on both sides. | 8 |
| ET048 | Insufficient dumbbell elevation, resulting a shortened range of motion. | 7 |
| ET049 | Insufficient dumbbell descent, resulting a shortened range of motion. | 7 |
| ET050 | Uneven descent of the dumbbells on both sides. | 8 |
| ET051 | Initial position of the barbell is too far forward or too far back. | 5 |
| ET052 | Hips rising faster than barbell, causing asynchronous force distribution between legs and hips. | 10 |
| ET053 | Excessive forward or backward tilt of the shoulder joint. | 9 |
| ET054 | Knees bend prematurely, causing the barbell to circle around them. | 9 |
| ET055 | Excessive forward movement of the knees. | 10 |
| ET056 | Knees caving inward. | 10 |
| ET057 | Elbows flex before the scapula retracts. | 8 |
| ET058 | The hip position is set too high or too low. | 6 |
| ET059 | The angle between the upper arm and forearm is too small or too large. | 6 |
| ET060 | Forearms not aligned in a straight line with the upper arms. | 6 |
| ET061 | Feet placed too narrowly or too widely during the squat. | 9 |
| ET062 | Barbell deviating from the legs. | 8 |
| ET063 | The knees lock out, with minimal extension in the hip and knee joints. | 10 |

