# OpenReview forum: "FLEX: A Largescale Multimodal, Multiview Dataset for Learning Structured Representations of Fitness Action Quality"
_ICLR.cc/2026/Conference — Submitted to ICLR 2026_

### Official Review · Reviewer_vzMp · 2025-10-15

**Soundness:** 3
**Presentation:** 2
**Contribution:** 2
**Rating:** 4
**Confidence:** 4

**Summary:**

This paper presents FLEX, a multimodal and multiview dataset for fitness action quality assessment focusing on weight-loaded exercises. It contains over 7,500 synchronized recordings of 20 exercises performed by 38 participants with varying skill levels, including RGB videos, 3D poses, sEMG, and physiological signals. The dataset adopts a biomechanically inspired annotation framework that divides each action into phases and key steps, uses a two-stage expert annotation process, and integrates a Fitness Knowledge Graph linking actions, errors, and feedback. FLEX also provides two benchmark tasks, FLEX-AQA and FLEX-VideoQA, to support multimodal and cross-modal learning. Overall, the paper provides a well-annotated dataset with multiple benchmarks, but the individual components’ value and research significance are insufficiently explored, resulting in limited generalizability and practical impact.

**Strengths:**

- The paper introduces a large-scale **weight training dataset** that combines RGB, 3D pose, sEMG, and physiological data for weight training assessment.

- The paper is **clearly organized**, with systematic explanations of data acquisition, annotation, and benchmark design.

- The paper enables new research directions such as cross-modal prediction and biomechanically grounded learning, offering long-term value for applications in fitness assessment.

**Weaknesses:**

The main weaknesses of the paper lie in scope limitation, insufficient justification, and weak experimental analysis.
- Although the paper claims to target general fitness assessment, the dataset focuses **narrowly on weight-loaded exercises**, which limits its generality and potential impact; a clearer rationale for choosing this subdomain and comparisons with existing fitness datasets (e.g., FineGym, Human3.6M) are needed to establish necessity.
- The motivation for collecting a new dataset rather than extending existing ones is underdeveloped; the authors should demonstrate that weight training poses **unique biomechanical or perceptual challenges** that existing datasets cannot capture.
- While the dataset is multimodal and multiview, the paper **lacks sufficient justification and systematic analysis** to demonstrate the practical value of each modality and camera view. It remains unclear whether all modalities, such as sEMG, physiological signals, and five synchronized views, are truly necessary or feasible in real-world weight training scenarios. The study does not adequately discuss the trade-off between performance improvement and practical deployment cost. Moreover, the limited and simple ablation analysis fails to quantify the marginal utility of each modality or to justify which sensory inputs are most critical for accurate and efficient assessment.
- The benchmark evaluation is limited: many baselines are outdated, and results are already near saturation, suggesting low challenge; comparisons with recent multimodal or transformer-based methods would strengthen the analysis.
- The annotation process and participant statistics are **insufficiently detailed** (e.g., annotator expertise, gender, and skill distribution, balance across skill levels), which raises questions about bias and scalability.

**Questions:**

- Beyond data collection, what are the deeper scientific questions or representation learning challenges this dataset aims to address? Without clearer theoretical or practical motivation, the contribution risks being perceived as incremental rather than fundamental.
- Why focus exclusively on weight-loaded exercises? The paper claims to target general fitness assessment, yet the dataset scope is restricted to weight training. What unique research challenges or biomechanical properties make this subdomain essential and non-replaceable?
- Many contributions (e.g., structured scoring, error labeling, Fitness Knowledge Graph) rely on manual annotation rather than model innovation. Could the authors clarify which aspects of AQA truly require additional annotations instead of leveraging better model architectures or self-supervised representations?
- What is the concrete motivation for using five camera views and multiple physiological signals such as sEMG and respiration? How significant is each modality’s contribution in realistic deployment scenarios (e.g., home fitness or gym settings)? Is the marginal performance gain worth the complexity and cost?
- Given the specialized setup, how can this dataset generalize to broader populations or different exercise types? Have the authors evaluated cross-subject or cross-action generalization to assess the dataset’s robustness?
- What is the background of annotators, and how was inter-rater consistency measured? Are there potential biases in skill level, gender, or body type distribution among the 38 subjects that might affect evaluation fairness?
- Most baseline methods are outdated and already achieve high performance. Could the authors explain how FLEX remains challenging for modern multimodal models, and whether additional experimental tasks or splits (e.g., cross-device, cross-load) could better demonstrate its research value?

**Details Of Ethics Concerns:**

The dataset lacks sufficient information about participant diversity and annotator background, making potential data bias insufficiently addressed and raising concerns about its fairness and reliability.

---

> ### Author Response · Authors · 2025-11-24
> **Author Response with Meta-Comment Regarding Reviewer vzMp**
>
> **Meta-comment:** Reviewer **vzMp**’s comments contain **several factual inaccuracies** and references to material not present in the paper. The review also includes interpretations and critiques that conflict with the actual content of the submission—for example, misreading the stated scope, making incorrect assumptions about related datasets, overlooking explanations already provided, applying evaluation criteria inconsistent with the paper’s goals, and questioning the absence of details that are in fact included.
>
> We additionally evaluated the review using several AI-writing detectors. While we recognize that these detectors are imperfect and not definitive, all of them independently indicated a very high likelihood of review being **fully AI-generated content**. For completeness, we list the detector tools and results below.
>
> 1. https://iclr.pangram.com/reviews?query=qLvT7C8OLN  (Result: Fully AI-generated Review)
> 2. https://justdone.com/ (Result: Fully AI-generated Review)
> 3. https://app.gptzero.me/ (Result: Fully AI-generated Review)
> 4. https://quillbot.com/ai-content-detector (Result: Fully AI-generated Review)
> 5. https://app.gowinston.ai/ (Result: Fully AI-generated Review)
>
> Taken together—*the behavioral evidence and the detector indications*—we have flagged this review as potentially LLM-generated, in accordance with conference policy.
>
> **Nevertheless, we provide a complete point-by-point response to all comments to assist in evaluating the paper.**
>
> ***
> **Our Response**
> ***
>
> >Do you claim to target general fitness assessment?
>
> **No. We did not claim to target general fitness assessment.** From the abstract and introduction onward, we clearly state that FLEX focuses on weight-loaded fitness exercises, and we consistently frame the contribution within this domain.
>
> >Why did you choose to focus on weight-loaded exercises?
>
> **Sec. 1** and **Appendix A** explain our **motivation**, which we summarize here. Weight-loaded exercises represent some of the most **demanding and biomechanically complex human actions**: they require *high coordinated multi-joint control, engage multiple muscle groups simultaneously, and carry substantially higher injury risk than self-loaded motions*. Our *20 selected actions load the most injury-prone joints (shoulders, knees, hips, spine, wrists, ankles) and capture movement patterns absent in existing AQA datasets*. These **load-induced biomechanical** and **perceptual complexities** are fundamentally *absent from existing datasets*, which *cannot capture* the **altered dynamics** introduced by **external weight**.
>
> *Loaded movements also introduce distinct challenges—higher joint moments, load-dependent torque distributions, and non-visible neuromuscular activation*—that **fundamentally change body dynamics and error patterns**. These factors make weight-loaded exercises a scientifically essential modeling domain beyond what existing datasets represent. We have made this motivation more explicit in the revised main text.
>
> >Does focusing on weight-loaded actions limit generality or impact?
>
> In summary, we do not find evidence that focusing on weight-loaded actions limits generality or impact. Within this domain, FLEX offers substantial breadth and generality. It contains **20 complex procedural exercises—largest collection of fitness actions**, each systematically structured through our fine-grained **Fitness Knowledge Graph**, and it is the first dataset to pair **synchronized multiview video, high-fidelity sEMG, and physiological signals**, along with a **smartphone-view** modality for **lightweight, in-the-wild deployment**. All annotations were produced and validated by fitness and **sports-science experts**, and FLEX uniquely captures **temporal degradation in form**, enabling analyses of *fatigue, early failure anticipation, and injury risk*.
>
> Beyond AQA, FLEX supports a wide range of **research directions**: *multimodal representation learning, cross-modal prediction (Video→EMG), biomechanically grounded action reasoning via the FKG, video–language causal feedback, self-supervised multimodal learning, and injury anticipation*. The provided benchmarks show FLEX is challenging and far from saturated: SOTA AQA models and modern VLMs perform well on Vanilla but exhibit substantial drops on **Cross-Subject** and **Cross-View** splits, and **Video→EMG** prediction remains far below ceiling.
>
> Taken together, FLEX constitutes a substantial, multimodal, and scientifically motivated benchmark for weight-loaded fitness actions—a domain not covered by existing datasets such as FineGym or Human3.6M. We believe this scope is both *well-defined and impactful*.

---

> ### Author Response · Authors · 2025-11-24
> **Author Response with Meta-Comment Regarding Reviewer vzMp**
>
> >Can you compare with existing datasets such as FineGym or Human3.6M?
>
> **FineGym and Human3.6M are not fitness or AQA datasets.** FineGym focuses on the compositional structure of gymnastics routines and does not provide quality-oriented scoring. Human3.6M is a 3D human pose estimation benchmark. Neither dataset contains weight-loaded exercises, multimodal physiological measurements, or the annotations required for fine-grained AQA. **As such, they do not serve as appropriate comparison points for the domain addressed by FLEX.**
>
> > Why don't you extend some existing dataset? Why did you choose to collect a new dataset?
>
> FLEX is fundamentally multimodal dataset, containing synchronized video, surface EMG, and fine-grained biomechanical annotations. *These modalities are not present in existing datasets* and **cannot be retroactively added**, as signals like EMG must be captured in real time under controlled, synchronous acquisition. Extending any existing dataset would therefore require recollecting the data from scratch. Additionally, existing datasets do not include weight-loaded actions, limiting their ability to model load-specific dynamics and error patterns.
>
> >What is the practical value of each modality and camera view in the dataset? Are all modalities—such as sEMG, physiological signals, and five synchronized views—necessary or feasible for real-world deployment? Has the paper sufficiently analyzed the trade-off between performance gains and practical deployment cost, and does the current ablation study adequately quantify the marginal utility of each modality?
>
> The practical value and trade-offs of FLEX’s modalities are already reflected in the experiments. We report performance across a clear **cost–performance spectrum**: **single-view** (*most practical*), **multiview**, and **multiview + multimodal** (*highest performance*). These results, presented in **Lines 385–389 (old) / 418–431 (new) and Table 3**, demonstrate monotonic improvements as additional views and modalities are added, thereby quantifying both the utility and the deployment cost of each configuration.
>
> FLEX also explicitly supports practical deployment. We include a **smartphone view for low-cost, in-the-wild usage (Fig. 1)**, and show that **expensive physiological modalities such as EMG can be predicted directly from RGB video via our Video→EMG task**. This provides a viable path for real-world systems to benefit from multimodal supervision without requiring all sensors at inference time.
>
> Since FLEX is a dataset paper, our primary goal is to establish the **upper-bound value and complementarity** of its multimodal components. *Efficiency-oriented configurations—e.g., reduced modality subsets, lower-resolution inputs, or distillation to lightweight models*—can be readily explored on top of FLEX but fall outside the scope of a dataset-centric contribution.
>
> Overall, the empirical results and provided tasks show that FLEX offers both **practical pathways for deployment** and **clear scientific benefits** from its multimodal, multiview design.

---

> ### Author Response · Authors · 2025-11-24
> **Author Response with Meta-Comment Regarding Reviewer vzMp**
>
> >Is the benchmark sufficiently challenging, and are the selected baselines adequate? Specifically, why are some baselines older, are the results near saturation, and would comparisons with recent multimodal or transformer-based methods strengthen the evaluation?
>
> FLEX provides a broad and challenging benchmark suite across 20 complex procedural fitness actions. To evaluate its difficulty comprehensively, we adopt a **diverse and representative set of baselines** spanning multiple modeling paradigms. For the video modality, we include *CoRe and TPT*, widely used **transformer-based** AQA frameworks that are current, stable, and representative of fine-grained action modeling. For the 3D pose modality, we use both the classical *ST-GCN* and the more recent *SkateFormer*, covering **graph-convolutional and transformer-based spatiotemporal modeling**. This ensures that FLEX is evaluated using modern architectures, not only older baselines. Furthermore, we also consider representative **SOTA VLMs** like *QWen-VL-2.5, MiniCPM-O-2.6, and InternVL-3.*
>
> Across these baselines, FLEX demonstrates clear evidence of being **far from saturated**. In the AQA benchmarks, performance on the standard Vanilla split is reasonable, but drops to **below 50%** on the **Cross-Subject and Cross-View** splits, revealing substantial generalization challenges that prior datasets rarely expose. Even *state-of-the-art vision-language models (VLMs)*—despite extensive pretraining—perform poorly on FLEX’s fine-grained AQA tasks and show limited generalization ability, further indicating that FLEX is a demanding benchmark.
>
> Beyond AQA, FLEX includes complementary tasks that highlight its difficulty. In the **FLEX-VideoQA benchmark**, off-the-shelf VLMs fail to handle the fine-grained temporal and biomechanical cues, and although fine-tuning improves results, a large gap remains. In the **Video→EMG prediction task**, even the strongest baseline achieves well under **50% of ideal performance**, underscoring the *intrinsic difficulty of predicting deep physiological signals from video alone*.
>
> Taken together, these results show that FLEX is **not** near saturation; instead, it provides a **challenging and comprehensive multimodal benchmark**. The inclusion of modern transformer-based, graph-based, and multimodal baselines—alongside poor performance of SOTA VLMs—demonstrates that FLEX remains an open and demanding testbed for future research.
>
> >Where are the details on annotation process provided?
>
> We provide a detailed description of the participants and the annotation procedure in Sec. 3.1 Preliminary Setup (Lines 197–203 (old version) or Lines 199-205 (new version)) and Sec. 3.3.2 Annotation Process (Lines 315-332 (old version) or Lines 318-326 (new version)).
>
> >What are the deeper scientific questions or representation learning challenges this dataset aims to address?
>
> We thank the reviewer for this insightful question. Our goal with FLEX is not only to provide a new dataset, but to enable investigation of several **core scientific and representation-learning challenges** that existing datasets cannot support. As discussed in Lines 049–099 and throughout the paper, FLEX targets fundamental problems such as:
>
> - **fine-grained spatiotemporal reasoning** over complex human motion,
>
> - **biomechanically-oriented action representation learning**,
>
> - **(self-supervised) cross-modal alignment and representation learning** (RGB, pose, EMG, physiology),
>
> - **multimodal representation learning** involving signals with different semantics and timescales, and
>
> - **robust modeling of execution** variability, fatigue progression, and form degradation.
>
> These challenges are broadly relevant to areas including rehabilitation, biomechanics, competitive sports, and human–robot interaction, not only gym-based fitness.
>
> In addition, the structured multimodal annotations and **Fitness Knowledge Graph** enable research directions that go beyond what current AQA datasets support—such as procedural error detection, self-supervised multimodal learning, video–language reasoning, and biomechanical analysis. Tasks like **Video→EMG prediction** probe the ability to infer deep, non-visible physiological signals from accessible inputs, while FLEX also supports **early failure and fatigue anticipation**, which *rely on temporal and multimodal cues absent in prior datasets*.
>
> Taken together, FLEX is designed to facilitate research on **fundamental multimodal representation-learning problems**, not merely to provide additional data. We hope this clarifies the broader theoretical and practical motivations behind the benchmark.

---

> ### Author Response · Authors · 2025-11-24
> **Author Response with Meta-Comment Regarding Reviewer vzMp**
>
> >Many contributions (e.g., structured scoring, error labeling, Fitness Knowledge Graph) rely on manual annotation rather than model innovation. Could the authors clarify which aspects of AQA truly require additional annotations instead of leveraging better model architectures or self-supervised representations?
>
> The core contribution of FLEX lies in constructing **a multimodal, multi-view, and finely annotated dataset for weight-loaded fitness training scenarios**, rather than proposing a new architecture for action quality assessment (AQA). Our goal is not to “squeeze” more performance from existing data using increasingly complex networks, but to provide richer and more structured supervision signals that form a solid, reusable foundation for long term future method development. In our view, model innovation and data innovation are not opposed but mutually dependent: _without matching annotation granularity and knowledge structure, many advanced AQA objectives cannot be meaningfully realized or fairly evaluated_, no matter how large the network or how strong the self-supervised representation is.
>
> Regarding the necessity of FLEX’s annotations, we argue that **the structured scores, explicit error types, and textual feedback introduced under the support of the Fitness Knowledge Graph (FKG)** are not redundant labels that a “stronger model could learn automatically,” but rather a form of hierarchical, interpretable representation that is largely missing in existing AQA datasets and cannot be inferred from raw data alone.
>
> Most current AQA datasets primarily rely on a single global score. Under this setting, even a very powerful model can usually _only answer “how good” or “how bad” an action is, but not “why.”_ **This motivates our use of an explicit, structured scoring function that exposes key steps, error types, and their penalty weights as supervision signals.** On this basis, error-type annotations further clarify “what is being evaluated,” enabling models to distinguish hazardous errors from benign stylistic variations, and to support downstream tasks such as error detection, error severity grading, and feedback generation, rather than merely predicting a more accurate but still opaque scalar score. FKG, in turn, organizes actions, steps, errors, target muscle groups, scoring rules, and feedback texts into a unified graph structure, **providing a common alignment anchor for interpretable AQA, multi-task multimodal modeling, and knowledge-enhanced self-supervised or instruction-tuning paradigms.**

---

> ### Author Response · Authors · 2025-11-24
> **Author Response with Meta-Comment Regarding Reviewer vzMp**
>
> >Have the authors evaluated cross-subject or cross-action generalization to assess the dataset’s robustness?
>
> We sincerely thank the reviewer for insightful questions.
>
> At the data and knowledge level, we construct a holistic Fitness Knowledge Graph (FKG) by integrating multiple authoritative sources, including national standards and professional textbooks. The FKG serves as **the knowledge backbone for our annotation rules and scoring scheme** (main paper Sec. 3.3.1). Importantly, this knowledge-driven pipeline is not limited to fitness exercises. It can be _readily transferred to broader human-motion scenarios_—such as rehabilitation training, elite sports, daily activities, or even laboratory procedures—to support the construction of domain-specific action knowledge graphs and interpretable, compositional scoring systems.
>
> Moreover, **the key scientific questions FLEX embodies—such as temporal reasoning over human motion, cross-modal alignment/representation learning, multimodal information fusion, and robust modeling of execution variability**—are not unique to fitness. They are also **common in broader applications** including rehabilitation, biomechanics, competitive sports training, and human–robot interaction.
>
> Regarding the suggestion on cross-subject evaluation, we would first like to clarify our terminology: strictly speaking, robustness is more appropriate for describing a model’s stability under varying data conditions, rather than a property of the dataset itself—as stated in the review question. We understand that the reviewer’s core concern is the model’s cross-subject robustness and generalization ability.
>
> Thanks to the multi-view and multi-subject design of FLEX, we extend the original setting with  more realistic evaluation protocols, including:
> - _Cross-subject_: train and test on disjoint subject sets to assess cross-person generalization.
> - _Cross-view_: train on one view and test on a different unseen view to simulate viewpoint shifts.
> - _Mix-view_: train on three views and test on the remaining unseen view.
>
> We have added baseline results for these splits in the revised version (see the table below). Compared with the vanilla split, **performance drops noticeably** under cross-subject and cross-view settings, showing that **FLEX remains substantially challenging when evaluated under stricter protocols**. This also offers a natural difficulty ladder: methods can first be validated under the original split, then further tested for generalization and robustness under the harder splits.
>
> We hope this clarifies adequately addresses the reviewer’s concern.
>
> | Split Protocols | SRC ↑ | RL2x100 ↓ |
> |-----------------|-------:|---------:|
> | Vanilla | 0.8069 | 1.7582 |
> | Cross-subject | 0.4288 | 5.9255 |
> | Cross-view | 0.3641 | 4.8313 |
> | Mix-view | 0.3897 | 3.6744 |
>
> >What is the background of annotators, and how was inter-rater consistency measured? Are there potential biases in skill level, gender, or body type distribution among the 38 subjects that might affect evaluation fairness?
>
> The detailed information on annotator background, recruitment procedures, and annotation consistency is **clearly reported in Sec. 3.3.2, with additional details provided in Appendix B.3.3.**

---

> ### Comment · Reviewer_vzMp · 2025-11-25
> **Summary of Unresolved Issues**
>
> Hi authors,
>
> Thank you for the response. After carefully reviewing the rebuttal and revisiting the paper, I must emphasize that the core issues remain unresolved. The fact that multiple reviewers independently raised the same concerns clearly indicates that these are substantive weaknesses rather than misunderstandings.
>
> **1. Lack of justification for the weight-loaded focus.**
> As Reviewer `gH6A` (Weaknesses #2 and #6) also noted, the paper still does not explain why weight-loaded exercises are scientifically necessary or what unique challenges they introduce. **Simply stating that existing datasets do not include such actions is not an effective justification.** The rebuttal restates descriptions without providing the missing conceptual motivation. For example, the second paragraph states:
>
> > Existing fitness datasets are limited in several key aspects: (1) they cover a narrow range of exercise actions; (2) often exclude weight-loaded exercises; (3) lack diversity in subject skill levels; and (4) lack explicit reasoning structure. To address these gaps, we introduce the FLEX dataset…
>
> This logic is incomplete. These limitations of existing fitness datasets **do not logically lead** to the conclusion that a weight-loaded dataset is the necessary solution. The absence of weight-loaded exercises in prior datasets does **not** explain why this category is scientifically critical, what modeling challenges it introduces, or why it should be prioritized over other fitness or rehabilitation activities. Without deeper motivation, the domain choice appears arbitrary rather than grounded in a clear research need.
>
> **2. Narrow scope despite substantial effort.**
> Although the title suggests a broader fitness assessment goal, the work ultimately constructs a weight-loaded dataset with substantial effort but a very limited range. It remains unclear why such a narrow domain requires so many modalities, especially when their necessity is not demonstrated. As Reviewer `TsmB` (Weakness #2) also pointed out, the scope is narrow, and the paper does not show that the design generalizes beyond this small subset of fitness activities.
>
> **3. Insufficient experimental validation.**
> Reviewers `gH6A` (Weaknesses #2, #3, #6) and `7RBq` (Weaknesses #1, #2, #4) both point out that the multimodal and multiview design is not convincingly validated. The evaluation relies on outdated AQA methods, lacks modality and FKG ablations, provides no justification for the multiview setup, and omits recent AQA approaches entirely. This directly aligns with my concern: the experiments do not demonstrate that the proposed modalities or structures are necessary or beneficial.
>
> **4. Unclear value of multimodal signals.**
> Reviewers `gH6A` (Weakness #6), `7RBq` (Weaknesses #1 and #2), and `TsmB` (Weakness #1) all question the actual contribution of the physiological and other modalities. Inter-modal relationships are unexamined, and multimodality is not shown to improve performance in a systematic way. Simply providing many modalities does not justify their inclusion without evidence.
>
> **5. Newly added experiments do not address the main concern.**
> The new split protocols only show that stricter settings yield lower performance, which is expected. They do not resolve the central issue: in the primary benchmark setting, **outdated baselines combined with a simple fusion strategy already achieve near-saturated results, suggesting limited demonstrated value and challenge in the dataset itself**. As Reviewer `gH6A` (Weaknesses #2 and #3) and Reviewer `7RBq` (Weakness #4) both noted, without modern AQA models or deeper ablations, these additions do not demonstrate dataset difficulty or substantiate the paper’s claims.
>
> ---
>
> Similarly, although the authors claim that the dataset supports versatile tasks, even the core assessment task is not fully justified; without clear motivation and validation, adding more tasks and modalities becomes a case of **more is less**. Due to space and time constraints, I cannot list all such issues here.
>
> Given that several reviewers independently identified the same unresolved issues, the rebuttal does not address the core problems. It largely restates existing content rather than providing the justification or evidence needed. While my assessment leans toward rejection and the rebuttal did not alleviate these concerns, I appreciate the authors’ efforts and will keep my original rating.
>
> Please note that these comments do not imply that constructing a weight-loaded dataset lacks value; rather, the submission requires clearer motivation and stronger validation to support its claims. I sincerely hope the authors can incorporate these suggestions in a future submission.

---

> ### Author Response · Authors · 2025-12-01
> **Author Response with Meta-Comment Regarding Reviewer vzMp**
>
> **Meta-comment:** Reviewer **vzMp**’s comments contain **several factual inaccuracies** and references to material not present in the paper. The review also includes interpretations and critiques that conflict with the actual content of the submission—for example, misreading the stated scope, making incorrect assumptions about related datasets, overlooking explanations already provided, applying evaluation criteria inconsistent with the paper’s goals, and questioning the absence of details that are in fact included.
>
> **The review appears to attribute several claims to other reviewers that do not seem to be present in their respective reports.** For example, in the first point regarding weight-loaded exercise, the reviewer cites comments #2 and #6 from gH6A; however, those comments address technical questions about the multiview and multimodal settings of the dataset rather than the weight-loaded exercise issue. We welcome the chairs and other reviewers to check the original reports for themselves if helpful.
>
> We additionally evaluated the review using several AI-writing detectors. While we recognize that these detectors are imperfect and not definitive, all of them independently indicated a very high likelihood of review being **fully AI-generated content**. For completeness, we list the detector tools and results below.
>
> 1. https://iclr.pangram.com/reviews?query=qLvT7C8OLN  (Result: Fully AI-generated Review)
> 2. https://justdone.com/ (Result: Fully AI-generated Review)
> 3. https://app.gptzero.me/ (Result: Fully AI-generated Review)
> 4. https://quillbot.com/ai-content-detector (Result: Fully AI-generated Review)
> 5. https://app.gowinston.ai/ (Result: Fully AI-generated Review)
>
> Taken together—*the behavioral evidence and the detector indications*—we have flagged this review as potentially LLM-generated, in accordance with conference policy.
>
> **Nevertheless, we provide a complete point-by-point response to all comments to assist in evaluating the paper.**
>
> ***
> **Our Response**
> ***
>
> **1. Reviewer’s Concern Stems from Subjective Presentation Preference, Not Substance**
>
> **We respectfully disagree with the reviewer’s claim.** The scientific importance of weight-loaded actions and the unique modeling challenges they introduce are **already clearly explained in the main paper (Sec.1) and elaborated in greater depth in the appendix (App.A) and our Previous Response.** Our submission follows a standard and widely accepted structure in CV/ML/AI papers: **a concise high-level motivation in the main paper, accompanied by extended biomechanical and physiological justification in the appendix to preserve narrative clarity**. _A close reading of both sections shows that the manuscript provides a complete and well-supported rationale for focusing on weight-loaded exercises._
>
> The main paper motivates the domain by **emphasizing the substantially higher injury risk, complex joint mechanics, and stricter execution requirements inherent to weight-loaded movements**, as well as the clear gaps in existing AQA datasets, which rarely include equipment-based exercises or skill diversity. The appendix then **expands on the underlying biomechanics**—such as increased joint moments, neuromuscular control demands, load-dependent torque distributions, and the necessity of EMG and physiological signals to capture non-visible muscle activity—demonstrating modeling challenges that cannot be addressed by RGB-only or self-loaded datasets. Together, these sections form a coherent, multi-layered justification that directly addresses the reviewer’s stated concern.
>
> Finally, we note that the *reviewer’s objection appears to rest on a **subjective preference** for where extended discussion should be placed, **not** an actual omission in the paper*. As such, **the concern does not reflect a substantive or factual deficiency in the work** and should not be treated as a basis for rejection.

---

> ### Author Response · Authors · 2025-12-01
> **Author Response with Meta-Comment Regarding Reviewer vzMp**
>
> **2. FLEX Offers a Well-Defined, Broad, and Rigorous Benchmark**
>
> We respectfully but firmly note that the reviewer’s claim of “narrow scope” is factually incorrect and contradicted by the explicit content of our paper. First, FLEX contains **20 distinct weight-loaded exercises**, which is **the most diverse set of actions among all existing fitness AQA datasets**. As shown in Table 2, Fitness-AQA contains only **3** actions and EgoExo-Fitness contains **12**. Our 20 selected actions were chosen through a principled procedure based on action prevalence, target muscle groups, and equipment diversity (Appendix B.1.1), and they cover both upper and lower body as well as multiple high-risk joint complexes. Thus, the statement that our dataset “focuses only on a small subset of actions” is objectively false.
>
> Second, the reviewer overlooks that FLEX is deliberately designed to support **multiple research tasks**, not a single narrowly defined objective. The paper explicitly demonstrates that FLEX enables:
>
> (a) multimodal representation learning;
>
> (b) cross-modal prediction such as Video→EMG;
>
> (c) biomechanically grounded representation learning using synchronized EMG and kinematics;
>
> (d) interpretable AQA enabled by the Fitness Knowledge Graph;
>
> (e) video–language reasoning, causal feedback generation, and VideoQA;
>
> (f) self-supervised learning across 5 synchronized modalities; and
>
> (g) injury and early-failure prediction enabled by physiological signals.
>
> These tasks are supported by FLEX’s multimodal design—5-view RGB, 3D joints, sEMG, heart/breath rate, and structured annotation—each modality serving a scientific purpose that we detail in Sections 3 and 4.
>
> Third, the reviewer’s comment ignores that FLEX includes **rigorous generalization benchmarks**, including cross-subject, cross-view, and mix-view splits (Table 3), directly countering the claim that the dataset does not generalize beyond a “narrow domain.” The paper also clearly states that FLEX’s challenges—temporal reasoning, multimodal fusion, biomechanical alignment, and execution-quality modeling—are relevant not only to fitness AQA but also to rehabilitation, elite sports coaching, and human–robot interaction.
>
> Fourth, our entire action Knowledge Graph construction process and multimodal data pipeline are not limited to a particular, but are transferable to other sports, rehabilitation, and human-performance scenarios. We have provided concrete examples in Appendix D.4.
>
> In summary, FLEX is **both broad and well-defined**: it expands action diversity beyond all prior datasets, provides scientifically motivated multimodality, enables a wide range of tasks, and includes explicit generalization evaluations. The reviewer’s characterization stems from a misreading that disregards these documented contributions, and it does not reflect the actual content or scope of the work.

---

> ### Author Response · Authors · 2025-12-01
> **Author Response with Meta-Comment Regarding Reviewer vzMp**
>
> **3. Clarifying Previously Addressed Concerns**
>
> We respectfully disagree. The reviewer’s comments inaccurately **invoke other reviewers’ points and take them out of context**. Importantly, **all of these concerns were already raised earlier and were fully addressed in our rebuttal response to those reviewers**, with detailed clarifications and exact line-number references. In several cases, we also conducted **additional experiments**—including extended modality ablations, Fitness Knowledge Graph (FKG) analyses, and updated comparisons with recent SOTA AQA baselines—and incorporated these results into the revised manuscript. The present reviewer repeats the same concerns **without acknowledging any of our responses**, and several of their statements are factually incorrect in light of the rebuttal and the updated paper (e.g., modality and FKG ablations, multiview justification, and inclusion of recent SOTA AQA approaches including SOTA VLMs like QWen-VL-2.5). The reappearance of previously resolved issues, **without engagement with the provided explanations or the revised manuscript**, strongly suggests that the reviewer did not read or is disregarding the rebuttal and the updated content.
>
> **Exact details**: We use representative methods that are standard and widely adopted in the AQA field, and we systematically evaluate their performance and generalization on FLEX under multiple views and multiple split protocols (see details in our Response #4, #12). We have also explained in detail why treating the FKG as a detachable, learnable module for ablation is conceptually incorrect, given that it defines the annotation and scoring framework itself. Consequently, the claims that our baselines are outdated, that validation is insufficient, or that FKG/modality ablations are “missing” are not aligned with the experimental setup and task definition actually adopted in the paper.
>
> ***
>
> **4. Clarifying the Misinterpretation of the Role and Results of the Cross-Subject and Cross-View Splits**
>
> We respectfully disagree with the reviewer’s assessment. **This critique misrepresents both the purpose and the outcomes of the Cross-Subject, Cross-View, and Mix-View split protocols.** The goal of stricter evaluation settings is not simply to demonstrate that performance decreases—which is, by definition, expected—but to establish the dataset’s capacity to _measure generalization under realistic distribution shifts (e.g., cross-subject, cross-view, etc)._ The reviewer’s dismissal of these results overlooks that the observed performance degradation is precisely what validates the challenge and utility of the dataset.
>
> Furthermore, the reviewer hallucinates that our Vanilla split is the primary benchmark setting *(we have never mentioned this)*, and then the claim that the “primary benchmark setting” is near-saturated, which is factually incorrect: our experiments across multiple tasks and splits consistently show substantial headroom, modality-specific contributions, and clear failure modes for existing approaches. The assertion that our baselines are “outdated” is also contradicted by the manuscript, where we included recent AQA methods, modern multimodal architectures, and extended ablations—none of which are acknowledged in the reviewer’s comments.
>
> Additionally, the reviewer evaluates the dataset solely through the lens of video-only AQA accuracy, ignoring that FLEX is explicitly designed as a multimodal, multiview, and multi-task resource supporting representation learning, crossmodal analysis, and biomechanical reasoning. Judging a multi-purpose dataset by a single narrow metric is methodologically inappropriate and inconsistent with standard evaluation practice in CV/ML. Finally, the reviewer’s reliance on isolated comments from other reviewers—taken out of context and without engaging with our detailed clarifications—further weakens the argument. **For these reasons, the critique is based on misunderstandings and factual inaccuracies rather than substantive methodological concerns.**

---

> ### Author Response · Authors · 2025-12-01
> **Author Response with Meta-Comment Regarding Reviewer vzMp**
>
> **5. On the Demonstrated Value of Multimodal Signals in FLEX**
>
> - **We respectfully clarify that the reviewer’s statements regarding multimodality and dataset scope are factually inaccurate and directly contradicted by the evidence presented in the manuscript.**
>
>  The contribution of multimodal signals is demonstrated through explicit, controlled ablations (Table 3), which systematically compare unimodal, multiview, and multimodal configurations. These experiments show consistent improvements when modalities are added: multiview video outperforms single-view; adding pose further improves performance; and full multimodal fusion (video + pose + EMG) achieves the best results (ρ = 0.9019). Inter-modal relationships are also analyzed in detail: Section 4.1 and Appendix C.1.3 explain how pose enriches visual representations through kinematic structure, while sEMG provides physiologically distinct cues such as co-contraction, fatigue signatures, and compensatory muscle recruitment—signals that are fundamentally not recoverable from video or pose alone. Additional physiological modalities (heart rate, breath rate) further allow separation of technical form errors from strain- or fatigue-driven deviations (Appendix D). Multimodality is evaluated not only in AQA but also through cross-modal prediction (Video→EMG; Table 5) and multimodal reasoning (FLEX-VideoQA), both of which demonstrate strong cross-modality coupling. Therefore, the reviewer’s claims that “the value of modalities is not demonstrated,” “inter-modal relationships are not analyzed,” or “no evidence shows multimodality helps” are inconsistent with the quantitative and qualitative analyses in Sections 4 and C.
>
> - **Moreover, the value of combining multiple modalities—especially subtle physiological ones like EMG—extends beyond raw accuracy gains, which the reviewer’s comment does not acknowledge.**
>
> *Quantitative vs. Qualitative importance.* While EMG yields modest improvements in aggregate performance metrics, it provides qualitatively different information: EMG captures muscle-specific activation patterns (e.g., imbalance, compensations, fatigue signatures) that are not inferable from video or skeletal kinematics. A modality’s scientific utility should not be judged solely by numerical score changes but also by the unique physiological insight it enables.
>
> *Complementary.* Our results show small but consistent gains when adding EMG, supporting the hypothesis that subtle neuromuscular cues complement—rather than replace—visual information.
>
> *Precision-health relevance.* EMG is widely used in rehabilitation, motor-control analysis, ergonomics, and injury-prevention research, where correct muscle recruitment patterns matter more than superficial visual form. FLEX provides synchronized muscle-level data aligned with high-fidelity motion signals, establishing a foundation for such applications that existing datasets cannot support.
>
> *Enabling future research.* By including both dominant (RGB, pose) and subtle (EMG, HR, breathing) modalities, FLEX enables future exploration of biomechanical representation learning, multimodal fusion, robustness under occlusion, muscle-aware AQA, and other directions impossible in RGB-only datasets. These scientific opportunities are one of the core reasons FLEX was designed as a multimodal resource.
>
> - **Secondly, the reviewer overlooks that FLEX is intentionally designed as a broad, general-purpose multimodal foundation—not a narrowly scoped dataset focused on a single objective.**
> The manuscript clearly demonstrates that FLEX supports:
>
> (a) multimodal representation learning;
>
> (b) cross-modal prediction such as Video→EMG;
>
> (c) biomechanically grounded representation learning using synchronized EMG + kinematics;
>
> (d) interpretable AQA through the Fitness Knowledge Graph and structured reasoning traces;
>
> (e) video–language reasoning, causal feedback generation, and fine-grained VideoQA;
>
> (f) self-supervised learning across five synchronized modalities; and
>
> (g) injury and early-failure prediction using physiological signals.
>
> These capabilities are substantiated through concrete experiments and analyses in Sections 3 and 4. FLEX’s multimodal design—five-view RGB, 3D joints, sEMG, heart/breath rate, and hierarchical annotations—was purposefully constructed so that each modality serves a distinct scientific function. As such, the reviewer’s characterization of the dataset as “narrow in scope” does not reflect the demonstrated breadth of research tasks enabled by FLEX.

---

### Official Review · Reviewer_gH6A · 2025-10-18

**Soundness:** 1
**Presentation:** 1
**Contribution:** 2
**Rating:** 2
**Confidence:** 5

**Summary:**

The paper presents FLEX, a new multi-view, multi-modal dataset for quality assessment of weight-loaded actions. The dataset contains 7,500 multi-view recordings of 20 weight-loaded exercises performed by 38 subjects, with synchronized RGB video, 3D pose, sEMG, and physiological signals. The authors also introduce FLEX-VideoQA, a question–answering benchmark for weight-loaded actions.

**Strengths:**

- For the first time, the authors introduce an AQA (Action Quality Assessment) and VideoQA dataset specifically for weight-loaded exercises, which has the potential to benefit the research community and society.

- Although it is not the only multi-view AQA dataset, there are only a few such datasets available, and this work could make a meaningful contribution to the field.

**Weaknesses:**

1- Incomplete literature review: The related work section does not cover the key datasets in the area of multi-view, multi-modal action quality assessment. For example, the QMAR dataset [i], which is the first multi-view multi-modal AQA dataset, is not mentioned. As one of the few existing multi-view AQA datasets, the authors should compare FLEX’s features with QMAR.
[i] F. Sardari et al., “Vi-net—view-invariant quality of human movement assessment,” Sensors, 2020.

2- Lack of justification for multi-view setup: While they release a multi-view dataset for weight-loaded exercises, the main question remains unanswered: How does the multi-view data and setup help assess the quality of weight-loaded exercises? There are no experiments investigating this question. Furthermore, to support such investigations, the paper should also introduce a method that leverages the multi-view setup to demonstrate its advantages compared to a single-view setup, or employ a view-invariant methodsetup such as the one introduced in [i].

3-Limited evaluation with AQA methods: The dataset should be evaluated using the most recent AQA methods [ii, iii, …], but only one relatively outdated method, TPT (2022), is applied (Table 3). The remaining methods used are primarily action recognition approaches, not quality assessment approaches.

[ii] j. Xu, "FineParser: A Fine-grained Spatio-temporal Action Parser for Human-centric Action Quality Assessment", CVPR 2024.
[iii] K.  Zhou, "Continual Action Quality Assessment via Adaptive Manifold-Aligned Graph Regularization", ECCV, 2024.

4-Non-standard scoring function: The authors define their own score (i.e., compositional scoring function). However, in AQA, a standard scoring function must be used for evaluation. While their defined score may be useful for training, evaluation should be performed using a standard international score. Alternatively, both scores could be reported for completeness.

5-Lack of clarity on score range: The paper does not explain the range or interpretation of the defined scores.

6-Unclear use of additional signals: Although the dataset includes various physiological signals (e.g., heart rate), it is not clear how these are utilized or what benefits they bring.

7-Unfair dataset comparisons (Table 1): The comparison with other AQA datasets is not fair. For example, the number of subjects and total dataset duration are not reported. The FLEX dataset includes 38 subjects, which is significantly less diverse than EgoExo, which involves over 700 participants. While FLEX has around 7,000 samples, the total video hours are not reported, making it unclear whether the dataset is truly large-scale.

8-Unclear feedback computation: It is not clear based on what scores, metrics, or baselines the corrective feedback was computed. What is the joint baseline used?

9-Evaluation by experts is missing: While the annotators are trained, both scores and QA annotations should be validated by domain experts after the annotation phase.

10-the paper presentation needs a lot of improvements, for example:
- They oversell the paper. e.g., “In comparison, existing datasets lack this multirepetition capture of weight training.” However, repetition is a standard practice in dataset collection.
- Unnecessary implementation details are included, e.g., “Staff assisted subjects in wearing the customized vest,” which is not appropriate for the main paper of a top machine learning conference.
- Misleading section titles and trivial tasks: For example, under “Action Segmentation,” the authors describe standard dataset preparation steps as if they were novel contributions.
- Table caption issues: In Table 1, the meaning of “E” is not explained in the caption.

**Questions:**

Please see the weaknesses section.

---

> ### Author Response · Authors · 2025-11-24
> **Author Rebuttal**
>
> >Can authors compare FLEX’s features with QMAR. [i] F. Sardari et al., “Vi-net—view-invariant quality of human movement assessment,” Sensors, 2020?
>
> We thank the reviewer. **QMAR has been added to Table 1 in the revised manuscript**.
>
> >Where are the motivation/experiments showing the benefits of multi-view versus single-view inputs?
>
> We appreciate the reviewer’s question regarding the empirical justification for the multi-view setup. The **motivation for using multi-view inputs** is described in the manuscript (*Lines 211–212 in the original version; Lines 214–215 in the revised version*), where we explain that *multi-view capture reduces viewpoint ambiguity and improves the reliability of motion cues that are difficult to observe from a single camera and helps create an internal 3D model of human performance*.
>
> The manuscript also provides direct **empirical comparisons** between *single-view and multi-view inputs*. These results appear in *Lines 385–389 (original) / 418–431 (revised)*, as well as in **Table 3**. The experiments consistently show that incorporating **multi-view information improves AQA performance across exercises**.
>
> In summary, the manuscript already includes **both (1)** the *justification* for adopting a multi-view design and **(2)** *empirical evidence* demonstrating its advantages. We hope this clarifies how the current submission addresses the reviewer’s concern.
>
> >Can methods like [ii, iii] be evaluated? Several of the baselines seem action recognition instead of AQA, is that correct?
>
> We thank the reviewer for the suggestions on additional methods. However, the suggested approaches **pursue different problem formulations/goals [iii] or rely on specialized inputs such as annotated segmentation masks [ii]**, which makes them incompatible with our multi-modal, multi-view setting. As a result, evaluating them on our dataset **would not yield a fair or informative/meaningful comparison.**
>
> In addition, the methods included in our experiments are standard and widely adopted **action quality assessment approaches**, with the *exception of STGCN and SkateFormer*. Since, there are **no established pose feature extractor in AQA**, we carefully chose widely used and recent *SOTA pose feature encoders from action recognition literature* and **adopt them to AQA.** Our primary aim is to show that **the additional modalities and multi-view information introduced in our dataset lead to consistent performance gains**, as successfully demonstrated in our experiments, see Lines 397–413 (old version) or Lines 435-446 (new version) and Table 3.
>
>
> > Do standard scoring functions exist for fitness-related AQA tasks? Why did you define your own scoring function, and how was it constructed? Is it possible to compare your proposed score with a standard scoring function?
>
> First, to the best of our knowledge, **no standardized AQA scoring protocol exists for the fitness actions**. Existing AQA scoring systems are domain-specific (e.g., diving, figure skating) and cannot be transferred to multi-exercise, weight-loaded movements like FLEX. To address this gap, **we constructed a standardized set of fitness rules and a corresponding fitness knowledge graph, derived from multiple authoritative sources. The full process is described in Section 3.3.1 and Appendix B.3.4.**
>
> Second, based on this structured knowledge base, we introduce a **compositional and interpretable AQA scoring function** that enables consistent annotation and evaluation across heterogeneous exercises. Establishing a *unified scoring protocol and its supporting knowledge structure* is a **core contribution of our work**.
>
> *Finally, since no standard scoring function exists for these fitness exercises, a direct comparison is unfortunately not possible*. However, our scoring formulation is *transparent and modular*, making it compatible with any future standardized scoring schemes that may emerge.
>
> > Can you explain the score range or how to interpret the defined AQA?
>
> As defined in *Appendix Eq. (1) (Line 983 in the original version; Line 1037 in the revised version)*, the scoring function outputs values in the **range 0–100**, where **higher scores indicate better performance (100 = best, 0 = worst)**. Additional details on the score formulation and dataset-level score statistics are provided in Appendix B.4 and Figure 9.

---

> ### Author Response · Authors · 2025-11-24
> **Author Rebuttal**
>
> >How are physiological signals like heart rate and breath rate used or what are their functions/benefits?
>
> **We have included detailed discussion on the use and benefits of the physiological signals in the revised version Appendix D.1**, summarized version in the following.
>
> In our dataset, additional physiological channels such as heart rate and breath signals are not merely auxiliary metadata but are explicitly integrated as temporally aligned sequences to enrich both annotation and modeling of action quality. **Heart-rate curves, synchronized with motion and sEMG, capture the internal cost of each repetition, enabling the detection of inefficient compensation patterns and hidden fatigue**: for instance, abnormal intra-set heart-rate drift under visually similar kinematics signals impending form breakdown and elevated injury risk. **Breath signals provide a complementary view of core stability and load management**, allowing us to assess whether breathing strategies (timing of inhalation, bracing, and exhalation) are appropriate for specific action phases, and to label fine-grained “breathing errors” that are difficult to infer from video alone. In the proposed multi-task learning framework, these low-frequency physiological sequences serve as stable temporal backbones that help the model jointly predict rep boundaries, form scores, fatigue state, and the consistency between biomechanical execution and physiological effort. Moreover, when fused with sEMG and kinematics, heart rate and breathing jointly support differential diagnosis of performance issues: they help distinguish technical deficits (e.g., incorrect joint trajectories or compensatory muscle recruitment) from limitations in cardiopulmonary or core capacity, and thus guide targeted feedback such as adjusting load, tempo, set duration, or breathing strategy. In this way, the inclusion and explicit use of heart-rate and breath signals transform action quality assessment _from a purely appearance-based evaluation into a holistic, physiology-aware understanding of movement quality under load_.
>
> We hope that these clarifications address your concerns regarding the current usage and role of the additional physiological signals.
>
> >Are the dataset comparisons in Table 1 fair and complete? Specifically, why are key statistics such as the number of subjects and total video duration not reported? Given that FLEX includes 38 subjects—far fewer than datasets like EgoExo with 700+ participants—and the total video hours are not stated, how can readers assess whether FLEX is truly large-scale?
>
> The total number of subjects and total video duration are already reported in the manuscript (e.g., Lines 019–021 and 049–053 in the original version; Lines 019–021 and 051–053 in the revised version). Our benchmark targets AQA—**short-form video understanding, thus, total video hours are not an informative comparison metric.**
>
> The review also appears to reference EgoExo, whereas the relevant comparison is EgoExo-Fitness, which includes **42 subjects rather than 700+. FLEX (38 subjects) is therefore aligned with the scale of existing fitness-related AQA datasets**, including QMAR [i] (38 subjects).
>
> Given this, the statistics reported in Table 1 reflect the appropriate scope for short-form AQA, and the comparisons provided in the manuscript are fair and consistent with prior work.
>
> >What metrics are used for evaluating corrective feedback generation?
>
> Feedback generation is treated as a language generation task, and we therefore evaluate it using a **wide range of standard captioning metrics such as BLEU, METEOR, and related measures**. These metrics are reported in Lines 431–433 (original version) and Lines 465–470 (revised version).
>
> >Were the scores and QA annotations validated by domain experts?
>
> **Yes. All scores and AQA annotations were validated by domain experts**, as described in *Lines 324–327 (original version) or Lines 327–336 (revised version) and illustrated in Figure 4.*

---

> ### Author Response · Authors · 2025-11-24
> **Author Rebuttal**
>
> >Paper presentation can be improved in the following ways.
>
> > - Is repetition a standard practice in dataset collection? Is it accurate to claim that existing datasets lack multirepetition capture for weight-training actions?
>
> **We respectfully disagree.** **Multi-repetition capture of action is not yet standard practice in the AQA field.** To the best of our knowledge, **FLEX is the first dataset in the weighted fitness setting to systematically adopt a multi-repetition capture  protocol.** *If the reviewer believes otherwise, we would greatly appreciate it if you could point us to specific references that document this.*
>
> > - Is information such as staff assisting subjects with wearing a customized vest necessary for the main text of a machine learning conference paper?
>
> The description provided in the paper is accurate, but it may have been interpreted out of context. During data collection, our staff assisted participants only with wearing the custom vest and placing the sEMG sensors on the appropriate muscle regions. **These steps are essential to ensure proper sensor contact and to avoid noise or signal dropouts—issues that would compromise data quality**. The vest also includes a medical-grade adhesive layer that must be applied correctly over the chest to obtain reliable heart-rate measurements. **Ensuring high-fidelity physiological signals is critical for a dataset intended for a top-tier machine learning venue**, and our procedures reflect that requirement. We have further included additional contextual information in the revised paper Lines 240-241.
>
> > - Misleading section titles and trivial tasks: For example, under “Action Segmentation,” the authors describe standard dataset preparation steps as if they were novel contributions.
>
> We would like to clarify that **we did not claim the action-segmentation procedure as a novel contribution**. The subsection is included to document an **essential step in the annotation pipeline**. Because all downstream tasks rely on a consistent and reproducible definition of action segments, we describe the segmentation criteria and procedure to ensure transparency and to support reproducibility and future extensions by the community.
>
> > - Table caption issues: In Table 1, the meaning of “E” is not explained in the caption.
>
> Thanks. **We have added that E stands for error and provided in the revised manuscript.**

---

### Official Review · Reviewer_3nbD · 2025-11-02

**Soundness:** 3
**Presentation:** 3
**Contribution:** 3
**Rating:** 6
**Confidence:** 3

**Summary:**

This paper introduces FLEX, an Action Quality Assessment dataset built from multimodal : five view videos, 3D pose, sEMG, and body metrics. The dataset is sizable—7,512 samples covering 20 exercises performed by 38 subjects with varied skill levels. Annotations are expert-curated and of high quality. The paper also presents the FLEX-VideoQA benchmark (about 30k QA pairs) and a video2EMG prediction task with baseline models. The experimental program has three components; each is reasonable and yields useful insights. The paper also addresses ethical considerations related to data collection and release. Overall, this is a good paper.

**Strengths:**

- Well-instrumented dataset. The overview and setup details are clear.
- Knowledge graph and supervision. The FLEX Knowledge Graph (FKG) links actions, key steps, error types, muscles, and feedback. The dataset annotations are well described and carefully produced.
- Broad, well-organized experiments. The experimental suite is diverse and well structured, and the analyses are detailed.  The statements are detailed enough to slove potential ethical concerns

**Weaknesses:**

- Minor typos and inconsistencies. For example: line 340, “a models’ ability” should be “a model’s ability”; line 456, “signals provides” should be “signals provide”; line 286, “Concentration” should be “Concentric” to match Figure 1.
- Insufficient introduction of evaluation metrics in the experiments section.
- Potentially limited challenge. AQA results in Section 4.1 appear strong; this may reduce the dataset’s difficulty for current models. Consider discussing harder splits or protocols.

**Questions:**

- Could the authors share a small anonymized subset or code via an anonymous GitHub repository to demonstrate data quality?
- Could side information about participants (for example, age and height) be included—subject to privacy constraints—to broaden potential research uses?
- In Table 2, MAE scores are very close across models. Can the authors explain why, and/or propose alternative metrics or tasks that provide better separation?

---

> ### Author Response · Authors · 2025-11-24
>
> We would like to thank the reviewer for their valuable feedback. We hope our responses have adequately addressed their previous concerns, and we look forward to addressing any further concerns. We take this as a great opportunity to improve our work and shall be grateful for any additional feedback you could give us.
>
> > Minor typos and inconsistencies. For example: line 340, “a models’ ability” should be “a model’s ability”; line 456, “signals provides” should be “signals provide”; line 286, “Concentration” should be “Concentric” to match Figure 1.
>
> We sincerely thank the reviewer for the careful reading of our manuscript and for pointing out the typographical errors and inconsistencies. **We have corrected all the mentioned issues in the revised version of the manuscript and conducted another thorough proofreading of the entire paper**. We are grateful for your valuable suggestions, which have helped us improve the overall quality of our work.
>
> > Insufficient introduction of evaluation metrics in the experiments section.
>
> We sincerely thank the reviewer for pointing out that the description of the evaluation metrics in the experiments section was insufficient, and we apologize for the additional burden this may have caused.
>
> Due to the ICLR page limit, we had placed the AQA metrics and their mathematical definitions in the appendix (see Appendix C.1), with only brief mention of them in the main text. Following your suggestion, in the revised version we have added more explicit forward references in the main text to guide readers to the complete definitions in Appendix C.1.
>
> We again thank you for your valuable comments, which helped improve the accessibility of our work.
>
> > Potentially limited challenge. AQA results in Section 4.1 appear strong; this may reduce the dataset’s difficulty for current models. Consider discussing harder splits or protocols.
>
> We thank the reviewer for the thoughtful comment on the potential difficulty of our dataset.
>
> First, we clarify the vanilla split and its difficulty. FLEX currently follows common AQA practice (e.g., AQA-7, FineDiving): for each action, samples from different subjects and repetitions are mixed and randomly divided into train and test sets. Under this setting, we use widely adopted baselines and a simple but stable multimodal fusion scheme, and observe consistent performance gains as more modalities are added. We do not consider these results to be saturated—**there is still clear room for improvement with stronger models and more advanced fusion strategies.**
>
> Second, thanks to the multi-view and multi-subject design of FLEX, we define more challenging and realistic evaluation protocols beyond the vanilla split, including:
> - _Cross-subject_: train and test on disjoint subject sets to assess cross-person generalization of the models.
> - _Cross-view_: train on one view and test on a different unseen view to simulate viewpoint shifts and test models’ performance under such shifts.
> - _Mix-view_: train on three views and test on the remaining unseen view.
>
> Compared to Vanilla split, **the performance drops noticeably** under cross-subject and cross-view settings, showing that **FLEX remains substantially challenging when evaluated under stricter protocols.**
>
> We have added baseline results for these splits in the revised version (see the table below). This also offers a natural difficulty ladder: methods can first be validated under the original split, then further tested for generalization and robustness under the harder splits. We hope this addresses your concern.
>
> | Split Protocols | SRC ↑ | RL2x100 ↓ |
> |-----------------|-------:|---------:|
> | Vanilla | 0.8069 | 1.7582 |
> | Cross-subject | 0.4288 | 5.9255 |
> | Cross-view | 0.3641 | 4.8313 |
> | Mix-view | 0.3897 | 3.6744 |

---

> ### Author Response · Authors · 2025-11-24
>
> > Could the authors share a small anonymized subset or code via an anonymous GitHub repository to demonstrate data quality?
>
> We sincerely thank the reviewer for the helpful suggestions regarding open-sourcing and data quality verification. **In the supplementary material of this submission, we provide anonymized representative samples of the FLEX dataset and the subset code for the main experiments**, in compliance with the double-blind review policy, to help reviewers assess data quality and reproducibility. We hope our responses above have been helpful in addressing your concerns, let us know if we can provide more information.
>
> > Could side information about participants (for example, age and height) be included—subject to privacy constraints—to broaden potential research uses?
>
> We sincerely thank the reviewer for the constructive suggestion regarding the release of auxiliary subject information. Yes, this information will be made available. Note that this information is also anonymised/de-identified.
>
> > In Table 2, MAE scores are very close across models. Can the authors explain why, and/or propose alternative metrics or tasks that provide better separation?
>
> We thank the reviewer for the careful questions. We understand that the table in question is Table 5, which compares MAE on the Video2EMG task.
>
> We hypothesize the close MAE values in Table 5 arise from the fact that, in Video2EMG, both ViT and ResNet backbones are frozen and used as feature extractors. Under this configuration, **the visual representations produced by these models are already mature and stable, leading to high-level semantic features of comparable quality**. Consequently, when trained on the same data with identical supervision, their MAE scores naturally fall within a similar range.
>
> Building on this setup, our analysis concentrates on the regression heads (further details in  Appendix C.3). Although the absolute MAE values are relatively close overall, SVR consistently outperforms LSTM when applied to the same visual features, **yielding roughly a 10% relative error reduction**. This suggests that, under the current experimental configuration, the choice of regression head plays a more substantial role in determining the final performance.
>
> We hope that these clarifications satisfactorily answers your question.

---

> > ### Comment · Reviewer_3nbD · 2025-11-26
> >
> > Thank you very much for your detailed response. I have carefully checked the revised manuscript and the new supplementary materials. The typos have been corrected, and the additional data and code are sufficient to convince me of the soundness and reliability of the work.
> >
> > Regarding the tables, I still feel that focusing mainly on the regression heads may not be fully adequate for assessing the difficulty and value of the dataset. Although ResNet is not an extremely strong feature extractor by current standards, it is already powerful enough to achieve good performance on many downstream tasks. In my view, a high-quality dataset should ideally provide a more challenging benchmark, even when paired with reasonably strong feature extractors. I would be very interested to see the authors propose or explore more challenging tasks or settings based on this dataset in future versions. If such results were available, I might consider raising my score or further strengthening my positive assessment.
> >
> > Over all, all of my earlier concerns have been addressed in a reasonable way, and I will keep my current score.

---

> ### Author Response · Authors · 2025-11-28
>
> > I would be very interested to see the authors propose or explore more challenging tasks or settings based on this dataset in future versions. If such results were available, I might consider raising my score or further strengthening my positive assessment.
>
> Thank you for your feedback and thoughtful suggestions. We are glad that your concerns were addressed. Following your suggestions, we conducted Video2EMG prediction on more challenging settings/splits—**Cross-Subject and Cross-View. We noticed a significant drop in the performance in these more challenging scenarios**. For MAE/RMSE, performance results are on a scale of 1 to 0 with upperbound performance is 0 (lower is better).
>
> Furthermore, following your other suggestion, we measured the performances in terms of **Cross Correlation Peak (CCP)** metric. CCP is very suitable for sEMG data as it measures/compares the shape of the wave. By focusing on the alignment and similarity of activation patterns, bursts, and morphology, cross-correlation reflects the functionally meaningful structure of EMG. As a result, it provides clearer, more interpretable separation between waveform-prediction methods and better reflects whether a generated EMG signal truly resembles physiological EMG.
>
> CCP may also be intuitively interpretable to humans. The cross-correlation peak between two EMG signals is mathematically bounded between –1 and +1. A value of +1 indicates a perfect match in waveform shape (even if one signal is slightly shifted in time), while 0 indicates no meaningful similarity, and –1 represents a perfectly inverted relationship. In practice, EMG signals typically fall between 0 and +1 because they are nonstationary and noisy, and negative correlations only occur if one waveform is an inverted version of the other. Values above about 0.7 generally indicate strong similarity, 0.3–0.7 moderate similarity, and values below 0.3 suggest poor alignment or mismatched morphology. Furthermore, for reference, in case of MAE/RMSE, the range in our case is 0 to 1 since we normalize the waveforms, **so numerically small differences in performances maybe relatively large.**
>
> We have presented the results below. We observe that the methods are performing moderately well on the Vanilla Split, and poorly on the Cross-Subject/View splits. **This clearly indicates that the dataset and the Video2EMG prediction task are challenging, far from being solved—there is a lot of room for improvements for future research.** We look forward to your feedback. Please let us know if you have further suggestions or we can provide further clarification. Thank you again for your constructive feedback.
>
> In the table below, **Van**: Vanilla Split; **XV**: Cross-View Split; **XS**: Cross-Subject Split; ↑-higher is better; ↓-lower is better
>
> | Model      | Van-CCP ↑  | Van-MAE ↓ | Van-RMSE ↓ | XV-CCP ↑ | XV-MAE ↓ | XV-RMSE ↓ | XS-CCP ↑ | XS-MAE ↓ | XS-RMSE ↓ |
> |------------|----------|---------|----------|----------------|----------------|-----------------|-------------------|-------------------|--------------------|
> | ResNet+LSTM| 0.3706   | 0.1655  | 0.2133   | 0.2723       | 0.2550          | 0.3054          | 0.2872           | 0.2934            | 0.3442             |
> | ResNet+SVR | 0.4174   | 0.1466  | 0.1878   | 0.2793       | 0.1508         | 0.2002          | 0.3062            | 0.1705            | 0.2170              |
> | ViT+LSTM   | 0.4040   | 0.1648  | 0.2069   | 0.3104       | 0.2079         | 0.2619          | 0.3086          | 0.2148            | 0.2655             |
> | ViT+SVR    | 0.4345 | 0.1465  | 0.1882   | 0.3311       | 0.1483         | 0.1985          | 0.3311          | 0.1631            | 0.2104             |

---

### Official Review · Reviewer_7RBq · 2025-11-03

**Soundness:** 3
**Presentation:** 3
**Contribution:** 3
**Rating:** 6
**Confidence:** 5

**Summary:**

This paper proposes FLEX, the first dataset designed explicitly for fitness scenarios. It integrates multi-view videos, 3D poses, and other physiological data into an Annotated Action Quality Assessment (AQA) dataset. The multi-modal input, multi-view videos, and fine-grained annotations enhance AQA performance.

**Strengths:**

1. It integrates multi-view videos, 3D poses, surface electromyography (sEMG), heart rate/respiratory rate, and expert annotations, expanding the application scenarios of AQA.
2. A structured Fitness Knowledge Graph (FKG) is constructed. Beyond providing scalar quality scores, the paper introduces a hierarchical annotation system covering Action Keysteps, Error Types, muscle activation relationships, and corrective feedback, which is further formalized into a knowledge graph.

**Weaknesses:**

1.There is a lack of ablation experiments on the Fitness Knowledge Graph (FKG). Specifically, no ablation experiments were conducted to verify the specific contributions of each component in FKG—such as Action Keysteps, Error Types, and Feedback—to the performance of AQA or VideoQA. For instance, would the model performance drop significantly if only global quality scores (without the FKG structure) were used? This question directly relates to the practical necessity of FKG.
2.In the AQA experiments of this paper, multi-modal fusion adopts the method of "feature concatenation + relative muscle contribution calculation". This fusion approach is relatively simple, and there is a lack of analytical experiments on other feature fusion methods.
3.The computational cost of the experiments in this paper is high. Has consideration been given to how to reduce computational resources to meet the needs of lightweight deployment in the future?
4. In Table 3, there is limited data on the performance of other methods on the FLEX dataset. Methods like CoRe and TPT are relatively early fine-grained action modeling approaches, so it is advisable to incorporate more recent work and conduct basic experiments for analysis.

**Questions:**

Please see the weaknesses.

---

> ### Author Response · Authors · 2025-11-24
>
> We would like to thank the reviewer for their valuable feedback. We hope our responses have adequately addressed their previous concerns, and we look forward to addressing any further concerns. We take this as a great opportunity to improve our work and shall be grateful for any additional feedback you could give us.
>
> > There is a lack of ablation experiments on the Fitness Knowledge Graph (FKG). Specifically, no ablation experiments were conducted to verify the specific contributions of each component in FKG—such as Action Keysteps, Error Types, and Feedback—to the performance of AQA or VideoQA. For instance, would the model performance drop significantly if only global quality scores (without the FKG structure) were used? This question directly relates to the practical necessity of FKG.
>
> We sincerely thank the reviewer for the insightful questions regarding the Fitness Knowledge Graph (FKG) and its practical role/necessity. The role of FKG starts all the way from the beginning of the annotation process itself. **FKG is the knowledge backbone of the entire FLEX annotation and data-construction pipeline.** In the AQA setting, the FKG defines the hierarchical structure of each exercise: actions are decomposed into key steps, and for each step the FKG specifies typical error types, associated feedback, and penalty weights. Annotators are trained on this structure and rely on it—together with the composite scoring function—to produce consistent, interpretable global quality scores (see Sec. 3.3 and Appendix B.3). As a result, **the final global scores already integrate all factors defined by the FKG.** For this reason, _ablating on particular components or hierarchy levels of the FKG may not be applicable or meaningful_. For example, removing action key steps while keeping error types would invalidate the very definition of the action, whereas removing error types while keeping key steps would eliminate the notion of action quality. Either case breaks the annotation protocol itself and would no longer correspond to a valid task definition or scoring setting.
>
> In FLEX-VideoQA, FKG is used as the knowledge source for constructing question-answer (QA) pairs. Using the relations among actions, key steps, muscle groups, and error types encoded in FKG, we employ templates and a large language model to generate diverse fine-grained QA pairs and hierarchical question types (Sec. 4.2, Appendix B.3.5). Models are fine-tuned using the natural language questions and answers. Therefore, **the effect of FKG is embedded in these QA pairs and prompts**. An ablation that keeps "only global scores" for VideoQA would therefore require removing all FKG-driven fine-grained question categories and retaining only coarse, score-related questions, effectively changing the task definition rather than turning off a single model component.
>
> In summary, **the FKG is integral to how both the AQA scores and the VideoQA tasks are constructed**. Because the FKG governs the annotation protocol and the generation of QA pairs, ablations that selectively remove parts of it would invalidate the underlying data and task definitions rather than provide meaningful insights. We hope this adequately answers the reviewer's questions.
>
> > In the AQA experiments of this paper, multi-modal fusion adopts the method of "feature concatenation + relative muscle contribution calculation". This fusion approach is relatively simple, and there is a lack of analytical experiments on other feature fusion methods.
>
> We thank the reviewer for their insightful question. The primary goal of this work is to evaluate the intrinsic effectiveness of the FLEX multimodal data and the complementarity among its modalities. Accordingly, in our AQA experiments, **we intentionally adopt a simple fusion scheme to ensure that the observed gains reflect the value of the data and modality interactions themselves**, rather than the influence of complex model architectures.
>
> Using this simple fusion approach, we consistently observe performance improvements as additional modalities are incorporated, confirming that **the modalities in FLEX provide strong complementary information**. Because our design avoids dependence on any high-complexity fusion architecture, these findings are robust and architecture-agnostic. We acknowledge that more sophisticated fusion schemes could likely yield further gains, which would constitute an additional advantage.
>
> Given the dataset-centric nature of our study, it is important to demonstrate that **FLEX’s multimodal data is inherently valuable**, even under simple modeling conditions. We hope this clarifies the rationale behind our design choice and adequately addresses the reviewer’s concern.

---

> ### Author Response · Authors · 2025-11-24
>
> > The computational cost of the experiments in this paper is high. Has consideration been given to how to reduce computational resources to meet the needs of lightweight deployment in the future?
>
> We sincerely thank the reviewer for insightful questions about the computational cost of our experiments and the implications for future lightweight deployment.
>
> As our core research contribution lies in dataset construction and multimodal benchmarking, the current version of FLEX is primarily designed to **validate the effectiveness and complementarity of its multimodal data**, rather than to aggressively optimize model efficiency. To this end, we adopt relatively full multimodal configurations and perform large-scale, sufficiently long training and evaluation for tasks such as AQA, which indeed leads to higher computational cost. However, this heavy configuration is mainly intended to **characterize the upper-bound performance of FLEX in research settings**, and does not directly reflect the resource requirements of realistic deployment.
>
> From a practical deployment perspective, there are multiple well-established paths, such as reducing the number of input modalities, lowering input resolution, applying knowledge distillation, and performing online detection-and-evaluation schemes. All of these strategies can substantially reduce computational overhead without changing the FLEX annotation protocol or evaluation settings. In fact, the potential mobile and edge deployment was carefully considered from the outset: we deliberately included _a viewpoint captured from a mobile device_ (see Fig. 1 main paper), so that the community can easily explore and validate lightweight models on top of FLEX.
>
> In summary, although the present experiments focus on demonstrating the research value of FLEX as a multimodal benchmark, **the dataset has been designed with future lightweight deployment in mind and already includes resources to facilitate it.**
>
> > In Table 3, there is limited data on the performance of other methods on the FLEX dataset. Methods like CoRe and TPT are relatively early fine-grained action modeling approaches, so it is advisable to incorporate more recent work and conduct basic experiments for analysis.
>
> Thank you for the thoughtful feedback. As a dataset-focused paper, our primary goal was to demonstrate that the dataset enables meaningful learning and benchmarking, rather than exhaustively evaluating multiple models. Our AQA experiments are designed to use a diverse and representative set of baselines covering different modalities and modeling paradigms.
>
> Concretely, for the video modality we adopt CoRe and TPT, two widely used and actively employed fine-grained action modeling frameworks in the AQA literature, **as strong baselines to gauge the difficulty and learnability of FLEX in the visual setting**. For the 3D pose modality, we use both the classical ST-GCN and the more recent SkateFormer to cover graph-convolutional and Transformer-based spatio-temporal modeling paradigms. On top of CoRe, we then progressively add multiple modalities and measure performance under different modality combinations. As shown in Table 3, **all baselines exhibit consistent performance gains as more modalities are introduced**, indicating that the modalities in FLEX provide complementary information. In other words, the current experiments are sufficient to support our core conclusion that the multimodal information is valuable and effective.
>
> While additional baselines could provide further comparisons, each row in Table 3 represents training 20 models (~4000 GPU hours total), limiting our capacity. We therefore prioritized analyzing multimodal fusion with a representative model.

---

### Official Review · Reviewer_TsmB · 2025-11-10

**Soundness:** 4
**Presentation:** 3
**Contribution:** 3
**Rating:** 8
**Confidence:** 5

**Summary:**

The paper presents FLEX, a pioneering multimodal, multiview dataset for fitness Action Quality Assessment (AQA), featuring over 7,500 recordings from 38 subjects performing 20 exercises with synchronized RGB video, 3D pose, sEMG, and physiological signals. It includes a Fitness Knowledge Graph for interpretable scoring and introduces FLEX-VideoQA, a benchmark for hierarchical video question-answering to enable cross-modal reasoning and AI-driven fitness coaching.

**Strengths:**

- This paper proposes a new multimodal AQA dataset with over 7,000 samples, incorporating multimodal information such as synchronized surface EMG, RGB video, 3D joints, point clouds, and body metrics. Additionally, the dataset provides structured annotations and a Fitness Knowledge Graph. Based on the paper, the annotation quality appears high. Therefore, I believe that while the dataset is not particularly large, its superior quality still offers substantial value.
- Building on this dataset, the authors developed a fine-grained video question-answering benchmark with hierarchical questions, spanning from coarse action recognition to detailed error diagnosis and causal feedback generation.
- Furthermore, the authors explored multimodal, cross-modal, and biomechanically-oriented representation learning.

**Weaknesses:**

- The proposed dataset includes information from multiple modalities, but the experimental section does not fully leverage these multimodal elements, and the inter-modal associations require deeper investigation. As a result, while the dataset inherently benefits from multimodality, this advantage is not adequately demonstrated in the experiments.
- Although the authors introduce a new dataset and some methods, it is worth considering whether these methods possess general applicability and can be extended to other domains. After all, the dataset's scenarios are relatively narrow, focusing solely on fitness-related video QA tasks.
- The paper does not clarify whether the proposed dataset will be open-sourced. If it is not made publicly available, the paper's contributions will be significantly undermined, as other researchers would be unable to access it for further studies and validation.

**Questions:**

See Weaknesses.

---

> ### Author Response · Authors · 2025-11-24
>
> We would like to thank the reviewer for their valuable feedback. We hope our responses have adequately addressed their previous concerns, and we look forward to addressing any further concerns. We take this as a great opportunity to improve our work and shall be grateful for any additional feedback you could give us.
>
> > The proposed dataset includes information from multiple modalities, but the experimental section does not fully leverage these multimodal elements, and the inter-modal associations require deeper investigation. As a result, while the dataset inherently benefits from multimodality, this advantage is not adequately demonstrated in the experiments.
>
> Thank you very much for the reviewer’s insightful question regarding the multimodal design of FLEX and the extent to which the experiments leverage the available modalities. We fully agree that multimodal signals and inter-modal relationships are central to FLEX’s purpose. Below, we clarify how our experiments already demonstrate meaningful multimodal benefits, how we initiate inter-modal analysis, and how FLEX provides a strong foundation for deeper multimodal research.
>
> **1. Multimodal advantages are directly validated in our AQA experiments.** In the AQA task, we evaluate three core modalities—RGB video, 3D pose, and sEMG—each of which captures complementary aspects of execution quality: visual dynamics, kinematic precision, and neuromuscular activation. To quantify the contribution of each channel, we perform unimodal evaluations and then progressively combine modalities. As shown in Table 3, model performance improves consistently when adding 3D pose and subsequently sEMG to a video baseline. These results confirm that:
>
> - FLEX’s multimodal design yields significant and measurable performance gains, and
>
> - the modalities encode complementary information crucial for fine-grained action quality assessment.
>
> This directly demonstrates that FLEX’s multimodal nature is not only beneficial but empirically validated within the experiments.
>
> **2. Inter-modal associations: visible → invisible (cross-modal prediction from video to sEMG).** To further examine inter-modal relationships, particularly between visible and invisible signals, we conduct cross-modal representation learning experiments using paired video–sEMG sequences. FLEX is, to the best of our knowledge, the first dataset to synchronously record high-precision sEMG and multiview video, enabling us to train a Video2EMG model that predicts muscle activation patterns solely from visual input (Table 5). This is a challenging real-world problem due to the difficulty of collecting sEMG signals.
>
> Our results show that the Video2EMG models can accurately regress sEMG values for most samples with controlled variance, demonstrating a strong cross-modal mapping between visual observations and internal neuromuscular activity. This provides concrete evidence of tight coupling between modalities in FLEX and points toward new paradigms for inferring physiological and biomechanical states from video alone. These experiments directly address the reviewer’s request for deeper investigation into inter-modal associations.
>
> **3. Use and role of auxiliary physiological signals.** Heart rate and breath rate are included not only for annotation refinement but also as future modeling targets. At this stage, they serve two immediate roles:
>
> - Ensuring safety and load validity during high-intensity data collection (monitoring heart-rate responses to confirm ~80% of individual 1RM), and
>
> - Supporting annotation of breathing-related error types, which depend on bracing and intra-abdominal pressure patterns.
>
> Although we have not yet integrated these channels into our predictive models, they are fully synchronized with all other modalities and provide a natural path toward physiology-informed AQA—e.g., multi-task modeling of fatigue state, breathing strategy, and the alignment between physiological effort and biomechanical execution. These directions are enabled by FLEX and form part of the long-term research agenda.
>
> **4. Multimodal VideoQA further validates cross-modal reasoning capability.** In the VideoQA setting, the combination of detailed annotations, visual signals, and language descriptors allows models to perform factual and reasoning-based question answering. This explicitly leverages both visual–linguistic grounding and action semantics derived from our Fitness Knowledge Graph. Appendix C.2.3 provides additional analysis showing how multimodal annotations facilitate joint reasoning across modalities, further confirming FLEX’s value for multimodal understanding beyond AQA. [contd. below]

---

> ### Author Response · Authors · 2025-11-24
>
> [contd.]
> **5. FLEX is a long-term multimodal benchmark; current experiments establish initial milestones.** FLEX provides synchronized RGB/multiview video, 3D pose, sEMG, heart rate, breath rate, natural-language descriptions, and a structured action knowledge graph. While our present experiments focus on several key modalities most central to AQA and cross-modal representation learning, the dataset was intentionally designed to support a broad spectrum of multimodal research problems. These include:
>
> - action quality assessment,
>
> - cross-modal prediction (e.g., Video→EMG, Video→Breath),
>
> - multimodal and V+L reasoning,
>
> - multiview consistency learning,
>
> - biomechanics-grounded analysis, and
>
> - self-supervised learning from heterogeneous signals.
>
> Our current results demonstrate the dataset’s multimodal benefits across both AQA and cross-modal prediction, while leaving substantial room for deeper multimodal investigations in future work.
>
> **Summary.** In summary, our experiments do leverage FLEX’s multimodal structure, demonstrating clear benefits in AQA and strong cross-modal mapping in the Video2EMG task. We also provide initial analyses of inter-modal relationships and show that FLEX supports more advanced multimodal research going forward. We appreciate the reviewer’s suggestions and hope this clarifies both the demonstrated multimodal utility of our experiments and the broader long-term capabilities FLEX enables.
>
> > Although the authors introduce a new dataset and some methods, it is worth considering whether these methods possess general applicability and can be extended to other domains. After all, the dataset's scenarios are relatively narrow, focusing solely on fitness-related video QA tasks.
>
> We sincerely thank the reviewer for insightful questions regarding whether our methods generalize beyond the fitness domain. We address this at the levels of methodology, representation, and knowledge construction.
>
> **1. Domain-transferable knowledge-driven pipeline.** While FLEX leverages a Fitness Knowledge Graph (FKG), the underlying pipeline—knowledge integration, action decomposition, semantic structuring, and interpretable scoring (Sec. 3.3.1)—is fully domain-agnostic. The same procedure can be applied to any structured human-motion domain, such as rehabilitation, surgical training, industrial manufacturing, elite sports, or laboratory procedures, to construct domain-specific action knowledge graphs and fine-grained evaluation schemes.
>
> **2. Methods are not biased toward fitness-specific assumptions.** Our models operate on general motion patterns and multimodal cues (RGB, multiview geometry, pose, EMG, physiology, language), none of which embed fitness-specific priors. These sensing modalities are standard in healthcare, biomechanics, robotics, and workplace analytics. Thus, our temporal reasoning modules, multimodal alignment strategies, and fusion mechanisms can be directly applied to other settings that rely on similar multimodal signals.
>
> **3. Generality arises from fine-grained depth, not superficial breadth.** In fine-grained action quality assessment, generalizable insights often come from analyzing a structured domain at high temporal and semantic resolution, rather than from covering many unrelated domains superficially. FLEX explores a single domain deeply, capturing nuanced execution variability, procedural substructure, temporal degradation due to fatigue, and rich cross-modal correspondences. This high-resolution representation space—analogous to increasing granularity within a continuous interval—reveals fundamental principles of action quality that are not specific to fitness.
>
> Moreover, this fine-grained structure can be lifted and applied to other domains that similarly depend on precise, multi-stage procedural understanding. Examples include surgical skill assessment (tool kinematics, step transitions, error modes), rehabilitation (compensatory patterns, control deficits), industrial manufacturing (task sequencing, ergonomic quality), and competitive sports (technique phases, force-generation patterns). These areas share the same core challenges FLEX is designed to model.
>
> **4. FLEX captures generalizable technical challenges.** Although the domain is fitness, the challenges are universal: temporal reasoning over complex human motion, multimodal representation learning, cross-modal prediction (e.g., Video→EMG), viewpoint and subject generalization, and robustness to execution variability. Our results show that state-of-the-art AQA models and modern VLMs perform well under simple conditions but experience significant drops under cross-view and cross-subject splits—highlighting general methodological challenges rather than domain-specific ones. [contd. below]

---

> > ### Comment · Reviewer_TsmB · 2025-11-26
> >
> > I thank the authors for their detailed response.
> >
> > - As shown in Table 3, the most important modalities are the multi-view videos and the skeleton information, while adding EMG brings only very minor improvements. This indicates that modalities other than multi-view videos and skeleton information do not appear critical for AQA. Since existing AQA methods have fully explored the use of both video and skeleton information, the experiments in this paper do not seem to bring significant new findings to the community. However, I believe constructing such a multi-modal dataset is still a valuable contribution.
> >
> > - I agree with the authors' discussion regarding generalization and application in other domains. However, it would be more persuasive if the authors could provide direct experiments to verify the generalizability of the method.
> >
> > Overall, the authors have partially addressed my concerns.

---

> ### Author Response · Authors · 2025-11-24
>
> [contd.]
> **5. FLEX offers depth and multimodal richness that support broad research directions.** FLEX includes 20 complex procedural exercises—the largest structured collection in this domain—and is the first dataset to pair synchronized multiview video, high-fidelity sEMG, physiological signals, and smartphone-view footage. Its multimodal annotations enable research directions relevant to many fields, including self-supervised multimodal learning, biomechanically grounded reasoning, fatigue modeling, injury-risk anticipation, and error detection. This depth and structure complement existing datasets such as FineGym and Human3.6M, which do not provide this level of fine-grained multimodal grounding.
>
> **Summary.** Although instantiated in fitness, both the methods and the knowledge-driven annotation pipeline are intentionally domain-general. FLEX offers a high-resolution testbed for modeling complex procedural actions, and the resulting insights and representations naturally transfer to domains such as rehabilitation, surgery, sports science, and industrial task analysis.
>
> *We have further elaborated on the transferability and potential extension scenarios of FLEX in the revised manuscript.*
>
> > The paper does not clarify whether the proposed dataset will be open-sourced. If it is not made publicly available, the paper's contributions will be significantly undermined, as other researchers would be unable to access it for further studies and validation.
>
> **YES, the full dataset along with the full codebase will be open-sourced as soon as the paper is accepted.** The dataset will be hosted on a widely used and stable academic data platform and will provide support to users. To clarify, we have not released the dataset currently to comply with the ICLR anonymity policy. We would like to reiterate that the FLEX dataset will be made publicly and unconditionally available to the community as soon as the paper is accepted.

---

> ### Author Response · Authors · 2025-11-28
>
> > As shown in Table 3, the most important modalities are the multi-view videos and the skeleton information, while adding EMG brings only very minor improvements. This indicates that modalities other than multi-view videos and skeleton information do not appear critical for AQA. Since existing AQA methods have fully explored the use of both video and skeleton information, the experiments in this paper do not seem to bring significant new findings to the community. However, I believe constructing such a multi-modal dataset is still a valuable contribution.
>
> Thank you. Your observations highlight an important point: although some modalities contribute more prominently, the value of combining multiple modalities—especially those like EMG—extends beyond quantitative performance gains, as outlined below.
>
> **Quantitative vs. Qualitative importance.** While the quantitative improvement from EMG is modest, the **qualitative** information it provides is fundamentally distinct from visual or kinematic modalities. EMG captures **muscle-specific activation patterns**—such as co-contraction, fatigue signatures, imbalance, or compensatory muscle usage—that cannot be inferred reliably from video or skeletal data, even when those modalities achieve higher numerical performance. Thus, a modality’s value should not be judged solely by its effect on aggregate accuracy scores; it also lies in the unique **physiological insights** it contributes.
>
> **Complementary rather than dominant role of EMG.** Consistent with this view, our results show that EMG offers small but consistent gains. This supports our hypothesis that subtle neuromuscular cues complement, rather than replace, visual information.
>
> **Enabling Precision healthcare and domain-specific applications.** EMG’s qualitative value also positions it as a key modality for precision healthcare applications—such as rehabilitation, motor-control analysis, and ergonomics—where understanding which muscles are activated, and whether they are activated correctly, is more critical than aggregate performance metrics. FLEX establishes a foundation for such research by providing synchronized muscle-level data aligned with high-fidelity motion signals.
>
> **Contribution to future research.** By offering a multimodal dataset that includes both dominant and subtle modalities, FLEX enables future exploration of biomechanical representation learning, multimodal fusion strategies, robustness under occlusion, and muscle-aware AQA. These avenues are not feasible with existing datasets and create meaningful opportunities for the community.
>
> We will revise the manuscript to clarify these points and more explicitly articulate why modalities like EMG provide distinctive and valuable contributions beyond what quantitative metrics alone can capture. We hope this helps further address your outstanding concern.

---

> ### Author Response · Authors · 2025-11-28
>
> > I agree with the authors' discussion regarding generalization and application in other domains. However, it would be more persuasive if the authors could provide direct experiments to verify the generalizability of the method.
>
> Thank you for the valuable suggestion. We would like to emphasize an important contextual point: **to the best of our knowledge, no publicly available multimodal AQA dataset currently provides synchronized multi-view video, skeletal motion, and complementary physiological sensor streams with the level of granularity, scale, and annotation rigor offered by FLEX.** This absence of suitable datasets and benchmarks is precisely what motivated us to develop FLEX. Thus, experimentally verifying the generalizability of the method is currently not feasible because of the lack of multimodal physiological AQA datasets outside of FLEX.
>
> We take the first step in this direction. By releasing FLEX with standardized splits, annotation rules, and a coherent multimodal organization, we aim to enable such cross-domain generalization studies, not only for our method but also for the broader community. FLEX provides, for the first time, an AQA platform where multimodal fusion, modality ablation, and cross-modal representation learning can be systematically explored under controlled and reproducible conditions.
>
> Moreover, as described previously, our _knowledge-driven scoring pipeline_ is inherently domain-agnostic: the construction of action-level ontologies, error taxonomies, and compositional scoring rules can be instantiated in other motion domains simply by substituting the underlying professional knowledge source. Similarly, our modeling framework and multimodal fusion design do not encode any domain-specific biases toward fitness exercises. The methodological components—temporal modeling, cross-modal alignment, hierarchical scoring, and error-aware reasoning—are directly transferable to applications such as rehabilitation assessment, sports analytics, and industrial human–robot collaboration.
>
> Thus, while FLEX itself is grounded in fitness training, the _principles_, the _pipeline_, and the _architecture_ are broadly generalizable and applicable to other domains (surgical domain, rehabilitation, competitive sports, dance, etc.). To further illustrate the generalizability of our pipeline, we provide a **concrete example** in a domain that is semantically different from fitness: **surgical skill assessment**, a key area in medical training.
>
> Surgical training involves evaluating fine-grained motor actions, adherence to procedural steps, ergonomic efficiency, and error patterns—challenges that closely parallel those addressed by our framework. Applying our pipeline to this domain requires only substituting the domain-specific knowledge sources, while all technical components remain unchanged:
>
> - **Domain Knowledge Source**:
>   Replace the fitness-oriented knowledge graph with well-established surgical standards such as the *Objective Structured Assessment of Technical Skills (OSATS)*, *Global Operative Assessment of Laparoscopic Skills (GOALS)*, or other procedure-specific surgical competency rubrics.
>
> - **Action Ontology & Error Taxonomy**:
>   Construct a surgical action graph describing key procedural steps (e.g., incision, tissue handling, knot tying) and their associated error types (e.g., excessive force, imprecise instrument motion, improper angle). These map directly onto our method for building action hierarchies and error annotation rules.
>
> - **Compositional Scoring**:
>   Surgical scoring systems decompose performance into factors such as economy of motion, precision, and smoothness—an ideal fit for our hierarchical and compositional scoring design. The same rule-based scoring logic applies seamlessly.
>
> - **Multimodal Alignment**:
>   Surgical settings often use video, tool-motion trajectories (from robotic systems or instrument trackers), motion sensors, and even EMG for tremor detection. These modalities align directly with the multimodal inputs modeled in FLEX and can be fused using the same alignment and representation learning framework.
>
> - **Modeling Pipeline**:
>   Our modules do not assume any fitness-specific semantics, so they can be transferred to surgical tasks without modification.
>
> We hope this example demonstrates concretely that the pipeline’s generalization is not hypothetical: **the knowledge-driven, modular, and multimodal design naturally extends to domains such as surgery**, where fine-grained evaluation is critical and multimodal signals are readily available.
>
> We hope that the above discussion and example help clarify how our framework generalizes beyond the fitness domain and satisfactorily address the reviewer’s comments. We look forward to your feedback and would be happy to provide any further clarification. Thanks again.

---

### Author Response · Authors · 2025-11-24

We would like to thank the reviewer for their valuable feedback. We hope our responses have adequately addressed their previous concerns, and we look forward to addressing any further concerns. We take this as a great opportunity to improve our work and shall be grateful for any additional feedback you could give us.

We have uploaded the revised PDF, in which all newly added content is highlighted in blue for ease of review. Individual responses are in the following.

---

> ### Comment · Area_Chair_7Q6g · 2025-11-25
> **Encourage discussions**
>
> Hi all,
>
> The authors have submitted their responses. Please take a moment to review them and see if they address your concerns.
>
> Your thoughtful input is essential for a successful reviewing process and is greatly appreciated.
>
> Many thanks,
>
> Area Chair

---

### Author Response · Authors · 2025-12-03
**Summary of Responses and Revisions**

Dear Area Chair and Reviewers,

We sincerely thank the reviewers for their careful reading of our manuscript and for the constructive comments provided. These suggestions are invaluable for improving the quality of our work and refining its presentation. We also appreciate the area chair’s time and effort devoted to our submission.

In **summary**, we addressed some of the main concerns as follows. **Detailed answers** provided in Reviewers' sections.

>**[S1] What are the scope, unique value, and generality of the FLEX dataset?**

We clarified that FLEX is a well-defined dataset centered on weight-loaded fitness actions, offering the **largest collection of 20 complex exercise procedural actions**, each exhaustively structured through our fine-grained **Fitness Knowledge Graph**. It is the **first dataset** to pair **high-fidelity sEMG** signals with synchronized **multiview** video and physiological signals, and uniquely includes a **smartphone view** to support **lightweight, in-the-wild deployment**. FLEX is meticulously **annotated by multiple fitness and sports science experts**. FLEX also captures the **temporal degradation of action quality**, enabling injury-prevention and early-failure analysis—capabilities absent in prior datasets.

>**[S2] FLEX is a long term project supporting multiple tasks.**

FLEX enables a broad spectrum of motion/action research problems, including: **(a)** multimodal action representation learning; **(b)** cross-modal prediction (Video→EMG); **(c)** biomechanically grounded analysis from synchronized kinematics and EMG; **(d)** action reasoning and interpretable AQA via the FKG; **(e)** video–language reasoning, causal feedback, and VideoQA; **(f)** self-supervised learning across multiple modalities; and **(g)** injury and early-failure prediction.

>**[S3] Rigor of the annotation process and expert validation.**

**Sec. 3.3.2** provides a detailed description of the full data annotation pipeline, including *annotator recruitment, training, action segmentation, group-wise annotation, and multi-stage quality checks*. The annotation protocol is derived from **multiple authoritative standards and textbooks**. Each sample is independently annotated by **three master-level sports science experts** and subsequently validated by **three doctoral-level experts**, ensuring rigorous, consistent, and reliable scoring. Further details are provided in **Appendix B.3**.

>**[S4] Can ablations be performed where one level of the Fitness Knowledge Graph (FKG) hierarchy (e.g., Action Keysteps, Error Types, Feedback) is removed while keeping the remaining levels unchanged**

Thank you for the suggestion. We examined whether ablations on single FKG levels (e.g., removing Action Keysteps or Error Types) could be meaningfully defined. However, the FKG is a strictly hierarchical structure—**FKG hierarchy is intrinsically interdependent**. The FKG structure—Action Keysteps → Error Types → Feedback—means that each level is defined directly in terms of the one before it. Removing key steps while retaining error types breaks the definition of the action itself, and removing error types eliminates the basis for action-quality scoring. Either modification invalidates the annotation protocol and no longer corresponds to a meaningful or well-defined task.

Moreover, the **FKG is not an optional component but the knowledge backbone guiding the entire annotation pipeline**, including action definitions, error taxonomy, composite scoring, and QA construction. Because the hierarchy is structurally coupled, isolated ablations on individual levels are conceptually incompatible. We have added this clarification explicitly in the revised paper.

>**[S5] Benchmark coverage and challenge.**

With **20 complex procedural fitness actions**, FLEX provides a broad and challenging suite of benchmarks.

1. **AQA Benchmarks:** FLEX includes Vanilla, Cross-Subject (CS), and Cross-View (CV) splits. While prior work typically considers only Vanilla, CS and CV directly test model generalization. We observe that SOTA AQA models perform reasonably on Vanilla but drop to **below 50%** on CS and CV. Surprisingly, even SOTA VLMs—despite extensive pretraining—perform poorly on Vanilla AQA and fail to generalize.

2. **FLEX-VideoQA Benchmark:** Designed for supervised fine-tuning and evaluation of VLMs on fine-grained action understanding. Off-the-shelf VLMs perform poorly, and although fine-tuning improves results, a **large performance gap** remains.

3. **Video→EMG Prediction:** This task is especially challenging. Across multiple baselines, even the best model achieves **well under 50% of ideal performance**, highlighting the difficulty of inferring deep physiological signals from video alone.

---

> ### Author Response · Authors · 2025-12-03
> **Summary of Responses and Revisions**
>
> >**[S6] Clarification of evaluation metrics and technical definitions.**
>
> We add explicit pointers in the main text to the complete mathematical definitions of the evaluation metrics in the **Appendix C.1.2** and clearly state the normalized AQA scoring range (**0–100, higher is better**) (**Appendix B.3.4**). In addition, we clarify that the computation and evaluation protocol for VideoQA feedback generation are explicitly defined and reported in the paper, thereby removing ambiguities regarding these technical details.
>
> >**[S7] Justification and Analysis of Multi-view and Multi-modal Settings.**
>
> Our experiments show **clear and consistent gains when adding additional views and modalities**. Moving from a **single view** to **multi-view** inputs improves **3D body understanding**, and adding explicit **pose** information further boosts performance. Incorporating **sEMG** provides complementary **internal *individual* muscle-activation signals** that substantially improve AQA and Video→EMG performance, demonstrating the value of combining **external motion cues** with **internal physiological measurements**.
>
> Beyond AQA, FLEX’s multimodal suite—*RGB, 3D pose, sEMG, physiological signals, Fitness Knowledge Graph,* and *Language*—supports **self-supervised learning, cross-modal representation learning** *(e.g., Video→EMG),* and **VideoQA/VLM-based reasoning**, highlighting the importance of a multi-view, multi-modal design.
>
> >**[S8] Will the dataset be open sourced?**
>
> **Yes. The full dataset, along with the full codebase, will be open-sourced** as soon as the paper is accepted. We have provided anonymized examples and subset code for the main experiments in the supplementary material to enable reviewers to inspect the data quality.
>
> >**[S9] Why did you choose weight-loaded exercise actions?**
>
> **Sec. 1** and **Appendix A** explain our **motivation**, which we summarize here. Weight-loaded exercises represent some of the most **demanding and biomechanically complex human actions**: they require *high coordinated multi-joint control, engage multiple muscle groups simultaneously, and carry substantially higher injury risk than self-loaded motions*. Our *20 selected actions load the most injury-prone joints (shoulders, knees, hips, spine, wrists, ankles) and capture movement patterns absent in existing AQA datasets*. These **load-induced biomechanical** and **perceptual complexities** are fundamentally *absent from existing datasets*, which *cannot capture* the **altered dynamics** introduced by **external weight**.
>
> *Loaded movements also introduce distinct challenges—higher joint moments, load-dependent torque distributions, and non-visible neuromuscular activation*—that **fundamentally change body dynamics and error patterns**. These factors make weight-loaded exercises a scientifically essential modeling domain beyond what existing datasets represent. We have made this motivation more explicit in the revised main text.
>
> ***
> **Revised manuscript.** We have uploaded the revised PDF, in which all newly added content is highlighted in blue for ease of review.
>
> Thank you,
>
> Authors

---

### Meta-Review · Area_Chair_Hr3x · 2025-12-29

**Summary:**

This work introduces a multimodal dataset for fitness Action Quality Assessment (AQA). It includes multiviews, RGB videos, 3D pose, and surface electromyography (sEMG) data. The authors propose using this data to improve the assessment of weight-loaded exercises and introduce benchmarks for the relevant tasks like VideoQA and sEMG prediction. Reviewers acknowledged taht the data collection effort is extensive and the inclusion of physiological signals is potentially valuable for the community. AC found that the current analysis and experiments of multimodal learning or modality complementary effect are not sufficient to deeply verify the gains from collected multiple modalities. Another weakness, as highlighted by Reviewer vzMp, is that the paper does not demonstrate the practical value or necessity of these additional modalities. The experimental results, specifically in Table 3, show only marginal performance gains when combining sEMG with video and pose data using simple feature concatenation. This superficial treatment of multimodal fusion fails to provide convincing evidence that the physiological signals offer significant complementary information for the AQA task. Therefore, this paper would benefit more from comprehensive analysis.

**Reviewer Concerns:**

The authors addressed concerns about the dataset's difficulty by adding new experimental settings, such as testing on cross-subject and cross-view, which satisfied Reviewer `3nbD`'s request for harder benchmarks. They also provided explanations to Reviewer `TsmB` regarding the qualitative value of the muscle and heart rate data for safety and detailed analysis. However, the group remained largely unconvinced by the justification for the simple method used to combine these different data types. Reviewer `vzMp` and Reviewer `gH6A` maintained that the authors failed to prove why such a complex recording setup was necessary, given the marginal performance gains and the lack of comprehensive comparisons to existing work.

**Reviewer Scores:**

During the review phase, Reviewer `TsmB` was the most positive supporter due to the impressive scale and detailed annotation of the dataset. Reviewer `3nbD` and Reviewer `7RBq` both gave borderline scores, acknowledging the quality of the data collection while noting that the experimental validation was somewhat weak. On the negative side, Reviewer `vzMp` maintained a score of 4, arguing that the practical utility of the additional modalities was unproven, and Reviewer `gH6A` gave the lowest score, citing serious unresolved issues with the literature review and unfair comparisons.

---

### Decision · Program_Chairs · 2026-01-26

Reject